# Disordered enthalpy–entropy descriptor for high-entropy ceramics discovery

Simon Divilov[1,2,14], Hagen Eckert[1,2,14], David Hicks[1,2,14], Corey Oses[1,2,14], Cormac Toher[2,3,14], Rico Friedrich[2,4,5], Marco Esters[1,2], Michael J. Mehl[1,2], Adam C. Zettel[1,2], Yoav Lederer[2,6], Eva Zurek[7], Jon-Paul Maria[8], Donald W. Brenner[9], Xiomara Campilongo[2], Suzana Filipović[10,11], William G. Fahrenholtz[10], Caillin J. Ryan[12], Christopher M. DeSalle[12], Ryan J. Crealese[12], Douglas E. Wolfe[12], Arrigo Calzolari[1,2,13] & Stefano Curtarolo[1,2✉]

The need for improved functionalities in extreme environments is fuelling interest in high-entropy ceramics[1–3]. Except for the computational discovery of high-entropy carbides, performed with the entropy-forming-ability descriptor[4], most innovation has been slowly driven by experimental means[1–3]. Hence, advancement in the field needs more theoretical contributions. Here we introduce disordered enthalpy–entropy descriptor (DEED), a descriptor that captures the balance between entropy gains and enthalpy costs, allowing the correct classification of functional synthesizability of multicomponent ceramics, regardless of chemistry and structure. To make our calculations possible, we have developed a convolutional algorithm that drastically reduces computational resources. Moreover, DEED guides the experimental discovery of new single-phase high-entropy carbonitrides and borides. This work, integrated into the AFLOW computational ecosystem, provides an array of potential new candidates, ripe for experimental discoveries.

High-entropy ceramics is an emerging, yet blooming class of materials owing to its ever-growing set of applications, such as thermal barrier protection, wear- and corrosion-resistant coatings, thermoelectrics, batteries and catalysts[1,2]. They stand out for their mechanical properties, thermal and chemical stability, and high-temperature plasmonic resonance[1–3,5]. However, most single-phase high-entropy ceramic discoveries have been slowly driven by experiments[1–3] and only high-entropy carbides have been proposed computationally[4]. Here, we overcome this impasse by defining functional synthesizability and building a descriptor that quantifies it.

## Defining functional synthesizability

Synthesizability depends on both the material and the synthesis conditions, and the user needs to explore the appropriate path across these two degrees of freedom. The conundrum lies in the observer's reference. Usually, the priority is put on 'synthesizability as an intrinsic material property' with synthesis conditions to be determined later; that is, from the material's point of view. Unfortunately, this approach, along with the lack of consensus about 'synthesizability'[6–13], does not translate into an operative procedure for the synthesis, critical for autonomous materials discovery[14]. Instead, to maximize the outcome, we focus on the complementary direction, or the process's point of view; that

is, starting from a chosen process, we build a descriptor to establish which materials are synthesizable with it. We define this as 'functional synthesizability', to specify that it is intrinsically a function of the chosen manufacturing process. In the present case – transition-metal disordered ceramics – the process is hot-pressed sintering with all its various implementations[2], and the classes of materials are high-entropy carbides, carbonitrides and borides.

## Descriptor for functional synthesizability

For our purposes, the descriptor must relate to an observable quantity and also be able to answer tangible synthesis questions[14,15]. To tackle high-entropy materials, an effective descriptor should consider the 'entropic gain in generating disorder', which, through Boltzmann's view, is represented by the degeneration of the states (in a discretized population), or more generally, by the thermodynamic density of states spectrum $\Omega(E)\delta E$ (in a continuum population). This is the essence of the entropy-forming-ability (EFA) descriptor[4], which ranked the candidates' solid-solution synthesizability by associating the entropic gain to the variance (second momentum) of $\Omega$ (ref. 4). When EFA was introduced in 2018, it worked flawlessly for high-entropy carbides, but had difficulties in systems characterized by non-homogeneous enthalpy landscapes[16]. In addition, there were very little available ab initio enthalpy

[1]Department of Mechanical Engineering and Materials Science, Duke University, Durham, NC, USA. [2]Center for Autonomous Materials Design, Duke University, Durham, NC, USA. [3]Department of Materials Science and Engineering and Department of Chemistry and Biochemistry, The University of Texas at Dallas, Richardson, TX, USA. [4]Institute of Ion Beam Physics and Materials Research, Helmholtz-Zentrum Dresden-Rossendorf, Dresden, Germany. [5]Theoretical Chemistry, Technical University of Dresden, Dresden, Germany. [6]Department of Physics, NRCN, Beer-Sheva, Israel. [7]Department of Chemistry, State University of New York at Buffalo, Buffalo, NY, USA. [8]Department of Materials Science and Engineering, The Pennsylvania State University, University Park, PA, USA. [9]Department of Materials Science and Engineering, North Carolina State University, Raleigh, NC, USA. [10]Department of Materials Science and Engineering, Missouri University of Science and Technology, Rolla, MO, USA. [11]Institute of Technical Sciences of the Serbian Academy of Sciences and Arts, Belgrade, Serbia. [12]Applied Research Laboratory, The Pennsylvania State University, University Park, PA, USA. [13]CNR-NANO Research Center S3, Modena, Italy. [14]These authors contributed equally: Simon Divilov, Hagen Eckert, David Hicks, Corey Oses, Cormac Toher. ✉e-mail: stefano@duke.edu

convex-hull data for high-entropy ceramics in the various databases, so the evaluation of the 'enthalpy costs to generate disorder' could not be performed in its full extent. In fact, the entropy gain can be estimated through the statistical ensembles of configurational formation enthalpies weighted with their degeneracies (the reference ground state is not necessary owing to the differential nature of $\Omega$), the same cannot be said of the enthalpy cost: the latter requires the ensemble of the distances of such configurational formation enthalpies to the convex hull, the Gibbs free-energy line of the stable configurations (the underlying ground state). As such, an effective descriptor balancing 'entropic gain' versus 'enthalpy cost' for solid-solution formation requires two ingredients: the first is given by an ensemble quantifying the spread of $\Omega(E)$ to capture how many configurations accumulate around $E$, and the second is an ensemble quantifying the expectation of the distance $\Delta H$ of such phases from the ordered ground-state hull. Both quantities can now be quantified for several classes of high-entropy ceramics, as we have populated the aflow.org repository with ample and appropriate thermodynamic data[17] and developed the appropriate computational modules for recursive phase diagram searches[18]. Now, we can introduce a new forming ability descriptor, which includes the statistical information of the thermodynamic density of states. The DEED allows us to classify the functional synthesizability of multicomponent ceramics, including the ones with non-flat enthalpy landscapes. DEED uses the first and second momenta of $\Omega$, associating the enthalpy loss to the former and the entropy gain to the inverse of the latter, respectively. DEED is intrinsically connected to an observable. In fact, by DEED being defined as the ratio between entropy gain and enthalpy loss, it retains the physical meaning of the inverse of an order and/or disorder temperature, the cross-over between two ideal environments: a perfectly ordered and a perfectly disordered one. As such, the inverse of DEED is correlated to the miscibility gap critical temperature and can be used as a descriptor for functional synthesizability for hot-pressed sintering. Large values of DEED will indicate small miscibility gap temperature, easy to overcome by the sintering temperature for the formation of the solid solution. Conversely, small values will indicate large critical temperatures, possibly above the synthesis temperature, resulting in nucleation of multiphase microstructures, instead of the formation of a solid solution. The threshold for functional synthesizability predictions of DEED can be found self-consistently from available experiments, and then extrapolated to predict the formation, or not, of new solid solutions. In essence, different processes, structures and chemistries can be captured by the DEED framework, as long as functional synthesizability can be mapped into a logical relationship between synthesis and miscibility gap temperatures. Here we show that DEED accelerates the computational discovery of novel disordered ceramics by allowing rapid prescreening, avoiding expensive experimental preparation and enabling structure and property predictions[15].

To validate the practical application of DEED, we chose carbides, carbonitrides and borides because of their high-temperature, high-hardness and chemical-resistance applications in extreme environments. First, we characterized 952 disordered ceramics with the aflow++ ab initio framework[17,18] (Methods). Of these, 548 were metal-carbides, 70 metal-carbonitrides and 334 metal-borides (complete lists are in Extended Data Table 1–3).

Second, we compared our computational results with experimental findings. Among our calculations, only 46 systems had been previously studied experimentally. Within the unexplored compositional space, on the basis of the DEED ranking, we chose as testbeds nine new carbonitrides and eight new borides. DEED correctly predicted the functional synthesizability of all 17 systems. The single-phase compositions we discovered are (HfNbTiVZr)CN, (HfNbTaTiV)CN, (NbTaTiVZr)CN, (HfTaTiVZr)CN, (MoNbTaTiZr)CN, (HfMoNbTaZr)B$_2$, (HfNbTaTiV)B$_2$, (CrMoTiVW)B$_2$ and (CrHfNbTiZr)B$_2$.

Ultimately, the comparison between the DEED ranking and the experimentally validated synthesizability corroborates the accuracy

of the descriptor and even suggested systems with formation of microstructures (pearlite). All observed results are listed in Table 1 and visualized in Figs. 2 and 3. Our predictions provide an array of unexplored candidates for future synthesis.

## Discussion

### DEED descriptor

Enthalpy cost and entropy gain in a disordered system are characterized through the thermodynamic density of states, $\Omega(E)\delta E$, measuring the number of configurationally excited states within $[E, E + \delta E]$. For our large number of chemical compounds, to obtain $\Omega$, we find more precise techniques computationally unfeasible, such as cluster expansion[19] or Lederer–Toher–Vecchio–Curtarolo methods[20]. As such, we approximate $\Omega$ with AFLOW partial occupation (POCC)[21], which models a random alloy as an ensemble average of ordered representative states called POCC tiles calculated by density functional theory (DFT). POCC's success has been extended to studies beyond configurational issues[5,22]. Further details are in the Methods section.

The density of states allows us to define a probability distribution of a function $f(E)$ through the renormalization $\int f(E)\Omega(E)dE/\eta$, where $\eta \equiv \int \Omega(E)dE$ is the renormalization constant. This formalism is used to extract the statistical momenta of the POCC tiles, defined as:

$$\langle \Delta H_{\text{hull}} \rangle_\Omega \equiv \frac{1}{\eta} \int [H_f(E) - H_{\text{f,hull}}]\Omega(E)dE, \tag{1}$$

$$\sigma_\Omega^2[H_f] \equiv \frac{1}{\eta} \int [H_f(E) - \langle H_f \rangle_\Omega]^2 \Omega(E)dE, \tag{2}$$

where $H_f$ and $H_{\text{f,hull}}$ are the DFT formation energies of the POCC tiles and the convex hull, respectively.

For an order–disorder solid-solution transition, the expectation value of the POCC-tile energies to the convex hull is related to the enthalpy loss[20]. Similarly, the inverse of the standard deviation, $\sigma^{-1}$, gives a measure of the entropy gain: the narrower the spread, the larger the entropy content (EFA $\equiv \sigma^{-1}$, ref. 4). The balance between the entropy gain, $\sigma^{-1}$, and the enthalpy cost, $\langle \Delta H_{\text{hull}} \rangle$, leads to the definition of the DEED descriptor:

$$\text{DEED} \equiv \sqrt{\frac{\sigma_\Omega^{-1}[H_f]}{\langle \Delta H_{\text{hull}} \rangle_\Omega}}. \tag{3}$$

Figure 1a shows the workflow of calculating DEED for a disordered systems.

To estimate the trade-off between the entropy gain and the enthalpy cost, we also introduce the compensation temperature:

$$\Theta \equiv [k_B(\text{DEED})]^{-1}, \tag{4}$$

where $k_B$ is the Boltzmann constant[23].

Because high DEED indicates ease of synthesis, $\Theta$ is a fingerprint of the order–disorder transition, the critical temperature of the miscibility gap $T_c^{(\text{mg})}$. $\Theta$ is expected to be monotonically correlated with $T_c^{(\text{mg})}$ as it has the functional shape of a ratio between order–disorder enthalpy versus entropy estimated changes, and therefore it represents the temperature locus of the order–disorder Gibbs free energies cross-over. $\Theta$ is only an estimation of $T_c^{(\text{mg})}$ due to the approximations that are made to create DEED, for example, the finite configurational sampling of $\Omega$, lack of precursor effects or the finite-size effects of the calculations. However, $\Theta$ can still be used to compare and rank synthesizability of similar systems.

With growing number of species, the calculation of DEED might become impractical owing to the immense number of representative states to be computed. To overcome this challenge, we have

**Table 1 | DEED for classifying functional synthesizability of high-entropy ceramics**

| Composition | Group VIB | DEED | Θ | $\langle\Delta H_{hull}\rangle$ | $\sigma^{-1}$ | VEC | Experimental results | Reference |
|---|---|---|---|---|---|---|---|---|
| **Carbides** | | | | | | | | |
| (HfNbTaZr)C | | 65 | 180 | 22 | 92 | 8.50 | 🟢 | 34 |
| (HfNbTaTiZr)C | | 53 | 220 | 36 | 99 | 8.40 | 🟢 | 4,25–27 |
| (HfTaTiZr)C | | 44 | 263 | 43 | 83 | 8.25 | 🟢 | 34 |
| (HfScTaTiZr)C | | 37 | 311 | 39 | 55 | 8.00 | 🟢 | 16 |
| (HfNbTaTiV)C | | 37 | 316 | 77 | 103 | 8.60 | 🟢 | 4,35 |
| (HfMoNbTaTi)C | Mo | 32 | 362 | 81 | 83 | 8.80 | 🟢 | 36 |
| (MoNbTaTiV)C | Mo | 31 | 369 | 101 | 100 | 9.00 | 🟢 | 36 |
| (NbTaTiVZr)C | | 30 | 388 | 89 | 80 | 8.60 | 🟢 | 35 |
| (HfNbTaTiW)C | W | 28 | 411 | 82 | 66 | 8.80 | 🟢 | 4 |
| (HfNbTaVZr)C | | 27 | 436 | 93 | 65 | 8.60 | 🟢 | 35 |
| (HfTaTiVZr)C | | 26 | 439 | 104 | 72 | 8.40 | 🟢 | 35 |
| (HfNbTiVZr)C | | 26 | 440 | 102 | 71 | 8.40 | 🟢 | 35,37 |
| (MoNbTaVW)C | Mo, W | 26 | 443 | 171 | 117 | 9.40 | 🟢 | 4 |
| (NbTaTiVW)C | W | 26 | 446 | 111 | 75 | 9.00 | 🟢 | 4 |
| (HfMoTaTiZr)C | Mo | 25 | 458 | 95 | 61 | 8.60 | 🟢 | 36 |
| (NbTaTiWZr)C | W | 25 | 460 | 92 | 58 | 8.80 | 🟢 | 38 |
| (HfTaTiWZr)C | W | 23 | 495 | 90 | 49 | 8.60 | 🟢 | 4 |
| (CrMoNbTaW)C | Cr, Mo, W | 22 | 535 | 214 | 101 | 9.60 | 🟢 | 39 |
| (CrMoNbVW)C | Cr, Mo, W | 22 | 537 | 229 | 107 | 9.60 | 🟢 | 39 |
| (CrMoTaVW)C | Cr, Mo, W | 21 | 560 | 223 | 96 | 9.60 | 🟢 | 39 |
| (CrMoTiVW)C | Cr, Mo, W | 19 | 605 | 208 | 76 | 9.40 | 🟢 | 39 |
| (HfMoTaWZr)C | Mo, W | 17 | 681 | 152 | 44 | 9.00 | ❌ | 4 |
| (HfMoTiWZr)C | Mo, W | 17 | 702 | 141 | 39 | 8.80 | ❌ | 4 |
| (CrHfMoTiW)C | Cr, Mo, W | 14 | 819 | 210 | 42 | 9.20 | ❌ | 39 |
| (HfMoVWZr)C | Mo, W | 14 | 846 | 195 | 37 | 9.00 | ❌ | 4 |
| (CrHfTaWZr)C | Cr, W | 14 | 851 | 197 | 37 | 9.00 | ❌ | 39 |
| (CrMoTiWZr)C | Cr, Mo, W | 13 | 901 | 225 | 37 | 9.20 | ❌ | 39 |
| **Carbonitrides** | | | | | | | | |
| (HfTiZr)CN | | 36 | 326 | 91 | 116 | 8.50 | 🟢 | 40 |
| (CrNbTaTiV)CN | Cr | 16(c) | 719 | 260 | 68 | 9.50 | 🟢 | 28 |
| (HfNbTiZr)CN | | 16(c) | 736 | 146 | 36 | 8.75 | 🟢 | 40 |
| (HfNbTaTiZr)CN | | 15(c) | 768 | 182 | 41 | 8.90 | 🟢 | 28,40–42 |
| (HfNbTiVZr)CN | | 15(c) | 774 | 199 | 45 | 8.90 | 🟢 | validation |
| (HfNbTaTiV)CN | | 15(c) | 793 | 206 | 44 | 9.10 | 🟢 | validation |
| (CrHfNbTiZr)CN | Cr | 14(c) | 822 | 233 | 46 | 9.10 | 🟢 | 28 |
| (NbTaTiVZr)CN | | 14(c) | 835 | 220 | 42 | 9.10 | 🟢 | validation |
| (CrHfNbTaTi)CN | Cr | 14(c) | 847 | 239 | 45 | 9.30 | ❌ | 28 |
| (HfTaTiVZr)CN | | 14(c) | 854 | 220 | 41 | 8.90 | 🟢 | validation |

Continued

| Composition | Group VIB | DEED | $\Theta$ | $<\Delta H_{hull}>$ | $\sigma^{-1}$ | VEC | Experimental results | Reference |
|---|---|---|---|---|---|---|---|---|
| (CrHfTaTiZr)CN | Cr | 13(c) | 892 | 251 | 42 | 9.10 | 🟢 | 28 |
| (MoNbTaTiZr)CN | Mo | 12(c) | 980 | 260 | 36 | 9.30 | 🟢 | validation |
| (CrMoTaVW)CN | Cr, Mo, W | 10(c) | 1,109 | 435 | 48 | 10.10 | ✖ | 28 |
| (HfTaTiWZr)CN | W | 10(c) | 1,157 | 279 | 28 | 9.10 | ✖ | validation |
| (MoNbTiWZr)CN | Mo, W | 10(c) | 1,181 | 315 | 30 | 9.50 | ✖ | validation |
| (HfTiVWZr)CN | W | 10(c) | 1,190 | 284 | 27 | 9.10 | ✖ | validation |
| (HfMoNbTaW)CN | Mo, W | 10(c) | 1,207 | 393 | 36 | 9.70 | ✖ | validation |
| **Borides** | | | | | | | | |
| (HfNbTaTi)B$_2$ | | 175 | 66 | 8 | 251 | 10.50 | 🟢 | 43 |
| (HfNbTaTiZr)B$_2$ | | 126 | 92 | 15 | 239 | 10.40 | 🟢 | 32,44–49 |
| (HfMoNbTaZr)B$_2$ | Mo | 89 | 131 | 38 | 302 | 10.80 | 🟢 | validation |
| (HfNbTaTiV)B$_2$ | | 89 | 131 | 25 | 200 | 10.60 | 🟢 | validation |
| (HfMoNbTaTi)B$_2$ | Mo | 75 | 155 | 40 | 224 | 10.80 | 🟢 | 46,49–51 |
| (MoNbTaTiZr)B$_2$ | Mo | 71 | 164 | 44 | 220 | 10.80 | 🟢 | 49 |
| (HfMoNbTiZr)B$_2$ | Mo | 67 | 173 | 46 | 209 | 10.60 | 🟧 | 46,49,51,52 |
| (HfMoTaTiZr)B$_2$ | Mo | 64 | 180 | 48 | 200 | 10.60 | 🟢 | 32,45,49,53 |
| (HfNbTaWZr)B$_2$ | W | 62 | 188 | 60 | 228 | 10.80 | 🟧 | 54 |
| (HfTaTiVZr)B$_2$ | | 55 | 210 | 42 | 129 | 10.40 | 🟢 | 32 |
| (HfTaTiWZr)B$_2$ | W | 47 | 249 | 66 | 145 | 10.60 | 🟢 | 32 |
| (CrMoTiVW)B$_2$ | Cr, Mo, W | 40 | 293 | 93 | 147 | 11.40 | 🟢 | validation |
| (CrMoTaTiW)B$_2$ | Cr, Mo, W | 37 | 314 | 103 | 140 | 11.40 | 🟧 | 54 |
| (CrHfTaTiZr)B$_2$ | Cr | 36 | 322 | 63 | 81 | 10.60 | 🟢 | 32,45,49,51 |
| (HfMoTiWZr)B$_2$ | Mo, W | 35 | 328 | 89 | 111 | 10.80 | 🟧 | 32,45,49,54 |
| (CrHfNbTiZr)B$_2$ | Cr | 35 | 333 | 63 | 77 | 10.60 | 🟢 | validation |
| (CrHfMoTaW)B$_2$ | Cr, Mo, W | 34 | 346 | 113 | 127 | 11.40 | ✖ | validation |
| (HfMnTiVZr)B$_2$ | | 17 | 674 | 109 | 32 | 10.80 | ✖ | validation |
| (CrHfTiYZr)B$_2$ | Cr | 17 | 681 | 148 | 43 | 10.20 | ✖ | validation |
| (CrHfNbVY)B$_2$ | Cr | 17 | 702 | 139 | 38 | 10.60 | ✖ | validation |

DEED classifies the carbides and carbonitrides having a rock salt structure (AFLOW prototype AB_cF8_225_a_b), and the borides having an AlB$_2$ structure (AB2_hP3_191_a_d) (Methods). Components belonging to Group VIB, which do not form stable room-temperature rock salt monocarbides, are listed. DEED and $\sigma^{-1}$ (EFA) are in (eV per atom)$^{-1}$, $\Theta$ in Kelvin, $<\Delta H_{hull}>$ in (meV per atom) and VEC in (e$^-$ per cell). Experimental results showing single- and multiphase forming systems are designated with a circle 🟢 and a cross ✖, respectively, and the mixed reports are designated with a square 🟧. The validation label refers to results obtained in this work. The (c) in the DEED column indicates that cPOCC was used to parameterize the compound. Stoichiometry: carbides $M^1M^2M^3M^4M^5C_5$; carbonitrides $M_2^1M_2^2M_2^3M_2^4M_2^5C_5N_5$ and borides $M^1M^2M^3M^4M^5B^{10}$.

developed an approximated DEED that can be calculated out of a convolution of fewer, off-stoichiometry, POCC tiles. The approach, convolutional POCC (cPOCC), is illustrated in Fig. 1b and described in the Methods section. Global stoichiometry was chosen to be equiatomic for the transition metals, as the miscibility gap transition temperature tends to be higher around the central part of the phase diagram. This guarantees that the characterization of functional synthesizability by DEED is as restrictive as possible. Crystal structures are the ones forming in these classes of high-entropy ceramics at that particular composition: rock salt for carbides and/or carbonitrides and AlB$_2$ for borides (Methods). DEED is a thermodynamic descriptor and is not limited to stoichiometric compounds. The user can choose compositions, crystal structures and chemistries of interest.

## DEED for carbides
DEED improves the ordering of the carbide systems such that components belonging to group VIB (which do not form stable room-temperature rock salt structures) generally appear at the bottom of their respective classifications, reflecting the higher enthalpy costs associated with these structural transformations (column 'Group VIB' in Table 1). Overall, EFA also agrees with the classification, whereas valence electron concentration (VEC)[24] misclassifies experimentally synthesized single-phase carbides as multiphase, represented as the misclassification region in Fig. 2a.

In addition, $\Theta$ follows the expected trend for the transition temperature: having the highest values for multiphase systems. Values for $\Theta$ are reported in Table 1. For example, at 1,670 K, (HfNbTaTiZr)C

**a** DEED disordered enthalpy–entropy descriptor

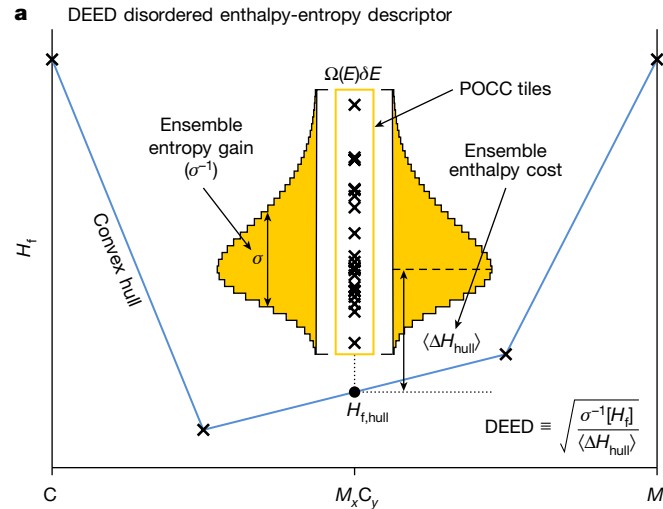

**b** cPOCC to accelerate spectra calculations

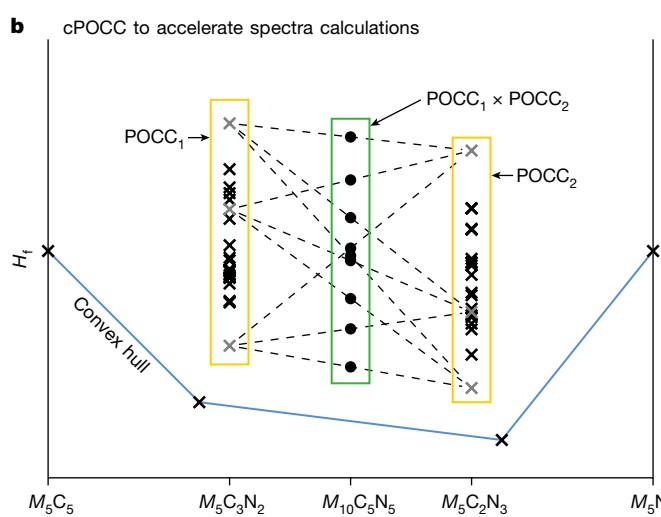

**Fig. 1 | DEED for a disordered system. a**, DEED workflow. $\langle\Delta H_{hull}\rangle$ and $\sigma^{-1}$ are calculated as the average distance of the POCC-tile energies to the convex hull and the spread (variance) of the POCC-tile energies, respectively. **b**, cPOCC workflow. The POCC expansion of a $M_{10}C_5N_5$ rock salt composition would require 17.5 million 20-atom tiles. This is overcome by a partition into $M_5C_2N_3$ and $M_5C_3N_2$ subsystems, each giving 490 unique ten-atom tiles and the subsequent energy convolutions. This cPOCC algorithm and its benchmarks are described in the Methods section.

(refs. 4,25–27) and (HfScTaTiZr)C (ref. 16) remain in the rock salt high-entropy phase whereas (HfNbTaTiV)C and (MoNbTaVW)C (ref. 4) are multiphase, suggesting the transition temperatures of (HfNbTaTiZr)C and (HfScTaTiZr)C to be lower than that of (HfNbTaTiV)C and (MoNbTaVW)C. From Table 1 and the Supplementary Information, we get $\Theta[(HfNbTaTiZr)C] < \Theta[(HfScTaTiZr)C] < \Theta[(HfNbTaTiV)C] < \Theta[(MoNbTaVW)C]$, which correlates with the order–disorder transition temperature.

### DEED for carbonitrides

DEED successfully classifies carbonitrides' synthesizability with low uncertainty, represented by a minuscule misclassification region shown in Fig. 2b. However, when using EFA, there is a noticeable misclassification region that gets even larger when using VEC.

To verify the integrity of DEED and to expand the limited high-entropy carbonitrides experimental data, we have synthesized nine new compounds. The precursors for all the carbonitrides were: TiC, VC, NbC, TaC, WC, Mo$_2$C, TiN, ZrN, HfN, NbN and TaN. All these precursors are rock salt except for WC (hexagonal, hP2), Mo$_2$C (orthorhombic, oP12),

NbN (hexagonal, hP4) and TaN (hexagonal, hP8). DEED can be used for two purposes: to find new solid solutions (large values of DEED, regardless of the structure of precursors or decomposition products[22]) and to search for microstructure formation in multiphase systems (small values of DEED, with concomitant structural incompatibility among precursors and/or decomposition products). For the solid-solution test, we randomly selected five compounds, (HfNbTiVZr)CN, (HfNbTaTiV)CN, (NbTaTiVZr)CN, (HfTaTiVZr)CN and (MoNbTaTiZr)CN, having the highest DEED and predicted to be single-phase. For the multiphase test, we randomly selected four compounds, (HfTaTiWZr)CN, (MoNbTiWZr)CN, (HfTiVWZr)CN and (HfMoNbTaW)CN with small DEED values. In addition, the compounds in this second batch, do not share the same precursor structure (more precisely, the same local atomic environment[22]), and therefore the formation of microstructures is possible.

The precursors were dry mixed for 24 h at 200 rpm with 9/1 ball to powder ratio, using a 2 mm YSZ media and 125 ml HDPE bottles. An extra 5 at% C was included in each powder blend to promote oxide reduction at elevated temperatures, without affecting the phase stability of the high-entropy ceramics, as previously reported in literature[28–30] (Supplementary Discussion 1). Then the precursors were sieved through a 60-mesh and sintered at various soak conditions (for example, 2,200 °C for 30 min, 2,300 °C for 60 min) using a 25 ton FAST system. A detailed description of the process can be found in the Methods section.

The addition of even more carbon and increasing its chemical potential (with extra pressure, for example) can also be described by DEED. It requires the recalculation of the enthalpy-stability landscape, the formation enthalpies distances of the POCC tiles and the statistical momenta as functions of the reference carbon chemical potential. It can then be a priori predicted if the carbide synthesizability is preserved or if other phases, with smaller metal solubility, are promoted.

The X-ray diffraction (XRD) pattern, shown in Fig. 2c, reveals that (HfNbTiVZr)CN 🟢, (HfNbTaTiV)CN 🟢, (NbTaTiVZr)CN 🟢 and (HfTaTiVZr)CN 🟢 and (MoNbTaTiZr)CN 🟢 are rock salt and composed of single face centred cubic (FCC) phase primary matrix grains, with trace oxides, namely HfO$_2$ and ZrO$_2$, as isolated particles (Extended Data Fig. 1). These oxides are in thermodynamic equilibrium with the carbonitride phase[29,30] and, therefore, should not affect the composition or phase stability of the compound. As further validation, we have also synthesized a high-entropy carbonitride with carbothermal reduction and found no difference in the final morphology (Supplementary Discussion 1). Meanwhile, the multiphase compounds are a combination of FCC and WC hexagonal phases. All DEED' predictions – solid solution (large DEED) and multiphases (small DEED) – were confirmed by the nine synthesized carbonitrides; on the contrary, EFA and VEC predictions were suboptimal, Fig. 2b. We conclude that DEED is a reliable descriptor for functional synthesizability of high-entropy transition-metal carbonitrides.

In the multiphase systems, the W-rich regions exist as irregular-shaped grains. In (MoNbTiWZr)CN ❌, (HfTiVWZr)CN ❌ and in particular in (HfMoNbTaW)CN ❌, the decomposition has promoted the growth of lamellar 'pearlite-like' microstructures[31]. The last three small panels in Fig. 2d show magnifications of (HfMoNbTaW)CN ❌. We analysed those pictures with the software AFLOW-Spinodal and determined the principal wavelength, $\lambda_p \cong 0.5\ \mu m$, of the lamellae (Extended Data Fig. 2). The result is not surprising. As mentioned before, small DEED values indicate very high $T_c^{(mg)}$, and synthesis would already occur inside the miscibility gap if $T_{synt} < T_c^{(mg)}$. Furthermore, if some of the precursors were not sharing the same parent lattice, then the miscibility gap would be bound by an invariant isotherm (for example, eutectoid). The lamellar microstructure is an indication of a eutectoid region, and it is mostly prominent in the composition (HfMoNbTaW)CN ❌ with the largest variety of precursor structures. The incomplete lamellae transformation in the three compounds with small DEED – (MoNbTiWZr)CN ❌, (HfTiVWZr)CN ❌, (HfMoNbTaW)CN ❌ – also

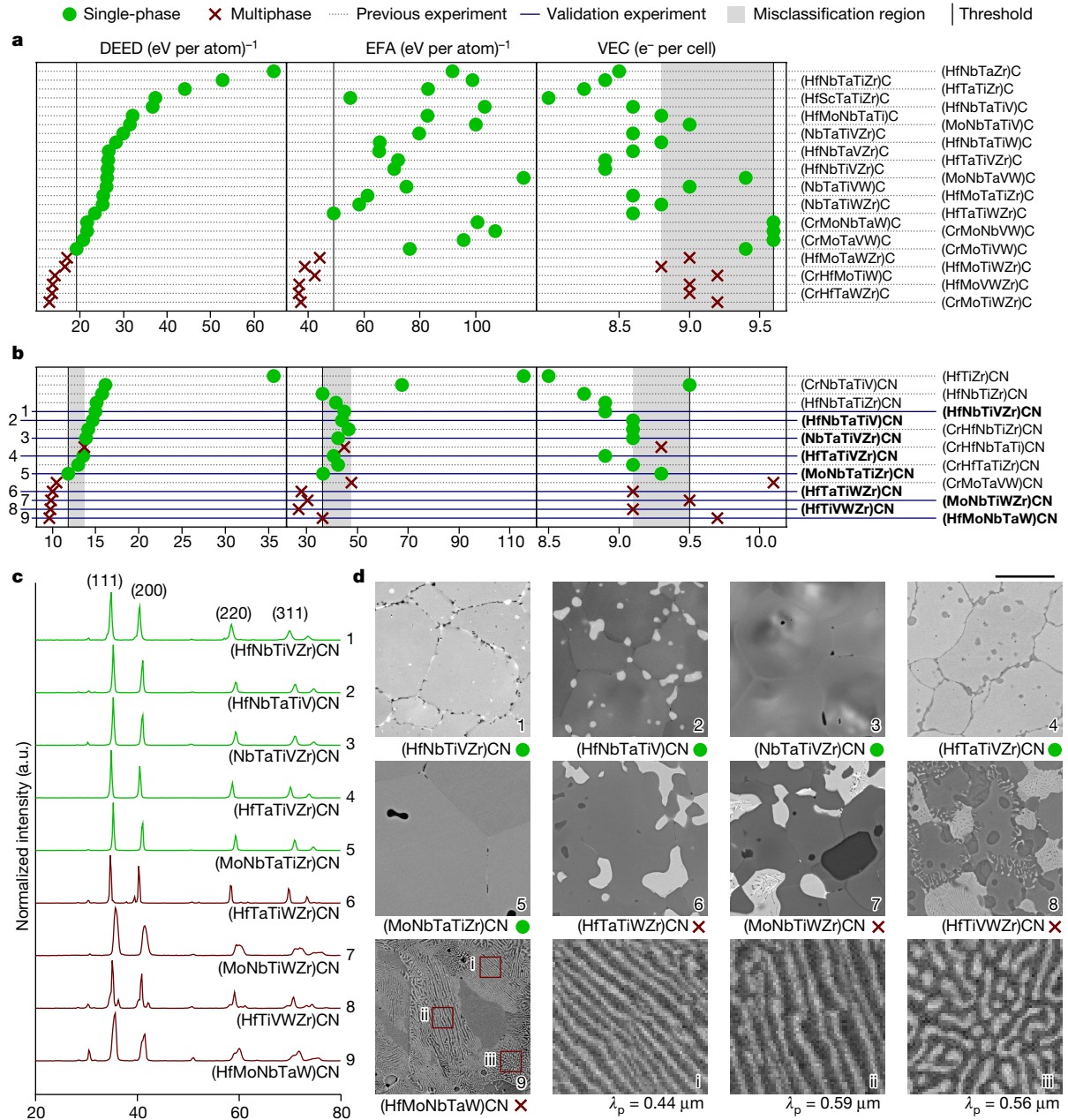

**Fig. 2 | DEED's functional synthesizability prediction performance for high-entropy carbides and carbonitrides.** DEED balances the entropy gains with enthalpy costs for forming a single-phase, predicting synthesizability. **a**,**b**, Here its superior performance is compared with EFA and VEC for carbides (**a**) and carbonitrides (**b**), ● and ✕ represent systems forming single- and multiphases, respectively. The references for the previous experimental results are listed in Table 1. Our validation experiments presented in this paper are set in bold. The misclassification regions, in grey, encompass all single-phase systems that are on the incorrect side of the single- and/or multiphase threshold (vertical line). **c**, The XRD spectra for carbonitrides. The five green spectra show single-phase FCC systems, whereas the four red lines reveal extra peaks related to the WC hexagonal phase. **d**, Microstructures from SEM analysis for the nine compounds and three magnified regions of (HfMoNbTaW)CN showing the lamellar 'pearlite-like' microstructures and their principle wavelength $\lambda_p$. a.u., arbitrary units. Scale bar, 10 μm.

suggests that we have not reached the exact eutectoid composition. In conclusion, the extra capability of DEED to help pinpoint regions of microstructure formation in multiphase systems, is extremely valuable as thermo-mechanical properties are greatly affected by microstructure morphology.

## DEED for borides

DEED always orders the boride systems that were experimentally observed as both single- and multiphase — represented as mixed reports in Fig. 3a — as less likely to form. These mixed reports show

that synthesizability depends on the experimental process[32]. Meanwhile, EFA does have a misclassification region and its ranking does not group the mixed reports close to the single- and/or multiphase threshold. Likewise, VEC shows an enormous misclassification region. Therefore, we conclude that VEC cannot resolve functional synthesizability of these high-entropy ceramics (Supplementary Discussion 2).

To better determine the DEED threshold (due to mixed reports in the literature), we synthesized eight new compounds. The four systems (HfMoNbTaZr)B₂, (HfNbTaTiV)B₂, (CrMoTiVW)B₂ and (CrHfNbTiZr)B₂ were selected by the highest DEED. As examples for multiphase systems,

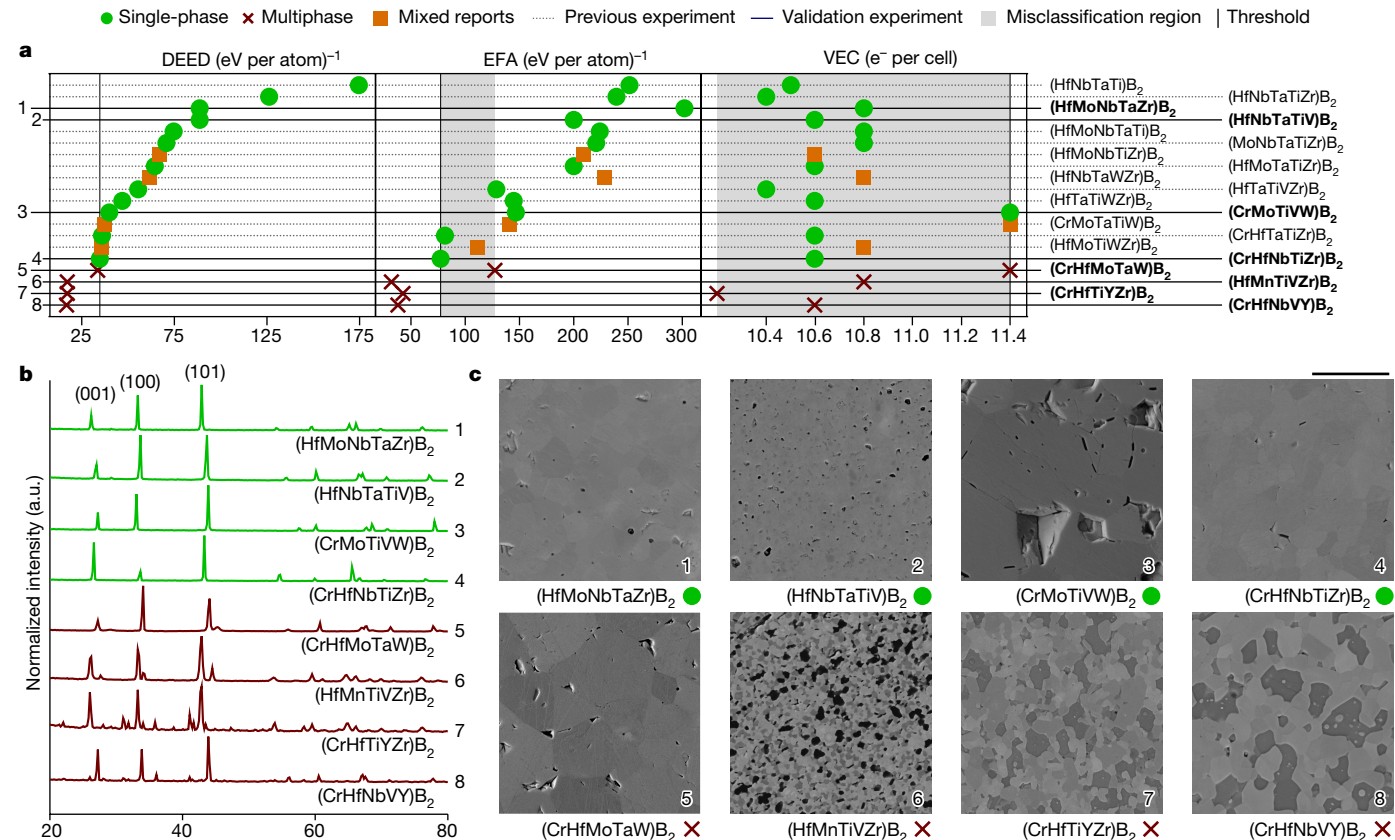

**Fig. 3 | DEED's functional synthesizability prediction performance for high-entropy borides.** DEED balances the entropy gains with enthalpy costs for forming a single-phase, predicting synthesizability. **a**, Here its superior performance is compared with EFA and VEC for borides. 🟢 and ✖ represent systems forming single- and multiphases, respectively. 🟧 show systems with mixed reports of forming both single- and multiphases. The references for the previous experimental results are listed in Table 1. Our validation experiments presented in this paper are set bold. The misclassification regions, in grey, encompass all single-phase systems that are on the incorrect side of the single- and/or multiphase threshold (vertical line). **b**, The XRD spectra for borides. The four green spectra show a AlB$_2$ hexagonal phase. By contrast, the red spectra show a combination of two distinct AlB$_2$ phases or a AlB$_2$ phase with a Y-based hexagonal phase. **c**, Microstructures from SEM analysis. Scale bar, 10 μm.

(CrHfMoTaW)B$_2$, (HfMnTiVZr)B$_2$, (CrHfTiYZr)B$_2$ and (CrHfNbVY)B$_2$ were randomly selected from the lower end of the calculated DEED range. The following precursors were chosen: MnO$_2$, WO$_3$, HfO$_2$, Nb$_2$O$_5$, Ta$_2$O$_5$, TiO$_2$, ZrO$_2$, V$_2$O$_5$, Cr$_2$O$_3$, MoO$_3$ and Y$_2$O$_3$.

Carbon black and B$_4$C were used to reduce these oxides. The precursors and reductants were mixed using ball milling and then pressed into discs at 2 MPa. Next, the boro- and/or carbothermal reduction was performed in a furnace under a mild vacuum at 1,660 °C for 2.5 h. Finally, the material was compacted in a two-step spark plasma sintering process: first, at 1,650 °C under 15 MPa pressure for 5 min to remove residual oxides and surface oxide impurities, and second, at 1,900 °C under 50 MPa pressure for 10 min (Supplementary Discussion 3).

The XRD of the boride ceramics is shown in Fig. 3b. All the materials predicted to be single-phase, (HfMoNbTaZr)B$_2$ 🟢, (HfNbTaTiV)B$_2$ 🟢, (CrMoTiVW)B$_2$ 🟢 and (CrHfNbTiZr)B$_2$ 🟢, show the clean peaks of a AlB$_2$ hexagonal phase. We find that (CrHfMoTaW)B$_2$ ✖ and (HfMnTiVZr)B$_2$ ✖ are composed of two different AlB$_2$ phases, whereas (CrHfTiYZr)B$_2$ ✖ is composed of AlB$_2$ and Y-based hexagonal phases, resulting in overlapping peak patterns. In (CrHfNbVY)B$_2$ ✖ the secondary peaks are less pronounced. Comparing the microstructure of the systems in Fig. 3b shows that (HfMoNbTaZr)B$_2$ 🟢 was almost fully dense and showed just a few residual carbon inclusions, (HfNbTaTiV)B$_2$ 🟢 contained residual carbon and notable porosity, (CrMoTiVW)B$_2$ 🟢, in addition to carbon, also contained residual B$_4$C and showed signs of micro-cracking, whereas (CrHfNbTiZr)B$_2$ 🟢 was mostly homogeneous. Furthermore, we found large grains of different phases in (CrHfMoTaW)B$_2$ ✖, whereas (HfMnTiVZr)B$_2$ ✖ contained a complex microstructure composed of several phases. In addition, we saw a reduction in grain size for (CrHfTiYZr)B$_2$ ✖ due to pinning based on the secondary phase and, in (CrHfNbVY)B$_2$ ✖, core shelling and micro-cracking were detected.

These experimental results are further evidence that DEED is a consistent descriptor to predict the functional synthesizability of high-entropy borides.

## Conclusion

We have introduced DEED, a thermodynamic descriptor that captures the balance between entropy gains and enthalpy costs on the formation of homogeneous solid solutions. We have also introduced the concept of functional synthesizability associated with the process' point of view. Our experimental results, combined with published data, confirm DEED as a reliable tool enabling computational discovery of novel high-entropy ceramics. We have also developed cPOCC, a convolutional algorithm to partition the immense number of configurations in complex solid solutions − such as carbonitrides − at a drastic reduction of computational requirements.

DEED accurately classifies functional synthesizability of disordered ceramics, enabling predictions across different material classes, regardless of chemistry and structure. Our method guided the experimental discovery of single-phase compositions (HfNbTiVZr)CN, (HfTaTiVZr)CN, (HfNbTaTiV)CN, (NbTaTiVZr)CN, (MoNbTaTiZr)CN, (HfMoNbTaZr)B$_2$, (HfNbTaTiV)B$_2$, (CrMoTiVW)B$_2$ and (CrHfNbTiZr)B$_2$.

This work also provides many new predictions, ripe for further experimental synthesis attempts. Extension to other types of disordered ceramics (for example, entropy-stabilized and high-entropy oxides) is straightforward, thanks to the coordination corrected enthalpy formalism[33], and the integration of DEED and cPOCC in the aflow++ computational ecosystem.

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

## Methods

### DEED as a thermodynamic descriptor

DEED can be easily extended to other classes of materials. It can be calculated as long as momenta of the thermodynamic density of states can be accessed. For instance, in non-crystalline and/or amorphous systems, DEED can be computed using the melt-quench method[55] or through the local atomic environment decomposition[56] to extract a thermodynamic density of states.

### Modelling chemically disordered materials

The AFLOW-POCC module models a random alloy (having sites with substitutional disorder) as an ensemble average of tiles[21]. Given a parent structure and occupancies of its sites, the algorithm first determines the smallest supercell needed to satisfy the specified occupancies and stoichiometry within input tolerances. POCC then constructs all configurations exhaustively, considering supercells of various shapes – the unique set corresponding to Hermite Normal Form matrices – and their decoration permutations. Duplicates are rapidly identified by grouping structures (by energy) as calculated by the Universal Force Field method, and then further reduced with the AFLOW-SYM[57] and the AFLOW-XtalFinder[58] modules. Only the unique set of tiles is further processed by AFLOW, which leverages the ab initio VASP package to calculate their properties. Calculation parameters have been set in accordance with the AFLOW Standard[59]. The properties of the disordered system are resolved as ensemble averages of those tiles. Details and equations are provided in ref. 21. If the alloy contains a sufficient concentration of impurities and/or vacancies, the problem can be treated explicitly by adding extra species and calculating larger POCC tiles. For small concentrations, perturbation theory can be used through variational approaches[60]. POCC's success has been extended to studies beyond configurational issues: for example, vibration in high-entropy carbides[22] and ultra-high-temperature plasmonic response[5].

### Parameterizing entropy descriptors

The energy distribution of the POCC tiles serves as a descriptor for the entropy of the system in which narrow distributions have greater accessibility to the various configurations[4]. For a thermodynamic density of states $\Omega$, comprising of a set of $n$ discrete POCC tiles with formation energies $H_{f,i}$ and factor-group degeneracies $g_i$, the momenta can be calculated as:

$$\langle \Delta H_{\text{hull}} \rangle \equiv \frac{\sum_{i=1}^{n} g_i (H_{f,i} - H_{f,\text{hull}})}{\sum_{i=1}^{n} g_i}, \tag{5}$$

$$\sigma\{H_{f,i}\} \equiv \sqrt{\frac{\sum_{i=1}^{n} g_i (H_{f,i} - \langle H_f \rangle)^2}{\left(\sum_{i=1}^{n} g_i\right) - 1}}, \tag{6}$$

$$\langle H_f \rangle \equiv \frac{\sum_{i=1}^{n} g_i H_{f,i}}{\sum_{i=1}^{n} g_i}, \tag{7}$$

where $H_{f,\text{hull}}$ is the formation energy of the convex hull.

### Convoluting variable concentration POCC tiles

When calculating properties of multicomponent disordered systems, one could face an impractical computational task owing to immensity (more representative states) and complexity (more atoms per unit cell). For example, a five-metal carbonitride system $\{5 \cdot M_{1/5}\}C_{1/2}N_{1/2}$ would require 17.5 million 20-atom POCC tiles. Here we propose a convolutional algorithm (cPOCC) that reduces the calculations to 490 ten-atom POCC tiles of $\{5 \cdot M_{1/5}\}C_{3/5}N_{2/5}$ and $\{5 \cdot M_{1/5}\}C_{2/5}N_{3/5}$, respectively. First, we assume that a system, with a thermodynamic density of states $\Omega(E)$,

can be arbitrarily divided into two subsystems (with complementary concentrations), having energies $\epsilon$ and $E - \epsilon$ and with thermodynamic density of states $\{\Omega(\epsilon) \cdot \Omega(E - \epsilon)\}$. Second, because of the finite number of sampled configurations, we approximate the $\Omega(E)$ by the distribution of configurationally excited POCC tiles. Third, using POCC, the thermodynamic density of states of the system could be approximated as the convolution of the two subsystems' thermodynamic density of states $(\Omega_1 * \Omega_2)(E) \equiv \int \Omega_1(\epsilon)\Omega_2(E - \epsilon)d\epsilon$, in which the convolution becomes the product in the thermodynamic limit. Finally, we can approximate the entropy as $k_B \log[(\Omega_1 * \Omega_2)(E)]$ with an associated entropy gain[4] equal to $\sigma^{-1}[\int \Omega_1(\epsilon)\Omega_2(E - \epsilon)d\epsilon]$. As long as the decomposition is phase-conserving – it does not exit the solid-solution region – the convolved $\sigma^{-1}$ is a valid approximation of the real $\sigma^{-1}$ as the intratile interactions are neglected in the POCC formalism. This carries a great advantage because we can choose the subsystems in which $\Omega$ is computationally accessible.

In the previous example, the disordered metal carbonitride $\{5 \cdot M_{1/5}\}C_{1/2}N_{1/2}$ (unobtainable thermodynamic density of states) is represented as two complementary subsystems with compositions $\{5 \cdot M_{1/5}\}C_{3/5}N_{2/5}$ ($\Omega_1$) and $\{5 \cdot M_{1/5}\}C_{2/5}N_{3/5}$ ($\Omega_2$). Both $\Omega_1$ and $\Omega_2$ are computationally accessible. For $\{5 \cdot M_{1/5}\}C_{3/5}N_{2/5}$ and $\{5 \cdot M_{1/5}\}C_{2/5}N_{3/5}$, the POCC algorithm enumerates 490 unique ten-atom configurations each. Figure 1b summarizes the idea. This scheme goes beyond carbonitrides or equimolar concentrations. It is a general strategy for approximating computationally challenging concentrations, given sufficient sampling statistics of the subsystems.

The cPOCC algorithm allows to calculate the thermodynamic density of states' statistical momenta for $\{5 \cdot M_{1/5}\}C_{1/2}N_{1/2}$ as a function of the ordered representatives enumerated at the two proxy stoichiometries. The combinations of $\{i\} - \{5 \cdot M_{1/5}\}C_{3/5}N_{2/5}$ and $\{j\} - \{5 \cdot M_{1/5}\}C_{2/5}N_{3/5}$ stoichiometries, generate the enthalpy of a pseudo $\{i,j\} - \{5 \cdot M_{1/5}\}C_{1/2}N_{1/2}$ structure, through straight-line interpolation ($H_{f,i,j}[H_{f,i}, H_{f,j}]$) (Fig. 1b). The degeneracy of the $\{i,j\}$ pseudo-configuration is $g_i \cdot g_j$. The set of $\{i,j\}$ enthalpies and degeneracies defines the energy distribution for the $\{5 \cdot M_{1/5}\}C_{1/2}N_{1/2}$ stoichiometry, from which the momenta can be calculated using equations (5)–(7).

To validate the method, $\sigma^{-1}$ has been obtained for three-metal carbonitrides, (HfNbZr)CN, (HfTaTi)CN and (TaTiZr)CN, where the number of calculations at the $\{3 \cdot M_{1/3}\}C_{1/2}N_{1/2}$ composition is accessible. The POCC algorithm enumerates 1,356 12-atom tiles for $\{3 \cdot M_{1/3}\}C_{1/2}N_{1/2}$. By comparison, cPOCC generates nine six-atom tiles for $\{3 \cdot M_{1/3}\}C_{1/3}N_{2/3}$ and $\{3 \cdot M_{1/3}\}C_{2/3}N_{1/3}$ each. We obtain (values in (eV per atom)$^{-1}$): (HfNbZr) CN $\{\sigma_{\text{POCC}}^{-1}, \sigma_{\text{cPOCC}}^{-1}, \text{error}\} = \{39.9, 41.7, +4.5\%\}$; (HfTaTi)CN $\{33.9, 35.3, +4.0\%\}$ and (TaTiZr)CN $\{32.4, 34.9, +7.6\%\}$. Overall, there is excellent agreement and the ranking is preserved.

### Calculating enthalpy costs

The enthalpy cost $\langle \Delta H_{\text{hull}} \rangle$ for forming a single-phase disordered system is calculated using equation (2), where $H_{f,\text{hull}}[\overline{x}]$ is obtained from an $n$-dimensional convex-hull analysis. Structures having $\{1, ..., 7\}$ components are extracted from the aflow.org repository, the enthalpy-composition space is populated, the convex hull is constructed and the enthalpy of the convex hull at the stoichiometry of interest $H_{f,\text{hull}}[\overline{x}]$ is calculated. The entire analysis is automated by the AFLOW-CHULL module[61] using the aflow.org REST and Search (AFLUX) APIs[62,63]. In total, 210,262 entries from the aflow.org repository have been used in the construction of the convex hull of carbonitride, carbide and boride compositions.

### Structure prototyping

Relevant ordered ceramic compounds are added to the aflow.org repository to supply the convex hull by decorating prototypes with cations from the high-entropy systems of interest. Carbides, borides and carbonitrides were chosen because of their useful applications[2,64,65], like thermal barrier protection[66–69], wear- and corrosion-resistant

coatings[70–74], thermoelectrics[75–77], batteries[78–80] and catalysts[81–85]. To find new prototypes, we extract carbides, borides and carbonitrides from the AFLOW-ICSD repository and compare them through AFLOW-XtalFinder[58]. In accordance with the AFLOW Prototype Encyclopedia[86–88], crystal structures are the ones forming in these classes of high-entropy ceramics at that particular composition — that is, rock salt for carbides and/or carbonitrides and AlB$_2$ for borides[1–4,4,25–27,35,39,41,47,89–91]. The prototype's label and internal degrees of freedom of each unique structure enables geometries to be automatically generated for subsequent simulations. New ceramics are created by decorating these prototypes with the relevant anions and the following metals: {Al, Cr, Hf, Ir, Mn, Mo, Nb, Sc, Ta, Ti, V, W, Y, Zr} for carbides; {Cr, Hf, Ir, Mn, Mo, Nb, Ta, Ti, V, W, Y, Zr} for borides and {Cr, Hf, Mo, Nb, Ta, Ti, V, W, Zr} for carbonitrides. Distinct decorations for a prototype are identified by comparing the possible permutations of the cation sites, whereas the anion sites remain fixed. A total of 988 binary and 8,892 ternary carbides; 846 binary and 7,962 ternary borides and 315 ternary and 1,872 quaternary carbonitrides geometries have been created. They are checked for uniqueness against the aflow.org repository, and only the distinct structures are calculated — using the the AFLOW standard[59] — and added to the database. These, together with the pre-existing data, generated a total of 210,262 structure-energy entries. AFLOW-CHULL then searched through all the possible 3,300 phase diagrams (including subcomponent phase diagrams) and extracted the 548 carbides, 70 carbonitrides and 334 borides, which were then parameterized both with DEED and EFA to pinpoint differences. A complete overview is presented in Extended Data Figs. 1–3. The detailed dataset is reported in the Supplementary Tables 1–3.

## High-entropy carbonitride sample preparation and characterization

The starting powders of TiC (99.7%, Inframat; lot no. IAM7261TIC1), VC (99.5%, Skyspring Nanomaterials; lot no. 8171-110620), NbC (99.5%, Inframat; lot no. IAM6071NbC), TaC (99.5%, H.C. Starck; lot no. TAC2138), Mo$_2$C (99.5%, Skyspring Nanomaterials; lot no. 5035-033022), WC (99.95%, Inframat; lot no. IAM3951W4), TiN (99.2%, US Research Nanomaterials, lot no. US1023M), ZrN (99.5%, HC Starck; lot no. 78307), HfN (99.5%, Heeger Materials; lot no. 20230110-5) and NbN (99%, NC Elements; lot no. NC-2015-0178), and TaN (99.5%, Stanford Advanced Materials; lot no. ST230215-17295) were batched accordingly into the respective high-entropy carbonitride compositions. About 25 g of each batch was dry mixed for 24 h at 200 rpm with a 9:1 ball to powder ratio using 2 mm of YSZ media and 125 ml HDPE bottles. These mixed powders were then sieved (60-mesh) and sintered at a soak temperature and time of 2,300 °C for 60 min using a 25 ton FAST system (FCT Systeme GmbH) with 20 mm outer-diameter graphite dies and/or punches, analogous to previous TiC-TiN carbonitride work[92]. The internal surfaces of the die body were covered with graphite foil to minimize die degradation and each cavity was loaded with about 10 g of powder. All sintering was performed under vacuum (roughly 0.4 mPa, pyrometer flush of 12 Pa N$_2$) over a constant heating rate of 100 °C min$^{-1}$ and a ramped pressure sequence from 30 to 50 MPa during this heating ramp. Each sintered sample was subsequently extracted, grit blasted with alumina (100 grit) to remove adherent graphite foil, and the Archimedes densities were measured with a precision digital analytical balance. These samples (Extended Data Fig. 3a) were then ground down to smooth planar surface (1,200 grit) to remove the outermost carbon-rich layer (roughly 0.5 mm) and subsequently characterized using XRD (CuKα/Ni filter, Empyrean III, Malvern Panalytical) for structural insights into the crystallinity of these synthesized high-entropy carbonitrides. These samples were then processed and characterized with scanning electron microscopy (SEM) (Tescan Mira3), XRD (Empyrean III, Malvern Panalytical) as shown in Fig. 2 and energy dispersive spectroscopy (Bruker Quantax) for microstructural and compositional insights as presented in Extended Data Fig. 4.

## High-entropy boride sample preparation and characterization

The starting materials included MnO$_2$ (99%, −10 mesh; Fisher Chemicals), WO$_3$ (99.9%, roughly 80 nm, Inframat Advanced Materials), HfO$_2$ (99%, −325 mesh; Alfa Aesar), Nb$_2$O$_5$ (99.5%, −100 mesh; Alfa Aesar), Ta$_2$O$_5$ (99.8%, 1–5 µm; Atlantic Equipment Engineers), TiO$_2$ (99.9%, 32 nm ASP; Alfa Aesar), ZrO$_2$ (99%, 5 µm; Sigma-Aldrich), V$_2$O$_5$ (99.6%, −10 mesh, Alfa Aesar), Cr$_2$O$_3$ (99.5%, 0.7 µm, Elementis), MoO$_3$ (99.9%, 6 µm, US Research Nanomaterials) and Y$_2$O$_3$ (99.95%, 0.6–0.9 µm; ABCR). Carbon black (BP1100, Cabot) and B$_4$C (96.8%, 0.6–1.2 µm; HC Starck) were used as reductants. The stoichiometric amounts of the oxides and reductants were calculated on the basis of the notional reaction: $(2/x) M_xO_y + (2y/x − 1) C + B_4C \rightarrow 2MB_2 + (2y/x)$ CO. Oxides plus the stoichiometric amount of carbon were subjected to dry high-energy ball milling (SPEX D8000, SPEX CertiPrep) using WC milling media and jar. Milling conditions were as follows: 1 h of milling, 30 min cooling period and another 1 h of milling. Milled powders were passed through a 100-mesh sieve. Next, the milled powder mixtures were mixed with 10 wt% excess B$_4$C by ball milling for 4 h using yttria-stabilized ZrO$_2$ media in acetone. Contamination from WC and ZrO$_2$ media was estimated by measuring the mass of the media before and after milling. The resulting slurries were dried at 65 °C using rotary evaporation (Model Rotavapor no. R-124: Buchi), and subsequently powders were passed through a 100-mesh sieve. The prepared powder mixtures were pressed into discs using a uniaxial pressure of 2 MPa. Boro- and/or carbothermal reduction was performed in a resistance-heated graphite element furnace (no. HP50-7010G, Thermal Technology) under mild vacuum (roughly 3 Pa), at 1,660 °C for 2.5 h. The reacted samples were crushed by mortar and pestle and sieved through a 100-mesh metallic sieve. Reacted powders were loaded in a graphite die that was lined with graphite foil. As-prepared samples were densified using two-step spark plasma sintering, as described in our previous studies[32,48]. The first step was heating to 1,650 °C under 15 MPa of applied pressure with heating rate 100 °C min$^{-1}$ and dwell time at the highest temperature for 5 min. This step was used to promote removal of residual oxides and surface oxide impurities. In the second step, pressure was increased to 50 MPa and temperature increased at 150 °C min$^{-1}$ to 1,900 °C where it was held for 10 min. The furnace was cooled at 10 °C min$^{-1}$ to 1,000 °C under a uniaxial pressure of 25 MPa and then the power was turned off to allow the furnace to cool at its natural rate to room temperature. Sintered specimens were ground before further examination to remove graphite foil and any reacted layers present on the surfaces (Extended Data Fig. 3c).

Phase compositions of the sintered specimens were investigated by XRD (PANalytical X-Pert Pro, Malvern Panalytical Ltd). XRD patterns were collected using Ni filtered Cu-Kα radiation by scanning the 2θ range from 5 to 90°. Rietveld refinement (RIQAS Software, MDI) was used to determine lattice parameters. Microstructural analysis (Fig. 3 and Extended Data Fig. 5) was done by SEM (Raith eLine Plus) with in situ chemical analysis by energy dispersive spectroscopy (Bruker Quantix). Specimens were polished to 0.25 µm using diamond suspensions, before SEM measurements.

## Data availability

The sample dataset to calculate DEED for a carbide system is available at https://doi.org/10.5281/zenodo.10015381. Data supporting this project are available from the corresponding author upon reasonable request.

## Code availability

The code of the algorithm to perform the DEED calculation is available at https://doi.org/10.5281/zenodo.10015381.

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

**Acknowledgements** We thank O. Levy, A. Garbesi, K. Vecchio, P. Sarker, M. Asta, A. Fortini, N. Goldman and S. Griesemer for fruitful discussions. Funding for this research was provided by the Office of Naval Research through a Multidisciplinary University Research Initiative programme under project nos. N00014-21-1-2515 and N00014-23-1-2615, and by the National Science Foundation (NSF grant no. NRT-HDR DGE-2022040). This work was supported by high-performance computer time and resources from the DoD High-Performance Computing Modernization Programme (Frontier).

**Author contributions** These authors contributed equally: S.D., H.E., D.H., C.O., and C.T. (listed alphabetically). S.C. conceptualized the momenta descriptor methods, the disorder expansion and the microstructure analysis, and designed and directed all aspects of the project. S.D., H.E. and S.C. developed the AFLOW-Spinodal module for microstructure search and analysis. C.O. and S.C. developed the AFLOW-POCC and AFLOW-CHULL modules. C.T., C.O. and S.C. derived the convolutional approach cPOCC. D.H. created the structural prototypes populating the convex-hull analyses. S.D., H.E., D.H., C.O., C.T., M.E. and R.F. prepared, performed and processed the DFT calculations needed for the descriptors. M.E. incorporated data into the aflow.org AFLUX-APIs. S.F. and W.G.F. synthesized and measured the high-entropy boride samples. C.J.R., C.M.D., R.J.C. and D.E.W. synthesized and measured the high-entropy carbonitride samples. All authors, S.D., H.E., D.H., C.O., C.T., R.F., M.E., M.J.M., A.C.Z., Y.L., E.Z., J.-P.M., D.W.B., X.C., S.F., W.G.F., C.J.R., C.M.D., R.J.C., D.E.W., A.C. and S.C. contributed to the discussion and to the writing of the article.

**Competing interests** The authors declare no competing interests.

**Additional information**
**Correspondence and requests for materials** should be addressed to Stefano Curtarolo.

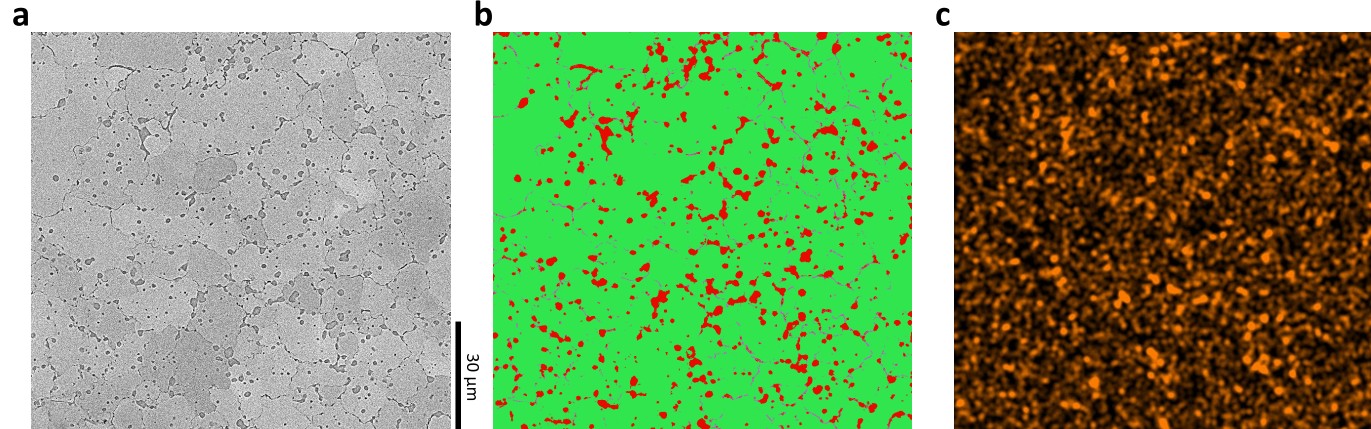

**a**

**b**

**c**

30 μm

**Extended Data Fig. 1 | Oxide inclusions. a**, SEM image of (HfNbTiVZr)CN, contrast optimized to show intra-grain inclusions. **b**, segmentation into inclusions (red), grain boundaries (purple) and grains (green) using Weka[93] **c**, oxygen SEM-EDS map. Based on the segmentation in **b** we can compare the average intensity in the inclusion region of the O EDSmap in **c** and in Extended Data Fig. 4. The elements O (1.94), Hf (1.31) and Zr (1.26) are enriched in the inclusions, while the intensities of Ta (0.70), Ti (0.66), and V (0.66) are reduced compared to the grains. Overall, the small amount of oxide in the studied ceramics would mean that the compositional differences in the high-entropy carbonitride phases would be very small (i.e., less than 1 mol%) with the small amount of oxide present in thermodynamic equilibrium with the carbonitride.

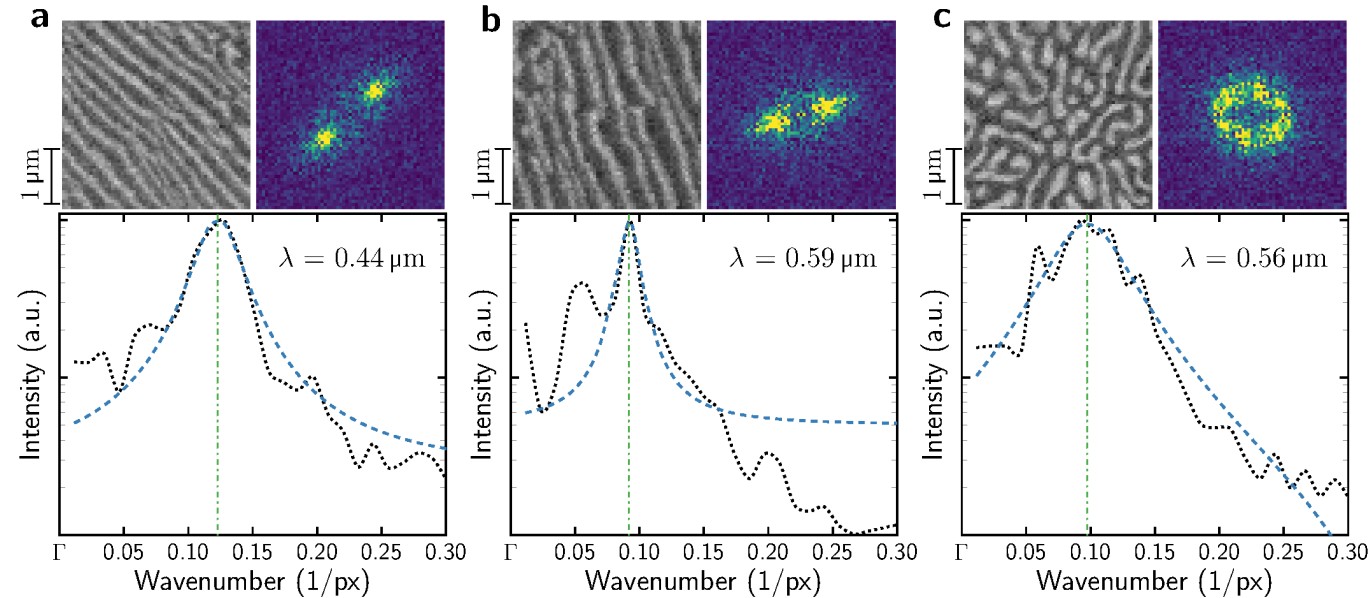

**Extended Data Fig. 2 | High-entropy ceramic microstructure.** Fourier analysis of the spinodal microstructure in the three magnified regions (**a, b, c**) of (HfMoNbTaW)CN in Fig. 2. Each sub-figure contains the image contrast (left) and the power spectrum of its Fourier transform (right). The black dotted line is the radial distribution function of the power spectrum which was smoothed using a Savitzky-Golay filter, to help identify the maxima in the signal. The blue dashed line is a fit of the Cauchy-Lorentz distribution to the radial distribution function. The green dotted-dashed line is the location of the distribution peak. a.u., arbitrary units.

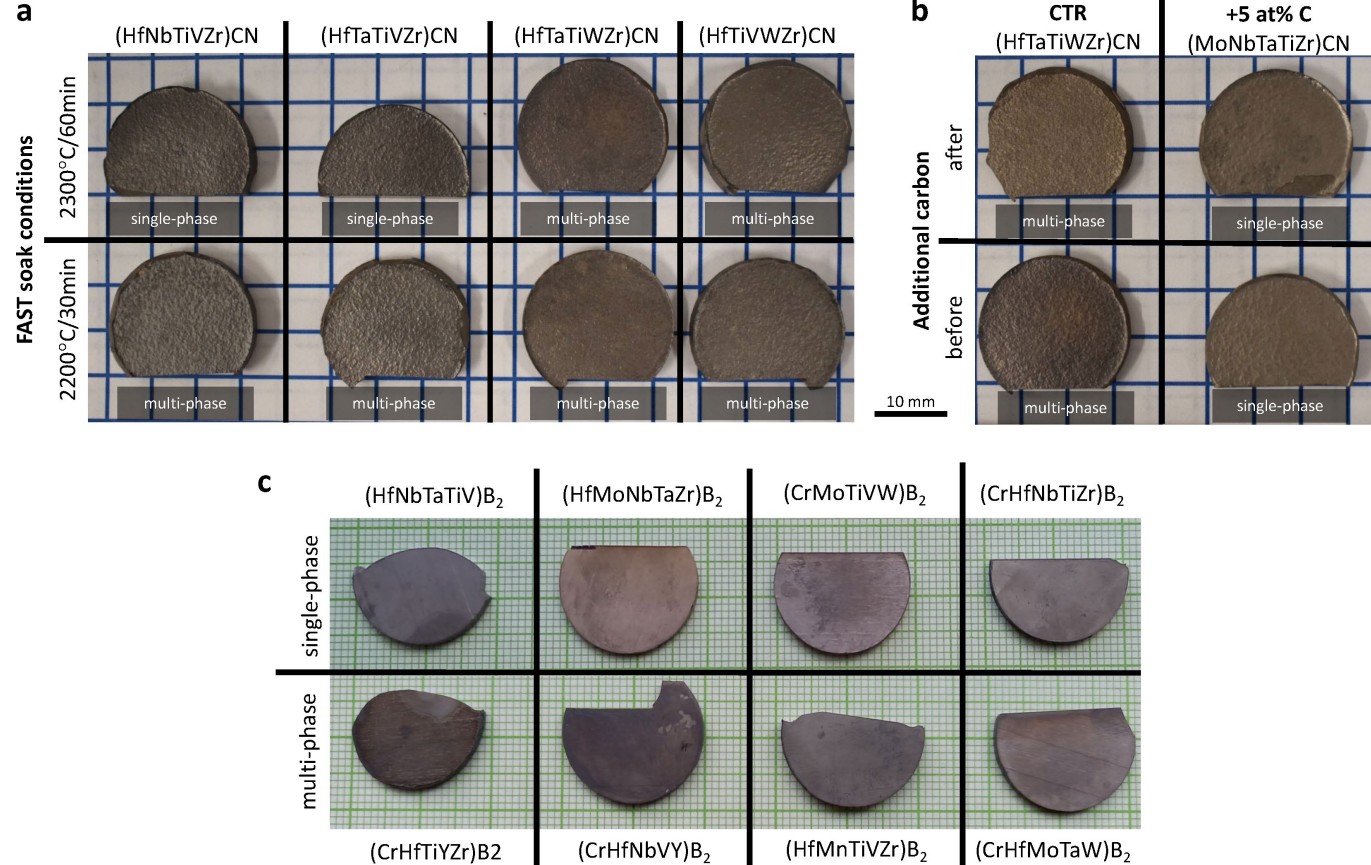

**Extended Data Fig. 3 | Sample disks. a|b**, show high-entropy carbonitride samples, Panel **a** presents the samples used to study different FAST soak conditions (2200 °C/30 min, 2300 °C/60 min) and samples in panel **b** were used to understand the effect of using additional carbon during the manufacturing of the samples. Either through carbothermal reduction treatment (CTR - 25 at% C, 1450 °C/3 h) or by direct addition of 5 at% trace carbon. All samples in **b** were sintered at 2300 °C/60 min. **c**, shows disks of high-entropy borides.

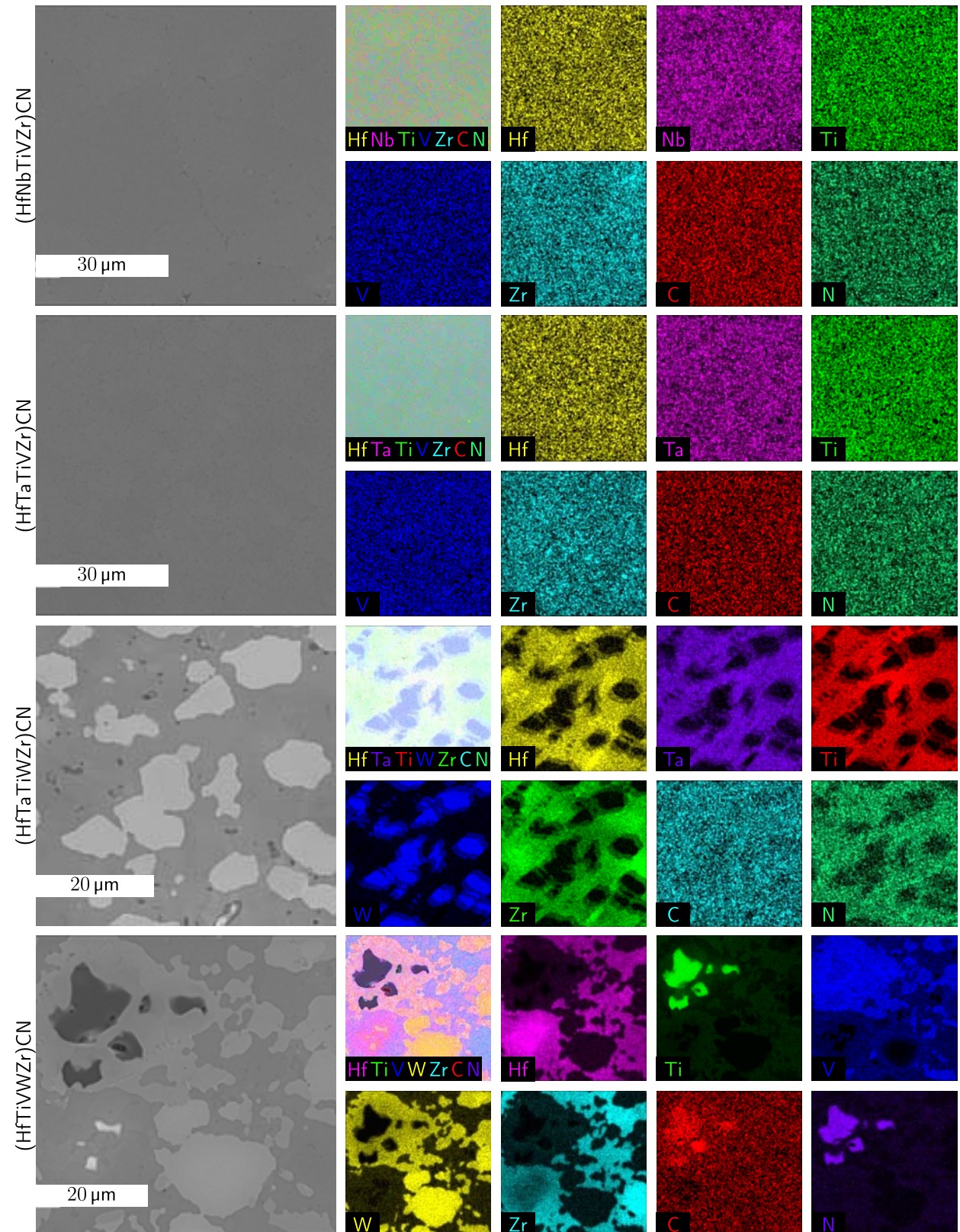

**Extended Data Fig. 4 | SEM-EDS maps of elements of high-entropy carbonitride microstructures.** From top to bottom: (HfNbTiVZr)CN, (HfTaTiVZr)CN, (HfTaTiWZr)CN and (HfTiVWZr)CN.

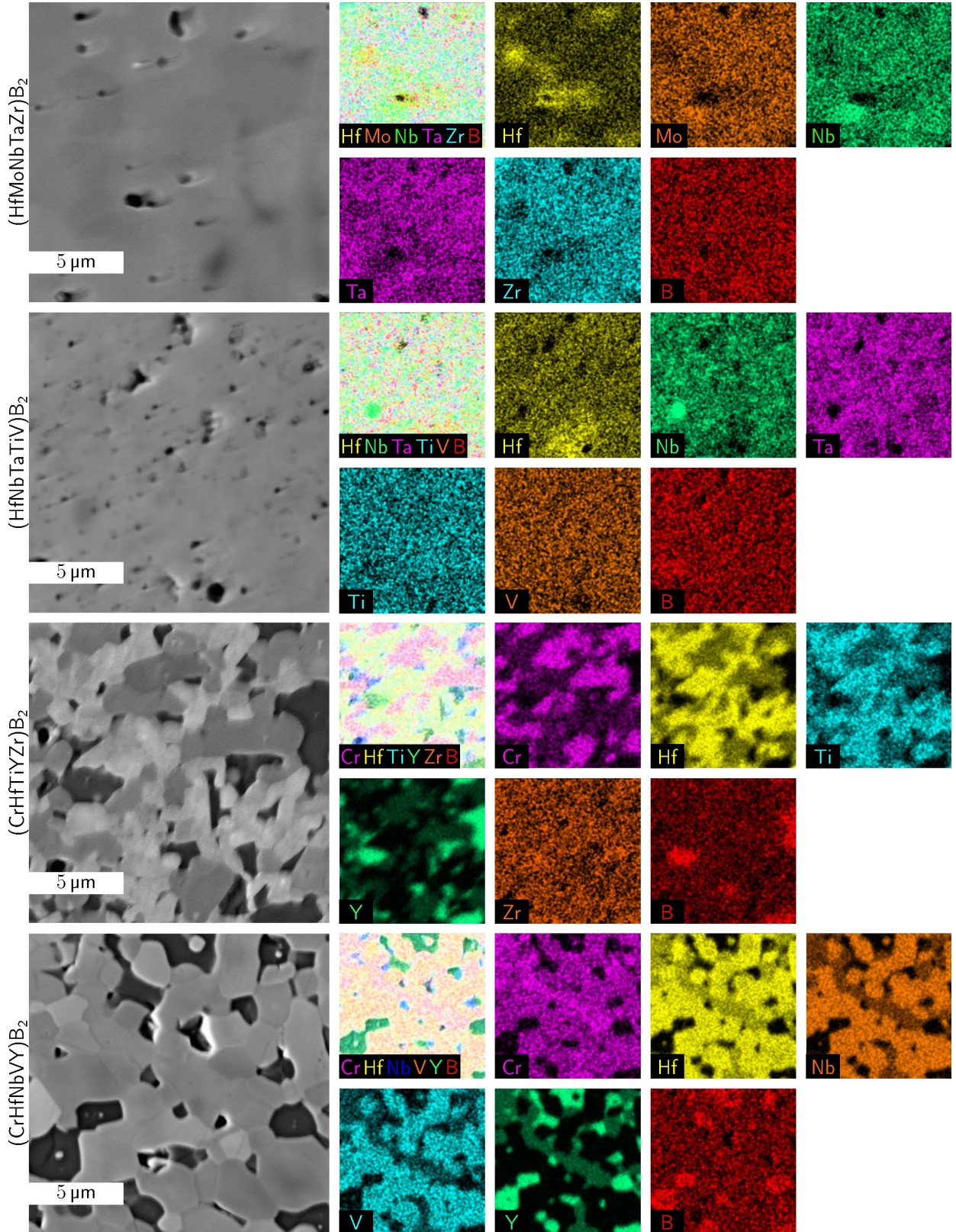

**Extended Data Fig. 5 | SEM-EDS maps of elements of high-entropy boride microstructures.** From top to bottom: (HfMoNbTaZr)B$_2$, (HfNbTaTiV)B$_2$, (CrHfTiYZr)B$_2$ and (CrHfNbVY)B$_2$.

## Extended Data Table 1 | Stability data for high-entropy carbides

| Composition | | | | | |
|---|---|---|---|---|---|
| (HfMoTaTiZr)C ● 53 | (MnMoTaTiW)C ✖ 17 | (CrMoTaVZr)C ✖ 14 | (HfNbSiVW)C ✖ 11 | (CrNbSiTaZr)C ✖ 9 | (HfIrMnNbTa)C ✖ 8 |
| (HfScTaTiZr)C ● 37 | (HfMoNbVW)C ✖ 17 | (IrMoNbTiW)C ✖ 14 | (HfIrNbTaW)C ✖ 11 | (IrMnMoNbTi)C ✖ 9 | (MnMoSiTaZr)C ✖ 8 |
| (HfNbTaTiV)C ● 37 | (HfMnNbTa)C ✖ 17 | (HfMnMoVW)C ✖ 14 | (HfIrMoTaTi)C ✖ 11 | (CrSiTaVZr)C ✖ 9 | (HfIrMnNbTi)C ✖ 8 |
| (HfMoNbTaTi)C ● 32 | (HfMoTaVW)C ✖ 17 | (CrMoNbVZr)C ✖ 13 | (AlSiTaTiW)C ✖ 11 | (IrMnNbTaTi)C ✖ 9 | (IrMnSiTaTi)C ✖ 7 |
| (MoNbTaTiV)C ● 31 | (CrHfNbTiZr)C ✖ 17 | (AlCrHfTaZr)C ✖ 13 | (HfIrNbTiW)C ✖ 11 | (CrMnMoSiTi)C ✖ 9 | (IrMnMoSiTa)C ✖ 7 |
| (NbTaTiVZr)C ● 30 | (AlHfMoTaTi)C ✖ 17 | (CrMoTaWZr)C ✖ 13 | (MoNbSiTaZr)C ✖ 11 | (AlCrMoSiTi)C ✖ 9 | (MnSiTiVZr)C ✖ 7 |
| (HfMoNbTaZr)C ● 28 | (HfTaVWZr)C ✖ 17 | (HfMnTaVW)C ✖ 13 | (HfIrMoNbTi)C ✖ 11 | (IrMoTiWZr)C ✖ 9 | (AlCrIrMnNb)C ✖ 7 |
| (HfNbTaTiW)C ● 28 | (MnMoTiVW)C ✖ 17 | (NbSiTiVW)C ✖ 13 | (HfIrMoTaV)C ✖ 11 | (CrMnMoSiW)C ✖ 9 | (AlCrMoSiZr)C ✖ 7 |
| (MoNbTaTiZr)C ● 28 | (HfMoTaWZr)C ✖ 17 | (CrMoNbWZr)C ✖ 13 | (CrMnSiTaV)C ✖ 11 | (AlMnSiVW)C ✖ 9 | (IrMnSiTaW)C ✖ 7 |
| (HfNbTaVZr)C ● 27 | (CrHfNbTaZr)C ✖ 17 | (MnMoTiWZr)C ✖ 13 | (HfNbSiTiV)C ✖ 11 | (AlIrMnMoV)C ✖ 9 | (MnNbSiTaZr)C ✖ 7 |
| (HfMoNbTiZr)C ● 27 | (MnMoTaTiV)C ✖ 17 | (CrMoTiVZr)C ✖ 13 | (HfIrMoNbTi)C ✖ 11 | (HfIrMnMoTa)C ✖ 9 | (IrMnMoNbV)C ✖ 7 |
| (HfTaTiVZr)C ● 26 | (HfMnNbTiW)C ✖ 17 | (AlCrMoTaW)C ✖ 13 | (CrMnSiTaV)C ✖ 11 | (IrTiVWZr)C ✖ 9 | (IrMoVWZr)C ✖ 7 |
| (HfNbTiVZr)C ● 26 | (AlMnNbTiV)C ✖ 17 | (MnNbTaTiZr)C ✖ 13 | (HfNbSiTiV)C ✖ 11 | (AlHfMnMoNi)C ✖ 9 | (IrMoSiTaV)C ✖ 7 |
| (MoNbTaVW)C ● 26 | (IrMoNbTaV)C ✖ 17 | (MnNbTaWZr)C ✖ 13 | (CrMnSiTaV)C ✖ 11 | (HfIrMnMoW)C ✖ 9 | (CrIrNbSiTi)C ✖ 7 |
| (NbTaTiVW)C ● 26 | (IrNbTaVW)C ✖ 17 | (CrHfMoVW)C ✖ 13 | (AlIrMoNbTa)C ✖ 10 | (IrMoTiVZr)C ✖ 9 | (IrMnMoVZr)C ✖ 7 |
| (MnMoNbTaV)C ● 26 | (MnMoNbVW)C ✖ 17 | (CrHfTaVZr)C ✖ 13 | (HfIrTiVZr)C ✖ 10 | (HfIrTiVZr)C ✖ 9 | (IrMnTaWZr)C ✖ 7 |
| (HfMoTaTiZr)C ● 25 | (AlCrHfNbTa)C ✖ 17 | (IrMnMoNbTa)C ✖ 13 | (AlSiVWZr)C ✖ 10 | (HfIrTiVZr)C ✖ 9 | (HfIrMnMoSi)C ✖ 7 |
| (HfNbTaVZr)C ● 25 | (HfMoTiVW)C ✖ 17 | (HfMnMoNbV)C ✖ 13 | (MnSiTiVW)C ✖ 10 | (HfIrTaVZr)C ✖ 9 | (IrMnTiVZr)C ✖ 7 |
| (CrMoNbTaTi)C ● 25 | (HfMoTiWZr)C ✖ 17 | (CrMoTiWZr)C ✖ 13 | (NbSiTaWZr)C ✖ 10 | (HfIrTaWZr)C ✖ 9 | (AlIrNbVZr)C ✖ 7 |
| (NbTaTiWZr)C ● 25 | (CrHfNbTaW)C ✖ 16 | (AlMnNbTaTi)C ✖ 13 | (HfMnTaVZr)C ✖ 10 | (CrSiTaVZr)C ✖ 9 | (AlIrMoNbZr)C ✖ 7 |
| (HfNbTaVZr)C ● 25 | (CrHfNbTaV)C ✖ 16 | (HfMnMoTaV)C ✖ 13 | (IrMoTaTiZr)C ✖ 10 | (HfIrMoNbTa)C ✖ 9 | (IrMnTiVW)C ✖ 7 |
| (CrNbTaTiV)C ● 25 | (CrNbTiWZr)C ✖ 16 | (MnMoNbWZr)C ✖ 13 | (AlHfMoNbTa)C ✖ 10 | (AlCrMnTaZr)C ✖ 9 | (AlIrNbVZr)C ✖ 7 |
| (CrMoNbTaV)C ● 25 | (IrMoNbTaTi)C ✖ 16 | (IrMnMoVW)C ✖ 13 | (AlIrMoTaTi)C ✖ 9 | (AlIrMoTaTi)C ✖ 9 | (IrMnTaVZr)C ✖ 7 |
| (HfMoNbTaV)C ● 24 | (CrHfIrVW)C ✖ 16 | (CrMoTiWZr)C ✖ 13 | (AlIrMoTiV)C ✖ 9 | (AlIrMoTiV)C ✖ 9 | (IrMnMoNbZr)C ✖ 7 |
| (HfMoNbTiV)C ● 24 | (MnMoNbTiV)C ✖ 16 | (CrHfTaVZr)C ✖ 13 | (AlHfMnTiZr)C ✖ 10 | (CrMnSiVW)C ✖ 9 | (IrMoNbSiW)C ✖ 7 |
| (MoNbTaVW)C ● 24 | (HfMnMoNbTi)C ✖ 16 | (IrMnMoNbTa)C ✖ 13 | (AlHfMoSiTa)C ✖ 9 | (IrMoVWZr)C ✖ 9 | (IrMnTaTiZr)C ✖ 7 |
| (AlHfNbTaV)C ● 24 | (CrMoNbTaZr)C ✖ 16 | (AlMnMoNbV)C ✖ 13 | (HfIrTaVZr)C ✖ 9 | (HfIrTaVZr)C ✖ 9 | (IrMnNbTiZr)C ✖ 7 |
| (HfTaTiVZr)C ● 23 | (AlMnNbVW)C ✖ 16 | (IrNbTaTiV)C ✖ 13 | (HfIrTaTiZr)C ✖ 9 | (CrSiTaVZr)C ✖ 9 | (IrMnNbTaZr)C ✖ 7 |
| (HfMoTaTiV)C ● 23 | (CrTaTiWZr)C ✖ 16 | (HfMnNbVW)C ✖ 13 | (HfMnTiZr)C ✖ 9 | (IrMoNbSiW)C ✖ 9 | (IrMnNbVZr)C ✖ 7 |
| (CrNbTaVW)C ● 23 | (MoNbVWZr)C ✖ 16 | (IrMnMoVW)C ✖ 13 | (HfMnMoTiZr)C ✖ 9 | (AlCrMnTaZr)C ✖ 9 | (AlCrHfIrTa)C ✖ 7 |
| (MnMoNbTaTi)C ● 23 | (CrHfMoTaV)C ✖ 16 | (HfMnMoNbV)C ✖ 13 | (AlIrMoTaTi)C ✖ 9 | (AlIrMoTaTi)C ✖ 9 | (CrIrMnSiTi)C ✖ 7 |
| (MnNbTiVW)C ● 23 | (CrMoTaTiZr)C ✖ 16 | (IrMnMoNbV)C ✖ 12 | (AlMnTaVZr)C ✖ 10 | (AlIrMoTiV)C ✖ 9 | (CrIrSiTaW)C ✖ 7 |
| (CrMoTaTiV)C ● 23 | (AlNbTiWZr)C ✖ 16 | (MnMoVWZr)C ✖ 12 | (HfMnMoNbZr)C ✖ 10 | (CrMnSiVW)C ✖ 9 | (AlMnMoSiZr)C ✖ 7 |
| (CrNbTiVW)C ● 23 | (CrNbTaWZr)C ✖ 16 | (MoNbSiTiV)C ✖ 12 | (SiTaTiVZr)C ✖ 10 | (IrMoVWZr)C ✖ 9 | (CrIrMoSiV)C ✖ 7 |
| (HfNbTiVW)C ● 23 | (MnMoNbTaZr)C ✖ 15 | (HfMnTaTiV)C ✖ 12 | (SiTiVWZr)C ✖ 10 | (AlIrMoTiV)C ✖ 9 | (AlHfIrVW)C ✖ 7 |
| (MnNbTaVW)C ● 22 | (IrTiVWZr)C ✖ 15 | (MoNbSiTiV)C ✖ 12 | (AlHfSiTaW)C ✖ 10 | (AlIrMoTiV)C ✖ 9 | (IrMnMoTiZr)C ✖ 7 |
| (CrMoNbTiV)C ● 22 | (MnTaTiWZr)C ✖ 15 | (HfMnTaTiV)C ✖ 12 | (HfSiTaTiZr)C ✖ 10 | (CrMnSiTaTi)C ✖ 9 | (IrMoSiTaTi)C ✖ 7 |
| (CrNbTaTiW)C ● 22 | (MoTaVWZr)C ✖ 15 | (MnMoTaVZr)C ✖ 12 | (HfMnMoTaW)C ✖ 10 | (AlCrIrTaV)C ✖ 9 | (CrMnMoSiZr)C ✖ 7 |
| (AlHfTaTiZr)C ● 22 | (CrHfMoTiV)C ✖ 15 | (IrMnTiVW)C ✖ 12 | (HfMnMoTiZr)C ✖ 10 | (CrSiVWZr)C ✖ 9 | (HfIrMnTiZr)C ✖ 7 |
| (MnNbTaTiV)C ● 22 | (CrNbTiVZr)C ✖ 15 | (MoSiTaTiW)C ✖ 12 | (AlNbSiTaV)C ✖ 10 | (HfIrNbVZr)C ✖ 9 | (AlHfMnMoSi)C ✖ 7 |
| (CrHfNbTaTi)C ● 22 | (AlHfTiVZr)C ✖ 15 | (HfMnNbTiV)C ✖ 12 | (MnMoNbSiTi)C ✖ 10 | (AlCrIrfTiTi)C ✖ 9 | (AlHfIrVZr)C ✖ 7 |
| (MnTaTiVW)C ● 22 | (IrTaTiVW)C ✖ 15 | (CrNbSiTaTi)C ✖ 12 | (HfIrTaTiV)C ✖ 10 | (HfIrMoTiZr)C ✖ 8 | (AlIrSiTaTi)C ✖ 7 |
| (HfNbTaVW)C ● 22 | (AlCrHfMoV)C ✖ 15 | (AlCrHfMoV)C ✖ 13 | (AlMnMoNbZr)C ✖ 10 | (AlCrMnNbZr)C ✖ 8 | (AlCrIrSiTi)C ✖ 7 |
| (AlHfNbTaZr)C ● 22 | (IrMoTaTiV)C ✖ 15 | (MnMoTaTiZr)C ✖ 12 | (HfIrMoTiW)C ✖ 10 | (AlIrNbTiW)C ✖ 8 | (AlIrSiVW)C ✖ 7 |
| (CrMoNbTaW)C ● 22 | (CrHfTiWZr)C ✖ 15 | (IrMnMoTaV)C ✖ 12 | (HfSiTiWZr)C ✖ 10 | (AlCrIrMoNb)C ✖ 8 | (CrHfIrNbSi)C ✖ 7 |
| (HfMoNbTaW)C ● 22 | (CrHfNbVW)C ✖ 15 | (HfMnMoNbTi)C ✖ 12 | (CrHfSiTaV)C ✖ 10 | (AlIrMnMoW)C ✖ 8 | (AlMnSiVZr)C ✖ 7 |
| (CrMoNbVW)C ● 22 | (CrHfMoNbV)C ✖ 15 | (HfMnNbTiV)C ✖ 12 | (HfIrTiVZr)C ✖ 10 | (AlHfMoSiW)C ✖ 8 | (HfIrMnWZr)C ✖ 7 |
| (HfTaTiVW)C ● 21 | (MnMoNbTiZr)C ✖ 15 | (CrMoVWZr)C ✖ 12 | (IrMoTaTiZr)C ✖ 10 | (AlCrIrTaTi)C ✖ 8 | (AlCrIrSiV)C ✖ 7 |
| (MoNbTaVZr)C ● 21 | (MnNbTiWZr)C ✖ 15 | (CrHfVWZr)C ✖ 12 | (CrMnSiTiV)C ✖ 10 | (HfIrMnTaTi)C ✖ 8 | (AlIrMoSiTa)C ✖ 6 |
| (CrTaTiVW)C ● 21 | (AlHfMoTiZr)C ✖ 15 | (MnTaVWZr)C ✖ 12 | (HfIrNbTiZr)C ✖ 10 | (AlMnNbSiTa)C ✖ 8 | (AlCrIrSiTa)C ✖ 6 |
| (MoNbTiVW)C ● 21 | (CrHfMoTaW)C ✖ 15 | (AlCrHfTiZr)C ✖ 12 | (AlNbMoNbZr)C ✖ 10 | (AlCrMnSiW)C ✖ 8 | (AlIrSiTaW)C ✖ 6 |
| (CrMoTaVW)C ● 21 | (IrNbTaTiW)C ✖ 15 | (HfMnMoTaZr)C ✖ 12 | (IrNbTiWZr)C ✖ 10 | (MnMoNbSiZr)C ✖ 8 | (CrIrMnSiW)C ✖ 6 |
| (MoNbTiVZr)C ● 21 | (HfMnMoTiW)C ✖ 15 | (IrMnNbTaV)C ✖ 12 | (IrMnMoTaV)C ✖ 10 | (AlHfIrTaTi)C ✖ 8 | (AlIrMnSiV)C ✖ 6 |
| (MoTaTiVW)C ● 20 | (MoTiVWZr)C ✖ 15 | (IrMnMoVW)C ✖ 12 | (HfMnMoTiV)C ✖ 9 | (CrHfSiVWZr)C ✖ 8 | (HfIrMoNbSi)C ✖ 6 |
| (MoTaTiVZr)C ● 20 | (HfMnNbTaW)C ✖ 15 | (CrNbSiTiW)C ✖ 12 | (HfMnNbNbW)C ✖ 9 | (HfIrMoWZr)C ✖ 8 | (HfIrMnTaV)C ✖ 6 |
| (AlMoNbTaV)C ● 20 | (HfMnTiVW)C ✖ 15 | (CrNbSiTaW)C ✖ 12 | (IrMoTaVZr)C ✖ 9 | (AlHfMnTaZr)C ✖ 8 | (AlIrMoNbSi)C ✖ 6 |
| (HfTaTiYZr)C ● 20 | (CrHfMoNbW)C ✖ 15 | (CrNbSiTaW)C ✖ 12 | (HfMnMoTiZr)C ✖ 9 | (AlHfSiTaZr)C ✖ 8 | (IrNbSiTiZr)C ✖ 6 |
| (NbTiVWZr)C ● 20 | (IrMoNbTiV)C ✖ 15 | (MnNbVWZr)C ✖ 12 | (SiTaVWZr)C ✖ 10 | (IrMnMoTaV)C ✖ 8 | (HfIrMnMoZr)C ✖ 6 |
| (HfMoNbTiW)C ● 20 | (IrMoTiVW)C ✖ 15 | (AlHfMnTaW)C ✖ 12 | (HfIrMoTiV)C ✖ 10 | (HfMnMoV)C ✖ 8 | (HfIrMnMoSi)C ✖ 6 |
| (CrNbTaTiZr)C ● 20 | (CrNbTaVZr)C ✖ 15 | (IrMnMoVW)C ✖ 12 | (HfIrMoVW)C ✖ 10 | (HfIrMnVW)C ✖ 8 | (CrHfIrMoSi)C ✖ 6 |
| (MoNbTaWZr)C ● 20 | (HfMnNbTaTi)C ✖ 14 | (MnTaTiVW)C ✖ 11 | (IrTaTiWZr)C ✖ 10 | (HfMnMoSiTa)C ✖ 8 | (AlHfIrSiTi)C ✖ 6 |
| (MnNbTaTiW)C ● 20 | (AlMnMoTaTi)C ✖ 14 | (MnMoNbVZr)C ✖ 11 | (IrMnMoTaTi)C ✖ 10 | (HfIrVWZr)C ✖ 8 | (HfIrMnTaZr)C ✖ 6 |
| (CrMoNbTiW)C ● 20 | (IrNbTaTiV)C ✖ 14 | (CrNbSiTiV)C ✖ 11 | (AlHfSiTiW)C ✖ 10 | (HfMnSiVW)C ✖ 8 | (HfIrMnMoSi)C ✖ 6 |
| (HfMoTaTiW)C ● 19 | (AlNbVWZr)C ✖ 14 | (IrMnTaTiV)C ✖ 11 | (IrTaTiVZr)C ✖ 10 | (HfIrMnTiV)C ✖ 8 | (IrNbSiTaZr)C ✖ 6 |
| (CrMoTiVW)C ● 19 | (IrMoTaTiW)C ✖ 14 | (IrMnMoTaW)C ✖ 11 | (IrMoTaTiW)C ✖ 9 | (HfIrMnTaW)C ✖ 8 | (AlHfIrNbSi)C ✖ 6 |
| (TaTiVWZr)C ✖ 19 | (CrHfMoTiV)C ✖ 14 | (HfIrMoNbTa)C ✖ 11 | (HfMnTiVZr)C ✖ 9 | (AlCrIrMoTi)C ✖ 8 | (IrMoSiVZr)C ✖ 6 |
| (NbTaVWZr)C ✖ 19 | (HfMnMoTaW)C ✖ 14 | (MnNbTaVZr)C ✖ 11 | (AlMoSiTaTi)C ✖ 9 | (HfIrMoVZr)C ✖ 8 | (AlHfIrSiV)C ✖ 6 |
| (MnMoNbVW)C ✖ 19 | (CrHfMoTiW)C ✖ 14 | (CrHfMoVZr)C ✖ 11 | (HfIrNbTaZr)C ✖ 9 | (HfIrMnNbW)C ✖ 8 | (IrNbSiWZr)C ✖ 6 |
| (CrMoTaTiW)C ✖ 19 | (MoNbSiTaTi)C ✖ 14 | (IrMnMoTiW)C ✖ 11 | (IrMoTaVZr)C ✖ 9 | (AlMnMoSiV)C ✖ 8 | (IrSiVWZr)C ✖ 6 |
| (HfMoNbVZr)C ✖ 19 | (CrHfNbWZr)C ✖ 14 | (CrNbSiVW)C ✖ 11 | (IrMoNbWZr)C ✖ 9 | (HfIrMnVW)C ✖ 8 | (IrMoSiTiZr)C ✖ 6 |
| (HfScTaTiZr)C ✖ 19 | (AlCrHfMoTa)C ✖ 14 | (CrNbSiTaW)C ✖ 11 | (AlHfNbSiW)C ✖ 9 | (CrIrSiTiV)C ✖ 8 | (AlCrIrMnSi)C ✖ 6 |
| (CrHfNbTiW)C ✖ 19 | (CrHfTaVZr)C ✖ 14 | (CrMoSiTaTi)C ✖ 11 | (IrNbTiVZr)C ✖ 9 | (AlMnMoSiTi)C ✖ 8 | (IrSiTaWZr)C ✖ 6 |
| (CrHfTaTiV)C ✖ 18 | (AlMnMoNbTi)C ✖ 14 | (HfMnTaWZr)C ✖ 11 | (IrMoNbTiV)C ✖ 9 | (HfIrMnMoNb)C ✖ 8 | (AlHfIrMnMo)C ✖ 6 |
| (HfMoTaVZr)C ✖ 18 | (CrHfMoNbZr)C ✖ 14 | (HfIrNbTaTi)C ✖ 11 | (CrMnNbSiTa)C ✖ 9 | (HfIrMnTiW)C ✖ 8 | (CrIrMoSiZr)C ✖ 6 |
| (HfMoTiVZr)C ✖ 18 | (AlCrHfMoZr)C ✖ 14 | (HfMnTiWZr)C ✖ 11 | (CrMoNbSiW)C ✖ 9 | (AlMnMoNbSi)C ✖ 8 | (AlIrSiTaZr)C ✖ 6 |
| (MnMoTaVW)C ✖ 18 | (MnNbTiVZr)C ✖ 14 | (AlCrHfMoZr)C ✖ 11 | (AlIrMoTaW)C ✖ 9 | (IrNbTaVZr)C ✖ 9 | (HfIrSiVZr)C ✖ 6 |
| (HfTiVWZr)C ✖ 18 | (SiTaTiVW)C ✖ 14 | (MnNbVZr)C ✖ 11 | (IrMoTaVZr)C ✖ 9 | (IrNbVWZr)C ✖ 9 | (AlHfIrMnW)C ✖ 6 |
| (MoNbTiWZr)C ✖ 18 | (MnTiVWZr)C ✖ 14 | (HfMnNbWZr)C ✖ 11 | (AlCrNbSiV)C ✖ 9 | (CrMnNbSiTa)C ✖ 9 | (CrIrMnSiZr)C ✖ 5 |
| (AlHfMoNbTi)C ✖ 18 | (IrMnTaTiW)C ✖ 14 | (IrMnTaTiV)C ✖ 11 | (IrNbTaVZr)C ✖ 9 | (HfMnMoVZr)C ✖ 9 | (AlHfIrSiW)C ✖ 5 |
| (CrHfTaTiZr)C ✖ 18 | (IrMoNbVW)C ✖ 14 | (CrMoNbSiTa)C ✖ 11 | (IrMoNbTiV)C ✖ 9 | (HfMoSiTiZr)C ✖ 9 | (AlHfIrSiZr)C ✖ 5 |
| (CrHfMoNbTi)C ✖ 18 | (AlCrMnTaV)C ✖ 14 | (CrMoNbSiW)C ✖ 11 | (AlIrMoTaW)C ✖ 9 | (IrTaVWZr)C ✖ 9 | (AlHfIrMoSi)C ✖ 5 |
| (CrHfNbTaTi)C ✖ 18 | (CrHfTiVZr)C ✖ 14 | (AlIrMoTaW)C ✖ 11 | (IrNbVWZr)C ✖ 9 | (MoSiTiWZr)C ✖ 9 | (AlIrMnMoZr)C ✖ 5 |
| (HfMnMoTaTi)C ✖ 18 | (IrMoTaVW)C ✖ 14 | (HfMnMoTaZr)C ✖ 11 | (HfMnMoTaZr)C ✖ 9 | (CrHfMnSiZr)C ✖ 8 | (AlCrHfIrSi)C ✖ 5 |
| (MnMoNbTiW)C ✖ 18 | (AlMnMoTaV)C ✖ 14 | (HfMnMoTaZr)C ✖ 11 | (HfIrMnNbV)C ✖ 8 | (HfIrMnNbV)C ✖ 8 | (AlCrIrMnZr)C ✖ 5 |
| (MoTaTiWZr)C ✖ 17 | (CrHfTaWZr)C ✖ 14 | (HfSiTaVW)C ✖ 11 | (MnMoSiTiW)C ✖ 9 | (IrMnSiTiW)C ✖ 8 | (AlHHfIrMnZr)C ✖ 5 |

The carbide compositions (AFLOW prototype AB_cF8_225_a_b)[86–88] are sorted by DEED in units of (eV per atom)$^{-1}$. Experimental results showing single- and multi-phase forming systems are designated with circle and cross, respectively, while mixed reports are designated with a square. The detailed dataset is reported in Supplementary Table 1.

# Extended Data Table 2 | Stability data for high-entropy carbonitrides

| Composition | | | Composition | | | Composition | | | Composition | | | Composition | | | Composition | | |
|---|---|---|---|---|---|---|---|---|---|---|---|---|---|---|---|---|---|
| (CrNbTaTiV)CN | ● | 16 | (HfMoNbTiV)CN | ● | 12 | (HfMoNbTaZr)CN | ✖ | 11 | (CrHfMoNbV)CN | ✖ | 11 | (NbTaVWZr)CN | ✖ | 10 | (MoTaTiWZr)CN | ✖ | 9 |
| (HfNbTaTiZr)CN | ● | 15 | (HfMoNbTaTi)CN | ● | 12 | (HfNbTiVW)CN | ✖ | 11 | (NbTiVWZr)CN | ✖ | 11 | (TaTiVWZr)CN | ✖ | 10 | (HfMoNbWZr)CN | ✖ | 9 |
| (HfNbTiVZr)CN | ● | 15 | (HfMoNbTiZr)CN | ● | 12 | (MoTaTiVW)CN | ✖ | 11 | (NbTaTiWZr)CN | ✖ | 11 | (HfTiVWZr)CN | ✖ | 10 | (HfTaVWZr)CN | ✖ | 9 |
| (HfNbTaTiV)CN | ● | 15 | (NbTaTiVW)CN | ● | 12 | (MoNbTaVZr)CN | ✖ | 11 | (CrMoTaVW)CN | ✖ | 10 | (HfMoNbTaW)CN | ✖ | 10 | (HfMoTaVW)CN | ✖ | 9 |
| (CrHfNbTiZr)CN | ● | 14 | (MoNbTiVW)CN | ● | 12 | (MoNbTaVW)CN | ✖ | 11 | (HfTaTiVW)CN | ✖ | 10 | (HfNbVWZr)CN | ✖ | 9 | (HfMoTiWZr)CN | ✖ | 9 |
| (MoNbTaTiV)CN | ● | 14 | (MoNbTaTiZr)CN | ● | 12 | (HfNbTaTiW)CN | ✖ | 11 | (HfMoTaVZr)CN | ✖ | 10 | (HfMoTiVW)CN | ✖ | 9 | (MoTaVWZr)CN | ✖ | 9 |
| (NbTaTiVZr)CN | ● | 14 | (MoNbTiVZr)CN | ✖ | 12 | (MoTaTiVZr)CN | ✖ | 11 | (HfNbTaVW)CN | ✖ | 10 | (HfMoNbVW)CN | ✖ | 9 | (HfMoTaWZr)CN | ✖ | 9 |
| (CrHfNbTaTi)CN | ✖ | 14 | (HfMoTaTiZr)CN | ✖ | 11 | (HfMoTiVZr)CN | ✖ | 11 | (HfTaTiWZr)CN | ✖ | 10 | (MoNbTaWZr)CN | ✖ | 9 | (HfMoVWZr)CN | ✖ | 8 |
| (HfTaTiVZr)CN | ● | 14 | (HfMoNbTaV)CN | ✖ | 11 | (HfNbTiWZr)CN | ✖ | 11 | (HfNbTaWZr)CN | ✖ | 10 | (HfMoTaTiW)CN | ✖ | 9 | | | |
| (CrHfTaTiZr)CN | ● | 13 | (MoNbTaTiW)CN | ✖ | 11 | (CrHfMoNbTa)CN | ✖ | 11 | (HfMoNbTiW)CN | ✖ | 10 | (MoNbVWZr)CN | ✖ | 9 | | | |
| (HfNbTaVZr)CN | ● | 13 | (HfMoTaTiV)CN | ✖ | 11 | (HfMoNbVZr)CN | ✖ | 11 | (MoNbTiWZr)CN | ✖ | 10 | (MoTiVWZr)CN | ✖ | 9 | | | |

The carbonitride compositions (AFLOW prototype AB_cF8_225_a_b)[86–88] are sorted by DEED in units of (eV per atom)$^{-1}$. All calculations in these table utilized cPOCC. Experimental results showing single- and multi-phase forming systems are designated with circle and cross, respectively, while mixed reports are designated with square. The detailed dataset is reported in Supplementary Table 2.

# Extended Data Table 3 | Stability data for high-entropy borides

Legend: ● single-phase (circle), ✘ multi-phase (cross), ■ mixed report (square). Values are DEED in units of $(\text{eV per atom})^{-1}$.

| Column 1 | Column 2 | Column 3 | Column 4 | Column 5 | Column 6 |
|---|---|---|---|---|---|
| $(HfNbTaTiZr)B_2$ ● 126 | $(HfMoTaVW)B_2$ ● 41 | $(MoNbTaWY)B_2$ ✘ 29 | $(HfMnTaWZr)B_2$ ✘ 22 | $(HfIrNbTaZr)B_2$ ✘ 9 | $(IrMnTiVW)B_2$ ✘ 7 |
| $(HfMoNbTaZr)B_2$ ● 89 | $(TaTiVWZr)B_2$ ● 41 | $(HfMnMoNbTa)B_2$ ✘ 29 | $(HfMnNbWZr)B_2$ ✘ 21 | $(IrNbTiVW)B_2$ ✘ 9 | $(IrMnMoTiV)B_2$ ✘ 7 |
| $(HfNbTaTiV)B_2$ ● 89 | $(HfMoTaWZr)B_2$ ● 40 | $(MnNbTaVW)B_2$ ✘ 28 | $(MnNbVWZr)B_2$ ✘ 21 | $(IrMoTaTiW)B_2$ ✘ 9 | $(IrMnMoNbW)B_2$ ✘ 7 |
| $(MoNbTaTiV)B_2$ ● 77 | $(CrHfNbTiV)B_2$ ● 40 | $(HfMnMoNbTi)B_2$ ✘ 28 | $(CrHfTaYZr)B_2$ ✘ 21 | $(IrMoTaTiZr)B_2$ ✘ 9 | $(IrMnMoNbTi)B_2$ ✘ 7 |
| $(HfMoNbTaTi)B_2$ ● 75 | $(HfMoNbTiW)B_2$ ● 40 | $(CrHfMoTaY)B_2$ ✘ 28 | $(MnMoNbVZr)B_2$ ✘ 21 | $(IrMoNbTaV)B_2$ ✘ 9 | $(IrMnNbTiW)B_2$ ✘ 7 |
| $(MoNbTaTiZr)B_2$ ● 71 | $(CrMoTiVW)B_2$ ● 40 | $(MnMoNbTiW)B_2$ ✘ 28 | $(CrMoNbTaY)B_2$ ✘ 21 | $(IrNbTaTiV)B_2$ ✘ 9 | $(IrMnMoTaTi)B_2$ ✘ 7 |
| $(NbTaTiVZr)B_2$ ● 68 | $(MoNbTiWZr)B_2$ ● 40 | $(HfMnMoTaTi)B_2$ ✘ 28 | $(TiVWYZr)B_2$ ✘ 21 | $(HfIrTaTiW)B_2$ ✘ 9 | $(IrMnMoNbV)B_2$ ✘ 7 |
| $(HfNbTaVZr)B_2$ ● 68 | $(MoNbVWZr)B_2$ ● 39 | $(HfMnNbTiW)B_2$ ✘ 28 | $(MnMoVWZr)B_2$ ✘ 21 | $(HfIrMoTiV)B_2$ ✘ 9 | $(HfIrMnMoTi)B_2$ ✘ 7 |
| $(HfMoNbTiZr)B_2$ ■ 67 | $(CrHfNbTiW)B_2$ ● 39 | $(MnNbTaTiZr)B_2$ ✘ 27 | $(HfMnNbVZr)B_2$ ✘ 21 | $(IrNbTaTiZr)B_2$ ✘ 9 | $(IrMnMoTaV)B_2$ ✘ 6 |
| $(HfMoTaTiZr)B_2$ ● 64 | $(CrHfMoTiV)B_2$ ● 39 | $(MnMoNbTaW)B_2$ ✘ 27 | $(CrHfNbYZr)B_2$ ✘ 21 | $(IrTaTiVW)B_2$ ✘ 9 | $(HfIrMnMoTa)B_2$ ✘ 6 |
| $(HfMoNbTaV)B_2$ ● 62 | $(CrMoNbTiW)B_2$ ● 38 | $(MnMoTaTiW)B_2$ ✘ 27 | $(HfMnMoWZr)B_2$ ✘ 20 | $(IrMoTiVW)B_2$ ✘ 9 | $(HfIrMnTiW)B_2$ ✘ 6 |
| $(HfNbTaWZr)B_2$ ■ 62 | $(CrHfNbTaW)B_2$ ● 38 | $(CrHfVWZr)B_2$ ✘ 27 | $(HfMnTaTiZr)B_2$ ✘ 20 | $(IrNbTaWZr)B_2$ ✘ 9 | $(HfIrMnMoW)B_2$ ✘ 6 |
| $(HfNbTaYZr)B_2$ ● 58 | $(MoTaVWZr)B_2$ ● 38 | $(HfMnNbTaW)B_2$ ✘ 27 | $(CrHfMoTaY)B_2$ ✘ 20 | $(HfIrNbTiZr)B_2$ ✘ 8 | $(IrMnMoTaW)B_2$ ✘ 6 |
| $(HfMoNbTiV)B_2$ ● 58 | $(HfMoTaTiW)B_2$ ● 38 | $(CrHfTiVZr)B_2$ ✘ 27 | $(HfMnNbTiZr)B_2$ ✘ 20 | $(HfIrMoNbV)B_2$ ✘ 8 | $(HfIrMnMoNb)B_2$ ✘ 6 |
| $(HfMoTaTiV)B_2$ ● 56 | $(MoTaTiWZr)B_2$ ● 38 | $(MnMoNbVW)B_2$ ✘ 27 | $(CrHfMoNbY)B_2$ ✘ 20 | $(IrMoNbVW)B_2$ ✘ 8 | $(HfIrMnTaW)B_2$ ✘ 6 |
| $(HfTaTiVZr)B_2$ ● 55 | $(CrMoNbTiZr)B_2$ ● 37 | $(CrHfMoVZr)B_2$ ✘ 27 | $(CrHfMoYZr)B_2$ ✘ 19 | $(HfIrNbVW)B_2$ ✘ 8 | $(IrMnNbTaTi)B_2$ ✘ 6 |
| $(MoNbTaVZr)B_2$ ● 53 | $(CrMoTaTiW)B_2$ ■ 37 | $(HfMnTaTiW)B_2$ ✘ 27 | $(CrHfTaWY)B_2$ ✘ 19 | $(HfIrMoNbW)B_2$ ✘ 8 | $(HfIrMnMoV)B_2$ ✘ 6 |
| $(MoNbTaVW)B_2$ ● 53 | $(HfMoTiVW)B_2$ ● 36 | $(MnMoNbTaZr)B_2$ ✘ 26 | $(CrHfWYZr)B_2$ ✘ 19 | $(HfIrTiVW)B_2$ ✘ 8 | $(IrMnMoTiZr)B_2$ ✘ 6 |
| $(CrMoNbTaTi)B_2$ ● 53 | $(CrHfTaTiZr)B_2$ ● 36 | $(HfMnNbTaV)B_2$ ✘ 26 | $(CrHfNbWY)B_2$ ✘ 19 | $(IrNbTaVW)B_2$ ✘ 8 | $(HfIrMnNbW)B_2$ ✘ 6 |
| $(NbTaTiVW)B_2$ ● 53 | $(HfMoTiWZr)B_2$ ■ 35 | $(MnMoTaVW)B_2$ ✘ 26 | $(HfMnVWZr)B_2$ ✘ 19 | $(HfIrMoTaV)B_2$ ✘ 8 | $(IrMnNbVW)B_2$ ✘ 6 |
| $(HfNbTiVZr)B_2$ ● 52 | $(CrHfMoNbV)B_2$ ● 35 | $(HfMnNbTiV)B_2$ ✘ 26 | $(CrHfNbTiY)B_2$ ✘ 19 | $(IrNbTiWZr)B_2$ ✘ 8 | $(IrMnMoNbZr)B_2$ ✘ 6 |
| $(HfNbTaVW)B_2$ ● 52 | $(CrNbTiWZr)B_2$ ● 35 | $(HfMnTaTiV)B_2$ ✘ 26 | $(HfMnMoVZr)B_2$ ✘ 19 | $(HfIrTaTiZr)B_2$ ✘ 8 | $(HfIrMnVW)B_2$ ✘ 6 |
| $(HfNbTaTiW)B_2$ ● 51 | $(CrHfNbTiZr)B_2$ ● 35 | $(HfMnTiVW)B_2$ ✘ 25 | $(CrTaTiWY)B_2$ ✘ 19 | $(HfIrTaWZr)B_2$ ✘ 8 | $(IrMnMoVW)B_2$ ✘ 6 |
| $(HfNbTiWZr)B_2$ ● 51 | $(CrHfMoTaTi)B_2$ ✘ 35 | $(HfMnMoTaW)B_2$ ✘ 25 | $(CrHfMoTiY)B_2$ ✘ 19 | $(HfIrMoTaW)B_2$ ✘ 8 | $(IrMnNbTiV)B_2$ ✘ 6 |
| $(CrHfNbTaTi)B_2$ ● 51 | $(CrHfMoTiZr)B_2$ ✘ 35 | $(HfMoTaVY)B_2$ ✘ 25 | $(CrTaTiYZr)B_2$ ✘ 18 | $(IrMoTaVW)B_2$ ✘ 8 | $(IrMnNbTaV)B_2$ ✘ 6 |
| $(NbTaTiVWZr)B_2$ ● 50 | $(CrHfNbVW)B_2$ ✘ 34 | $(HfMnMoNbW)B_2$ ✘ 25 | $(CrTiWYZr)B_2$ ✘ 18 | $(HfIrNbTaV)B_2$ ✘ 8 | $(IrMnMoTaZr)B_2$ ✘ 6 |
| $(CrMoNbTaW)B_2$ ● 49 | $(MoTiVWZr)B_2$ ✘ 34 | $(MnMoNbTiV)B_2$ ✘ 25 | $(CrHfMoWY)B_2$ ✘ 18 | $(HfIrMoTiW)B_2$ ✘ 8 | $(IrMnTaTiV)B_2$ ✘ 6 |
| $(MoNbTiVZr)B_2$ ● 48 | $(CrHfMoNbW)B_2$ ✘ 34 | $(HfMoWYZr)B_2$ ✘ 25 | $(HfMnTiVZr)B_2$ ✘ 17 | $(IrMoTiVZr)B_2$ ✘ 8 | $(IrMnTaTiW)B_2$ ✘ 6 |
| $(MoTaTiVZr)B_2$ ● 48 | $(HfMoVWZr)B_2$ ✘ 34 | $(MnNbTaWZr)B_2$ ✘ 25 | $(CrHfTiYZr)B_2$ ✘ 17 | $(HfIrNbTiV)B_2$ ✘ 8 | $(IrMnTiWZr)B_2$ ✘ 6 |
| $(HfMoTaVZr)B_2$ ● 48 | $(CrHfMoTaW)B_2$ ✘ 34 | $(MnMoTaTiZr)B_2$ ✘ 25 | $(CrTaVWY)B_2$ ✘ 17 | $(IrMoTaVZr)B_2$ ✘ 8 | $(IrMnTaWZr)B_2$ ✘ 6 |
| $(HfNbTiVW)B_2$ ● 48 | $(CrHfMoNbZr)B_2$ ✘ 33 | $(MnNbTiWZr)B_2$ ✘ 25 | $(CrNbVWY)B_2$ ✘ 17 | $(IrMoNbVZr)B_2$ ✘ 8 | $(IrMnMoVZr)B_2$ ✘ 6 |
| $(MoNbTaWZr)B_2$ ● 47 | $(CrNbTiVZr)B_2$ ✘ 33 | $(MnTaTiWZr)B_2$ ✘ 25 | $(CrTaTiVY)B_2$ ✘ 17 | $(HfIrMoNbTa)B_2$ ✘ 8 | $(IrMnMoWZr)B_2$ ✘ 6 |
| $(HfTaTiWZr)B_2$ ● 47 | $(MnMoNbTaTi)B_2$ ✘ 32 | $(HfMnNbTaZr)B_2$ ✘ 24 | $(CrHfNbVY)B_2$ ✘ 17 | $(IrTaTiVZr)B_2$ ✘ 7 | $(HfIrMnNbTa)B_2$ ✘ 6 |
| $(CrMoNbTaW)B_2$ ● 46 | $(CrHfNbWZr)B_2$ ✘ 32 | $(HfMnMoTiV)B_2$ ✘ 24 | $(CrNbTiVY)B_2$ ✘ 16 | $(IrTaVWZr)B_2$ ✘ 7 | $(IrMnTaVW)B_2$ ✘ 6 |
| $(HfMoNbTaW)B_2$ ● 46 | $(CrHfTaWZr)B_2$ ✘ 32 | $(HfMnTaVW)B_2$ ✘ 24 | $(CrHfVWY)B_2$ ✘ 16 | $(HfIrNbVZr)B_2$ ✘ 7 | $(HfIrMnTaTi)B_2$ ✘ 6 |
| $(NbTaVWZr)B_2$ ● 46 | $(CrMoTaVZr)B_2$ ✘ 32 | $(HfMnMoTaV)B_2$ ✘ 24 | $(CrHfMoVY)B_2$ ✘ 16 | $(HfIrMoWZr)B_2$ ✘ 7 | $(HfIrMnTiV)B_2$ ✘ 6 |
| $(HfMoNbVZr)B_2$ ● 46 | $(MnMoNbTiV)B_2$ ✘ 31 | $(MnNbTaVZr)B_2$ ✘ 24 | $(CrVWYZr)B_2$ ✘ 15 | $(HfIrTaVZr)B_2$ ✘ 7 | $(IrMnVWZr)B_2$ ✘ 6 |
| $(CrNbTaVW)B_2$ ● 45 | $(CrHfMoTiW)B_2$ ✘ 31 | $(HfMnNbVW)B_2$ ✘ 24 | $(CrMoVYZr)B_2$ ✘ 15 | $(IrTaTiVZr)B_2$ ✘ 7 | $(HfIrMnNbTi)B_2$ ✘ 6 |
| $(MoNbTaTiW)B_2$ ● 44 | $(MoNbTaTiY)B_2$ ✘ 31 | $(HfMnMoNbV)B_2$ ✘ 23 | $(CrTiVYZr)B_2$ ✘ 14 | $(HfIrTiVZr)B_2$ ✘ 7 | $(IrMnNbTaZr)B_2$ ✘ 6 |
| $(HfTaTiVW)B_2$ ● 44 | $(NbTaTiVY)B_2$ ✘ 31 | $(HfMnMoTaZr)B_2$ ✘ 23 | $(IrMoNbTaTi)B_2$ ✘ 11 | $(IrMnMoNbTa)B_2$ ✘ 7 | $(HfIrMnMoZr)B_2$ ✘ 6 |
| $(MoNbTiVW)B_2$ ● 44 | $(MnNbTaTiV)B_2$ ✘ 31 | $(MnMoTaWZr)B_2$ ✘ 23 | $(HfIrMoNbTa)B_2$ ✘ 10 | $(IrMnMoTiW)B_2$ ✘ 7 | $(HfIrMnNbV)B_2$ ✘ 6 |
| $(CrHfMoTaTi)B_2$ ● 43 | $(MnNbTaTiW)B_2$ ✘ 31 | $(MnTaTiVZr)B_2$ ✘ 23 | $(HfIrMoNbTi)B_2$ ✘ 10 | $(IrMnNbTaW)B_2$ ✘ 7 | $(IrMnMoWZr)B_2$ ✘ 6 |
| $(CrHfNbTaZr)B_2$ ● 43 | $(CrHfTaVZr)B_2$ ✘ 31 | $(MnMoNbWZr)B_2$ ✘ 23 | $(HfIrMoTaTi)B_2$ ✘ 10 | | $(IrMnNbTiZr)B_2$ ✘ 6 |
| $(CrNbTaTiZr)B_2$ ● 43 | $(MnMoTiVW)B_2$ ✘ 31 | $(HfMnMoNbZr)B_2$ ✘ 23 | $(IrNbTaTiW)B_2$ ✘ 10 | | $(IrMnTaTiZr)B_2$ ✘ 6 |
| $(CrHfNbTaV)B_2$ ● 43 | $(HfMnNbTaTi)B_2$ ✘ 31 | $(MnNbTiVZr)B_2$ ✘ 23 | $(IrMoTaVW)B_2$ ✘ 10 | | $(HfIrMnWZr)B_2$ ✘ 6 |
| $(NbTaTiVZr)B_2$ ● 43 | $(MnNbTiVW)B_2$ ✘ 30 | $(MnTiVWZr)B_2$ ✘ 22 | $(HfIrMoTiZr)B_2$ ✘ 9 | | $(IrMnTiVZr)B_2$ ✘ 6 |
| $(HfMoTiVZr)B_2$ ● 43 | $(CrHfMoVW)B_2$ ✘ 30 | $(CrHfNbTaY)B_2$ ✘ 22 | $(HfIrMoNbZr)B_2$ ✘ 9 | | $(HfIrMnTaZr)B_2$ ✘ 5 |
| $(CrHfMoNbTi)B_2$ ● 43 | $(CrTaVWZr)B_2$ ✘ 30 | $(HfMnMoVW)B_2$ ✘ 22 | $(IrMoNbTiV)B_2$ ✘ 9 | | $(HfIrMnNbZr)B_2$ ✘ 5 |
| $(HfMoNbVW)B_2$ ● 42 | $(HfTiVYZr)B_2$ ✘ 30 | $(HfMnTiWZr)B_2$ ✘ 22 | $(HfIrNbTaTi)B_2$ ✘ 9 | | $(IrMnNbVZr)B_2$ ✘ 5 |
| $(HfNbVWZr)B_2$ ● 42 | $(MnMoNbTaV)B_2$ ✘ 30 | $(MnTaVWZr)B_2$ ✘ 22 | $(IrMoNbTiW)B_2$ ✘ 9 | | $(IrMnTaVZr)B_2$ ✘ 5 |
| $(CrHfMoNbTa)B_2$ ● 42 | $(MnTaTiVW)B_2$ ✘ 30 | $(HfMnTaVZr)B_2$ ✘ 22 | $(HfIrMoTaZr)B_2$ ✘ 9 | | $(HfIrMnTiZr)B_2$ ✘ 5 |
| $(CrHfTaTiV)B_2$ ● 42 | $(MoTaTiYZr)B_2$ ✘ 30 | $(HfMnMoTiZr)B_2$ ✘ 22 | $(IrMoNbTaZr)B_2$ ✘ 9 | | $(HfIrMnVZr)B_2$ ✘ 5 |
| $(HfTaVWZr)B_2$ ● 42 | $(CrHfNbVZr)B_2$ ✘ 29 | $(MnMoTaVZr)B_2$ ✘ 22 | $(HfIrNbTaW)B_2$ ✘ 9 | | |
| $(HfMoNbWZr)B_2$ ● 41 | | $(MnMoTiWZr)B_2$ ✘ 22 | $(HfIrNbTiW)B_2$ ✘ 9 | | |
| $(HfTiVWZr)B_2$ ● 41 | | $(MnMoTiVZr)B_2$ ✘ 22 | $(IrMoNbTiZr)B_2$ ✘ 9 | | |
| $(MoTaTiVW)B_2$ ● 41 | | | $(IrMoTaTiV)B_2$ ✘ 9 | | |

The boride compositions (AFLOW prototype AB2_hP3_191_a_d)[86–88] are sorted by DEED in units of (eV per atom)[−1]. Experimental results showing single- and multi-phase forming systems are designated with circle and cross, respectively, while mixed reports are designated with square. The detailed dataset is reported in Supplementary Table 3.