## [Peer Review File · Nature]

Manuscript Title: Disordered enthalpy-entropy descriptor for high-entropy ceramics discovery

Reviewer Comments & Author Rebuttals

Reviewer Reports on the Initial Version:

Referees' comments:

Referee #1 (Remarks to the Author):

The authors describe a predictive tool for carbonitride and boride high-entropy carbides, validated by experimental results of certain predicted compounds. The computational work is solid. The experimental work could use improvement.

The selection of only four carbonitrides, with two representing predicted single-phase systems and two representing mixed-phased compositions, lacks rigor. For validation, I would have expected a few more compositions, especially considering that the materials end up with oxide impurities, which likely affects the phase stability in the systems. Can the authors describe if the oxide impurities have an effect on the capacity to obtain single- or mixed-phase? Can this effect be incorporated into DEED?

Regarding the borides, there is a description of sintering that results in the microstructures shown in Figure 3. Both carbonitrides and borides were sintered using similar techniques (i.e., spark plasma sintering), but in different equipment. An HP5--7010G system was used for the borides and an FCT system was used for the carbonitrides. Why change equipment? Would this have an effect on oxide impurities considering changes in vacuum in the two systems used? And again, does this have an effect on the phase stability in the materials? Also, in some cases, extra carbon was added to reduce oxides. Does this then have an effect on the phase stability? Can this be correlated to the computational work in some way?

If better controlled specimens can be prepared (without oxide contamination), this manuscript would be outstanding. At the moment, however, there is a lack of rigor in the experiments, variability is not clear from sample to sample (for example, did the authors prepare more than just one sample of each composition?), and there seems to be changes in methodologies for sintering that are not fully justified.

Referee #2 (Remarks to the Author):

In this manuscript, the authors present a disordered enthalpy-entropy descriptor, called DEED, that provides a classification of the so-called "synthesizability" of multicomponent ceramics. They show the application of DEED to high-entropy carbides, carbonitrides and borides based on the published experimental data together with several new compounds synthesized by the authors. The results, as summarized in Table 1, show improved classification compared to other descriptors such as valence electron concentration (VEC) and entropy-forming-ability (EFA).

Overall, the descriptor DEED serves to provide a classification of multicomponent ceramics only in terms of the formation of single vs. multi-phases. It has a limited value and lacks a general applicability for guiding the development of new multicomponent ceramics. Hence, the work lacks general relevance and potential impact to warrant its publication in Nature.

A serious issue of the work is that the candidate materials considered are limited to the

stoichiometric compounds consisting of early transition metal elements with a priori knowledge of a specific lattice structure. It is not clear if the DEED can be used to address some of the more basic questions on the synthesis of multicomponent ceramics, such as whether a crystalline or an amorphous structure will form; what type of crystal structure(s) will form for single or multi-phase systems; what is the applicability of DEED for non-stoichiometric ceramics.

Another serious issue is that the term of "synthesizability" is used vaguely throughout the manuscript. The calculated DEED value is not directly connected to any experimentally measurable material property. For the ceramic systems studied in the manuscript, the threshold value of DEED for classification is dependent on each specific set of constituent elements chosen, and likely dependent sensitively on (stoichiometric/non-stoichiometric) composition and crystal structure, etc. It is not clear how the threshold value is determined and what is the amount of experimental input is needed. Moreover, as the authors state, "synthesizability depends on the experimental process". Considering all of those limitations, the descriptor of DEED does not have a general and strong value for guiding the synthesis of multicomponent ceramics.

I regret to summarize that I feel this work does not represent a clear and significant advancement for the development of multicomponent ceramics. It shows one improved descriptor for classification of formation of single vs. multi-phases within a limited range of material design space. The work does not represent a strong breakthrough which should be the main criterion for a journal as Nature.

Author Rebuttals to Initial Comments:

Reviewer #1 (Comments to the Authors)

Q1. *The authors describe a predictive tool for carbonitride and boride high-entropy carbides, validated by experimental results of certain predicted compounds. The computational work is solid. The experimental work could use improvement.*

A1. We thank Reviewer 1 for the positive praise about the computational work. We also appreciate the suggestions to improve the manuscript by extending its experimental component, as discussed in the text below and with modifications and additions to the article. We have also discussed additional capabilities of DEED in suggesting microstructure formation in systems with small DEED values. Note that the modifications to the article will be made in blue both in this response and in the resubmitted document.

Q1.1. *The selection of only four carbonitrides, with two representing predicted single-phase systems and two representing mixed-phased compositions, lacks rigor. For validation, I would have expected a few more compositions, especially considering that the materials end up with oxide impurities, which likely affects the phase stability in the systems.*

A1.1. To increase validation reliability, we extended the overall number of our test systems from 8 to 17. We have added 5 new transition-metal carbonitrides and 4 new transition-metal borides. In all new systems, DEED performs flawlessly in the prediction of single-phase solid-solutions and multi-phase formation.

Carbonitrides: Five additional, previously unexplored, compositions were synthesized: three are found to be single-phase solid-solutions, (HfNbTaTiV)CN, (NbTaTiVZr)CN, (MoNbTaTiZr)CN, and two are found to be multi-phase, (MoNbTiWZr)CN and (HfMoNbTaW)CN. DEED classified all cases perfectly. **Microstructure formation:** The combination of carbonitrides with low DEED (hence multi-phase) as well as precursors that aren't sharing the same local atomic environments [Esters *et al.*, Nat. Commun. 12, 5747 (2021)], creates the possibility of formation of microstructures. Fundamentally, DEED is the inverse of the miscibility gap temperature (see answer **A2.6.** to Reviewer 2), so a DEED value lower than the threshold, implies a miscibility gap temperature above the sintering synthesis one. Therefore, we are operating inside the gap. In addition, if some of the precursors were not sharing the same parent lattice, then such a gap would be bound by a eutectoid isotherm. The lamellar “pearlite-like” formations in three multi-phase carbonitrides indicate that we are in the eutectoid region. The most prominent example (HfMoNbTaW)CN had 4 non-rock-salt precursors, as is shown in the updated Figure 2. We also developed a software code (AFLOW-Spinodal, part of the AFLOW suite), which calculates the radial distribution function of the power spectrum from micrographs, producing angles, orientations, and wavelengths. With AFLOW-Spinodal, we characterized the lamellar microstructure having $\lambda_p \approx 0.9 \mu\text{m}$ principal wavelength. This is quite advantageous, in addition to identifying solid-solutions, DEED also helps to find compositions with microstructure formation. We have included additional text in the article and figures in the Supplementary Information.

Borides: Four additional previously unexplored compositions were synthesized: 2 are found to be single-phase solid-solutions, (CrMoTiVW) B_2 and (CrHfNbTiZr) B_2 , and 2 are found to be multi-

phase, (HfMnTiVZr)B₂ and (CrHfMoTaW)B₂. For the boride samples, DEED again predicted the experimental outcome perfectly.

The article has been modified as follow (changes are highlighted in blue):

Page 2

Within the unexplored compositional space, based on the DEED ranking, we chose as testbeds **nine** new carbonitrides and **eight** new borides. DEED correctly predicted the **functional** synthesizability of all 17 systems. The single-phase compositions we discovered are (HfNbTiVZr)CN, (HfNbTaTiV)CN, (NbTaTiVZr)CN, (HfTaTiVZr)CN, (MoNbTaTiZr)CN, (HfMoNbTaZr)B₂, (HfNbTaTiV)B₂, (CrMoTiVW)B₂ and (CrHfNbTiZr)B₂.

Ultimately, the comparison between the DEED ranking and the experimentally validated synthesizability corroborates the accuracy of the descriptor and even suggested systems with formation of microstructures (pearlite).

Pages 5, 7 – in the “DEED for carbonitrides” section

DEED for carbonitrides. DEED successfully classifies carbonitrides’ synthesizability with low uncertainty — represented by a minuscule misclassification region, shown in Figure 2b. However, when using EFA, there is a noticeable misclassification region that gets even larger when using VEC. To verify the integrity of DEED and to expand the limited high-entropy carbonitrides experimental data, we have synthesized nine new compounds. The precursors for all the carbonitrides were: TiC, VC, NbC, TaC, WC, Mo₂C, TiN, ZrN, HfN, NbN and TaN. All these precursors are rock-salt except for WC (hexagonal, hP2), Mo₂C (orthorhombic, oP12), NbN (hexagonal, hP4), and TaN (hexagonal, hP8). DEED can be used for two purposes: to find new solid-solutions (large values of DEED, regardless of the structure of precursors or decomposition products [Esters *et al.*, Nat. Commun. 12, 5747 (2021)]) and to search for microstructure formation in multi-phase systems (small values of DEED, with concomitant structural incompatibility among precursors/decomposition products). For the solid-solution test, we randomly selected five compounds, (HfNbTiVZr)CN, (HfNbTaTiV)CN, (NbTaTiVZr)CN, (HfTaTiVZr)CN, and (MoNbTaTiZr)CN, having the highest DEED and predicted to be single-phase. For the multi-phase test, we randomly selected four compounds, (HfTaTiWZr)CN, (MoNbTiWZr)CN, (HfTiVWZr)CN and (HfMoNbTaW)CN having small DEED values. In addition, the compounds in this second batch, do not share the same precursor structure (more precisely, the same local-atomic-environment [Esters *et al.*, Nat. Commun. 12, 5747 (2021)]), and therefore the formation of microstructures is possible.

...

The X-ray diffraction (XRD) pattern, shown in Figure 2c, reveals that (HfNbTiVZr)CN ●, (HfNbTaTiV)CN ●, (NbTaTiVZr)CN ●, (HfTaTiVZr)CN ● and (MoNbTaTiZr)CN ● are rock salt and composed of single face centered cubic (FCC) phase primary matrix grains, with trace oxides, namely HfO₂ and ZrO₂, as isolated particles. These oxides are in thermodynamic equilibrium with the carbonitride phase [Réjasse *et al.*, RSC Adv. 6, 100122 (2016); Réjasse *et al.*, J. Am. Ceram. Soc. 100, 3757 (2017)] and therefore should not affect the composition

or phase stability of the compound. As further validation, we have also synthesized a high-entropy carbonitride with carbothermal reduction and found no difference in the final morphology (see Figure SV in the Supplementary Information). Meanwhile, the multi-phase compounds are a combination of FCC and WC hexagonal phases. All DEED' predictions – solid-solution (large DEED) and multi-phases (small DEED) – were confirmed by the nine synthesized carbonitrides; on the contrary, EFA and VEC prediction were sub-optimal, Figure 2b. We conclude that DEED is a reliable descriptor for functional synthesizability of high-entropy transition-metal carbonitrides.

In the multi-phase systems, the W-rich regions exist as irregular-shaped grains. In (MoNbTiWZr)CN \times , (HfTiVWZr)CN \times , and in particular in (HfMoNbTaW)CN \times , the decomposition has promoted the growth of lamellar “pearlite-like” microstructures [Schaffer *et al.*, *The Science and Design of Engineering Materials*. 2nd ed. WCB/McGraw-Hill, (1999); Barrett *et al.*, *Structure of metals*. 3rd ed. Pergamon Press, (1980); Avner, *Introduction to physical metallurgy*. McGraw-Hill, (1964)]. The last three small panels in Figure 2d show magnifications of (HfMoNbTaW)CN \times . We analyzed those pictures with the software AFLOW-Spinodal and determined the principal wavelength, $\lambda_p \approx 0.9\mu\text{m}$, of the lamellae. Larger pictures are available in the Supplementary Information. The result is not surprising. As mentioned before, small DEED values indicate very high $T_c^{(\text{mg})}$, and synthesis would already occur inside the miscibility gap, if $T_{\text{synt}} > T_c^{(\text{mg})}$. Furthermore, if some of the precursors were not sharing the same parent-lattice, then the miscibility gap would be bound by an invariant isotherm (e.g. eutectoid). The lamellar microstructure is an indication of a eutectoid region, and it is mostly prominent in the composition (HfMoNbTaW)CN \times having the largest variety of precursors structure. The incomplete lamellae transformation in in the three compounds with small DEED – (MoNbTiWZr)CN \times , (HfTiVWZr)CN \times , (HfMoNbTaW)CN \times – also suggests that we have not reached the exact eutectoid composition. In conclusion, the additional capability of DEED to help pinpointing regions of microstructure formation in multi-phase systems, is extremely valuable as thermo-mechanical properties are greatly affected by microstructure morphology.

FIG. 2. DEED functional synthesizability prediction performance for high-entropy carbides and carbonitrides. DEED balances the entropy gains with enthalpy costs for forming a single-phase, predicting synthesizability. Here its superior performance is compared with EFA and VEC for (a) carbides and (b) carbonitrides. ● and ✕ represent systems forming single- and multi-phases, respectively. The references for the prior experimental results are listed in Table 1. Our validation experiments presented in this paper are highlighted in blue. The misclassification regions, in gray, encompass all single-phase systems that are on the incorrect side of the single/multi-phase threshold (vertical line). Panel (c) shows the XRD spectra for carbonitrides. The five green spectra show single-phase FCC systems, while the four red lines reveal extra peaks related to the WC hexagonal phase. Panel (d) shows microstructures from SEM analysis for the nine compounds and three magnified regions of (HfMoNbTaW)CN showing the lamellar “pearlite-like” microstructures with principal wavelength of $\lambda_p \approx 0.9 \mu\text{m}$.

DEED for borides. DEED always orders the boride systems that were experimentally observed as both single- and multi-phase – represented as mixed reports in Figure 3a – as less likely to form. These mixed reports show that synthesizability depends on the experimental process [Feng *et al.*, *Scr. Mater.* 199, 113855 (2021)]. Meanwhile, EFA does have a misclassification region and its ranking does not group the mixed reports close to the single-/multi-phase threshold. Like- wise, VEC shows an enormous misclassification region.

Therefore, we conclude that VEC cannot resolve functional synthesizability of these high-entropy ceramics (see Figure SI and Figure SII in the Supplementary Information for further analysis).

To better determine the DEED threshold (due to mixed reports in literature), we synthesized eight new compounds. The four systems (HfMoNbTaZr) B_2 , (HfNbTaTiV) B_2 , (CrMoTiVW) B_2 and (CrHfNbTiZr) B_2 were selected by the highest DEED. As examples for multi-phase systems, (CrHfMoTaW) B_2 , (HfMnTiVZr) B_2 , (CrHfTiYZr) B_2 and (CrHfNbVY) B_2 were randomly selected from the lower end of the calculated DEED range. The following precursors were chosen: MnO₂, WO₃, HfO₂, Nb₂O₅, Ta₂O₅, TiO₂, ZrO₂, V₂O₅, Cr₂O₃, MoO₃, and Y₂O₃.

...

The XRD of the boride ceramics is shown in Figure 3b. All the materials predicted to be single-phase, (HfMoNbTaZr) B_2 ●, (HfNbTaTiV) B_2 ●, (CrMoTiVW) B_2 ● and (CrHfNbTiZr) B_2 ●, show the clean peaks of a AlB_2 hexagonal phase. We find that (CrHfMoTaW) B_2 ✕ and (HfMnTiVZr) B_2 ✕ are composed of two different AlB_2 phases, while (CrHfTiYZr) B_2 ✕ is composed of AlB_2 and Y-based hexagonal phases, resulting in overlapping peak patterns. In (CrHfNbVY) B_2 ✕ the secondary peaks are less pronounced. Comparing the microstructure of the systems in Figure 3b shows that (HfMoNbTaZr) B_2 ● was almost fully dense and showing just a few residue carbon inclusions, (HfNbTaTiV) B_2 ● contained residual carbon and notable porosity, (CrMoTiVW) B_2 ●, in addition to carbon, also contained residual B_4C and showed signs of micro-cracking, while (CrHfNbTiZr) B_2 was mostly homogeneous. Furthermore, we found large grains of different phases in (CrHfMoTaW) B_2 ✕, while (HfMnTiVZr) B_2 ✕ contained a complex microstructure composed of multiple phases. In addition, we saw a reduction in grain-size for (CrHfTiYZr) B_2 ✕ due to pinning based on the secondary phase and, in (CrHfNbVY) B_2 ✕, core shelling and micro-cracking was detected. These expanded experimental results are further evidence that DEED is a consistent descriptor to predict the functional synthesizability of high-entropy borides.

FIG. 3. DEED functional synthesizability prediction performance for high-entropy borides. DEED balances the entropy gains with enthalpy costs for forming a single-phase, predicting synthesizability. Here its superior performance is compared with EFA and VEC for (a) borides. ● and ✕ represent systems forming single- and multi-phases, respectively. ■ show systems with mixed reports of forming both single- and multi-phases. The references for the prior experimental results are listed in Table 1. Our validation experiments presented in this paper are highlighted in blue. The misclassification regions, in gray, encompass all single-phase systems that are on the incorrect side of the single/multi-phase threshold (vertical line). Panel (b) shows the XRD spectra for borides. The four green spectra show a AIB₂ hexagonal phase. In contrast, the red spectra show a combination of two distinct AIB₂ phases or a AIB₂ phase with a Y-based hexagonal phase. Panel (c) shows microstructures from SEM analysis.

Q1.2. Can the authors describe if the oxide impurities have an effect on the capacity to obtain single- or mixed-phase?

A1.2. We are dividing the answer by classes of materials.

Borides. According to XRD and SEM measurements, oxygen impurities are absent in the boride samples. In addition, even if the oxygen impurities were present, oxides do not affect the phase stability of the borides as they are insoluble (transition metal borides tend to be “line-compounds” in the phase diagrams) [Fahrenholtz *et al.*, *Sci. Sinter.* 52, 1 (2020)]. See an example of such a phase diagram below (ZrB₂).

Carbonitrides. On the contrary, for carbides and most likely for carbonitrides, the calculated phase diagrams for transition metal carbides, such as those by Rejasse *et al.* [J. Am. Ceram. Soc. 100, 3757 (2017); RSC Adv. 6, 100122 (2016)] show invariant points for three-phase equilibrium between carbide, oxide, and carbon (very low oxygen-concentration, yet not zero). This is due to the range of solubility of C at equi-composition with the transition metal. This is actually a valuable phenomenon, because it enables the creation of solid-solutions of C and N, even in the anion sublattice, by further stabilizing the compound with reciprocal entropy stabilization (e.g., entropy both in the {metals} and {CN} sublattices). In addition, these carbides and carbonitrides support anion vacancies at very high temperatures, which is advantageous for both mechanical properties and doping/alloying for corrosion/oxidation resistance (applications in energy production and heat management).

Because of this fact, both carbides and carbonitrides have non-zero solubility of oxygen and promote the formation of oxycarbides and, likewise, oxycarbonitrides. Fortunately, the oxygen solubility is relatively low, and the carbides/carbonitride can be in thermodynamic equilibrium with oxides in traces. An example of a ternary oxycarbide phase diagram (Zr-O-C) is pictured above [Réjasse *et al.*, RSC Adv. 6, 100122 (2016); Réjasse *et al.*, J. Am. Ceram. Soc. 100, 3757 (2017)]. The invariant point is at 3% atomic oxygen concentration. Carbothermal reduction of carbides (and carbonitrides), **blue arrows**, will reduce oxygen concentration to this small amount and reach such an invariant point. Further carbothermal reduction alone will not reduce carbon, but instead start precipitating graphite. To “dislodge” the compound from the invariant point, one must concurrently add metal-hydrate (ZrH_2 in the case of Zr-O-C), **brown arrow**, so oxygen reduction occurs with simultaneous formation of water (super-hot steam at 2000 °C). This situation is like the azeotropic point of water/alcohol during fractional distillation limiting the purity of distilled alcohol to high 90s %. The difficulty associated with this extra step and considering that all the oxycarbides have oxygen concentration below the Zr/Hf oxycarbides, make the test unnecessary for this article. Here, it suffices to understand where oxygen goes and how it affects the phase-stability of the solid-solution, once carbothermal reduction has reached the invariant point and graphite starts precipitating.

Metal oxide characterization. For this purpose, we have performed additional EDS-measurements (see Figure SVIII in the Supplementary Information), of our carbonitrides, finding that specific transition metals (Zr, Hf) preferentially segregate into oxide inclusions in the grain boundaries. However, the small amount of oxide in the studied ceramics would mean that the compositional differences in the high-entropy carbonitride phases would be very small (i.e., less than 1 mol%) with the small amount of oxide present in thermodynamic equilibrium with the carbonitride. Further decreasing the overall oxygen content should decrease the volume fraction of the oxide but should not affect the composition or phase stability of the high-entropy carbonitride phase, just its macroscopic fraction. Hence, we are confident that the materials that form single-phases (or multi-phases) in the present study would exhibit the same behavior even if the minor amount of residual oxide phase were eliminated. As a test, we have synthesized (HfTaTiWZr)CN using carbothermal reduction after FAST sintering to reduce oxygen impurities [Fahrenholtz *et al.*, J. American Ceramic Society 91, 1398 (2008); Feng *et al.*, Scr. Mater. 162, 90 (2019)] and found no difference in the phase, with and without the extra process. We also went beyond by adding extra carbon and this will be discussed in an answer **A1.6**.

The article has been modified as follow (changes are highlighted in blue):

Page 5 – in the “DEED for carbonitrides” section

The X-ray diffraction (XRD) pattern, shown in Figure 2c, reveals that (HfNbTiVZr)CN ●, (HfNbTaTiV)CN ●, (NbTaTiVZr)CN ●, (HfTaTiVZr)CN ● and (MoNbTaTiZr)CN ● are rock salt and composed of single face centered cubic (FCC) phase primary matrix grains, with trace oxides, namely HfO₂ and ZrO₂, as isolated particles. These oxides are in thermodynamic equilibrium with the carbonitride phase [Réjasse *et al.*, RSC Adv. 6, 100122 (2016); Réjasse *et al.*, J. Am. Ceram. Soc. 100, 3757 (2017)] and therefore should not affect the composition or phase stability of the compound. As further validation, we have also synthesized a high-entropy carbonitride with carbothermal reduction and found no difference in the final morphology (see Figure SV in the Supplementary Information). Meanwhile, the multi-phase compounds are a combination of FCC and WC hexagonal phases. All DEED’ predictions – solid-solution (large DEED) and multi-phases (small DEED) – were confirmed by the nine synthesized carbonitrides; on the contrary, EFA and VEC prediction were sub-optimal, Figure 2b. We conclude that DEED is a reliable descriptor for functional synthesizability of high-entropy transition-metal carbonitrides.

Page 9 – in the “Experiments: High-entropy carbonitride sample preparation and characterization” section

All sintering was performed under vacuum (~0.4 mPa, pyrometer flush of 12 Pa N₂) over a constant heating rate of 100°C/min and a ramped pressure sequence from 30 MPa to 50 MPa during this heating ramp.

Supplementary, Pages 16-18 – in “Supplementary Methods 1. Experimental information and for high-entropy carbonitrides”

We have evaluated the synthesis of high-entropy carbonitrides across various soak conditions to determine the extent of the entropy stabilization. For example, in Figure SIIIa-b, the constituent six individual powders consolidated into two cubic solid solution phases at 2200 °C/30 min and then stabilized into one cubic single phase at 2300 °C/60 min. This highlights the influence of thermokinetics on entropy-stabilized phases, as substantial diffusion and chemical driving forces are needed to facilitate the coalescence of these different species into a single disordered crystalline phase. In Figure SIIIc-d, more homogenous multiphase mixtures were observed at the 2300 °C/60 min soak relative to 2200 °C/30 min, reinforcing that these multiphase compositions cannot be further stabilized into even fewer phases at higher soak conditions.

The entropy stabilization is not only invariant to minor changes in relative cation content, but also in anion content as well. As shown in Figure SIV, two identical high-entropy carbonitrides were synthesized under the same processing conditions, but one with 5 at% C added and one without. It is seen that no significant change in phase can be discerned from this minor anion content variation. At most, a slight change in relative peak intensity (<5%) may be attributed to the slight differences in carbon stoichiometry. Therefore, the DEED descriptor-based

predictions remain accurate despite minor changes in anion content with respect to ideal equimolarity.

Supplementary, Page 16 – Figure SV added

Figure SV. X-ray diffraction comparison of carbothermal reduction treatment (25 at% C, 1450 °C/3h) on a typical high-entropy carbonitride, (HfTiZrTaW)CN sintered at 2300 °C/60min across (a) full angular range and (b) select oxide peaks.

To further examine the phase stability of high-entropy carbonitrides with respect to anion content, carbothermal reduction (CTR) on blended powders was performed to determine subsequent changes on the resulting high-entropy carbonitride. This CTR was performed under pressureless conditions with a modified FAST die assembly (60mm cavity), where 25 at% C was added and heat treated at 1450 °C/3 h. These conditions were chosen to simultaneously maximize both the CTR driving forces (>1200 °C) and available open porosity for reaction before densification (<1600 °C). The resulting reacted powder was pulverized/deagglomerated with a mortar and pestle, sieved, and sintered at 2300 °C/60 min. As seen in Figure SV, systematic decreases in oxide peak intensity are observed for the post-CTR high-entropy carbonitride and demonstrate oxide reduction. Minor peak intensity changes were observed as previously seen in Figure SIV (i.e. carbon stoichiometry effect), but the overall phases remained invariant to this change.

To further evaluate the compositional phase segregation the microstructures were examined with SEM and multi-phase elemental maps obtained with EDS in Figure SVII. The previously designated cubic phases A/B and hexagonal WC phases were confirmed accordingly, where phase A constituted the microstructure as the continuous, primary matrix phase and WC/phase B appeared as dispersed secondary phases, both isolated and grain boundary-segregated. Additional oxide inclusions were detected as dispersed phases, likely originating from trace impurities from the source powders (e.g. HfO₂, ZrO₂) and minor abrasive wear from blending with YSZ-based media. While the presented EDS-maps (Figure SVIII and SVX) cannot be used to gather quantitative insights into the material compositions, they can be used to examine the oxide inclusions in more detail. After we segmented the SEM image of (HfNbTiVZr)CN (Figure SVIIIa) into three regions: (i) inclusions (red), (ii) grain boundaries (purple) and (iii) grains (green) using Weka [Arganda-Carreras *et al.*, *Bioinformatics* 33, 2424 (2017)] we can compare the average intensity in the region of interest for the different elements. The results

presented in Table SIV show that O, Hf and Zr are enriched in the inclusions, while the intensities of Ta, Ti, and V are reduced compared to the grains. However, the small amount of oxide in the studied ceramics would mean that the compositional differences in the high-entropy carbonitride phases would be very small (i.e., less than 1 mol%) with the small amount of oxide present in thermodynamic equilibrium with the carbonitride. Nonetheless, the overall microstructures are composed of coarse carbonitride grains ($\sim 10\text{-}50\ \mu\text{m}$) with minimal pores. In particular with the tungsten-rich compositions, the immiscible multiphase mixture yielded dispersions of partially-dissolved precursor phases throughout the microstructure, indicating the highly unfavorable nature of attempting dissolve these mutually insoluble components into a single phase. To analyze the lamellar "pearlite-like" microstructure of $(\text{HfMoNbTaW})\text{CN}$ we calculated the radial distribution function of the Fourier-transformed image contrasts. Then, the radial distribution function was smoothed using a Savitzky-Golay filter, to help identify the maxima in the signal, and fitted to a Cauchy-Lorentz distribution, whose peak determines the lamellar wavelength of the microstructure, shown in Figure SVI.

Supplementary, Page 17 – Figure SVI added

Figure SVI. Fourier analysis of the lamellar microstructure in $(\text{HfMoNbTaW})\text{CN}$. The blue dashed line is a fit of the Cauchy-Lorentz distribution to the radial distribution function of the power spectrum of the image. The green dotted-dashed line is the location of the distribution peak.

FIG. SVIII. (a) SEM image of (HfNbTiVZr)CN, contrast optimized to show intra-grain inclusions (b) segmentation into inclusions (red), grain boundaries (purple) and grains (green) using Weka (c) oxygen SEM-EDS map.

Q1.3. *Can this effect be incorporated into DEED?*

A1.3. Absolutely! The presented experimental work has shown no effect of oxygen on phase stability, so for the classes of materials in this submission, incorporating oxygen in DEED is not necessary. However, for different types of ceramics, oxygen can be incorporated into DEED by using two approaches, depending on the expected concentration of the oxygen impurities. **i.** For non-diluted concentrations (greater than 10-20%), new sets of configurational expansions need to be generated with the addition of the new species. Then, quantum mechanical energies need to be calculated to create the convex hull and collect the excited spectrum states, so that DEED can be quantified. **ii.** For diluted concentrations, one can correct the existing quantum mechanical energies of the configurations using perturbation theory and thermodynamics of dilute lattice system [Chepulskii, Curtarolo, *Acta Materialia* 57, 5314 (2009)].

Due to the large oxidation landscape of the transition metals in the disordered system, quantum mechanical metal-oxide calculations would be corrected with the “*Coordination corrected enthalpy methods*”, as implemented in AFLOW-CCE, which perfectly adapts to the task [Friedrich *et al.*, *npj Comput. Mater.* 5, 59 (2019)]. We plan to tackle both research directions in the future to describe the high-temperature oxidation of high-entropy carbides into high-entropy oxy-carbides.

The article has been modified as follow (changes are highlighted in blue):

Pages 2, 3 – in the “DEED descriptor” section

If the alloy contains a sufficient concentration of impurities and/or vacancies, the problem can be treated explicitly by adding extra-species and calculating larger POCC-tiles. For small concentrations, perturbation theory can be employed through variational approaches [Chepulskii, Curtarolo, *Acta Materialia* 57, 5314 (2009)].

POCC’s success has been extended to studies beyond configurational issues; e.g., vibration in high-entropy carbides [Esters *et al.*, *Nat. Commun.* 12, 5747 (2021)] and ultra-high-temperature plasmonic response [Calzolari *et al.*, *Nat. Commun.* 13, 5993 (2022)]. DEED is not limited to

crystalline phases. Being a thermodynamic descriptor predicting functional synthesizability by ranking order/disorder transition temperatures in chemically disordered systems, it can be easily extended to other classes of materials as long as its ingredients (roughness of the enthalpy landscape and thermodynamic density of states) can be accessed. For instance, in non-crystalline/amorphous systems, these quantities can be computed using the melt-quench method [Drabold, *Eur. Phys. J. B* 68, 1 (2009)] to extract a thermodynamic density of states, or through considerations about local atomic environment decomposition [Perim *et al.*, *Nat. Commun.* 7, 12315 (2016)].

Page 5 – in the “DEED for carbides” section

The addition of even more carbon and increasing its chemical potential (with extra pressure, for example) can also be described by DEED. It requires the recalculation of the enthalpy-stability landscape, the formation enthalpies distances of the POCC-tiles, and the statistical momenta as function of the reference carbon chemical potential. It can then be a priori predicted if the carbide synthesizability is preserved or if other phases, with smaller metal solubility, are promoted.

Q1.4. *Regarding the borides, there is a description of sintering that results in the microstructures shown in Figure 3. Both carbonitrides and borides were sintered using similar techniques (i.e., spark plasma sintering), but in different equipment. An HP5–7010G system was used for the borides and an FCT system was used for the carbonitrides. Q1.4a Why change equipment? Q1.4b Would this have an effect on oxide impurities considering changes in vacuum in the two systems used? Q1.4c And again, does this have an effect on the phase stability in the materials?*

A1.4a Two specialized experimental groups produced these samples: the carbonitrides were produced at Pennsylvania State University (Wolfe group), while the boride samples were sintered at Missouri University of Science and Technology (Fahrenheit group), using their respective equipment.

A1.4b The oxygen activity in both systems is set by the equilibrium between the graphite dies and the mild vacuum atmosphere. For both systems, the nominal base vacuum is less than 2 Pa. In addition, the system that was used to prepare the carbonitride compounds has an active 12 Pa N₂ pyro-flush. Hence, oxygen activity during synthesis is very low.

A1.4c As mentioned before, borides do not have oxygen impurities due to lack of solubility, while in carbonitrides, oxygen impurities have not been found to influence phase stability, as stated in the previous comments.

The article has been modified as follow (changes are highlighted in blue):

Page 9 – in the “*Experiments: High-entropy carbonitride sample preparation and characterization*” section

These mixed powders were then sieved (60-mesh) and sintered at a soak temperature/time of 2300 °C/60 min using a 25ton FAST system (FCT Systeme GmbH) with 20mm outer-diameter graphite dies/punches, analogous to previous TiC-TiN carbonitride work [Feng *et al.*, *J. Eur. Ceram. Soc.* 43, 2708 (2023)]. The internal surfaces of the die body were covered with graphite foil to minimize die degradation and each cavity was loaded with about 10g of powder. All sintering was performed under vacuum (~0.4mPa, pyrometer flush of 12Pa N₂) over a constant heating rate of 100 °C/min and a ramped pressure sequence from 30 MPa to 50 MPa during this heating ramp.

Page 10 – in the “*Experiments: High-entropy boride sample preparation and characterization*” section

The prepared powder mixtures were pressed into disks using a uniaxial pressure of 2 MPa. Boro/carbothermal reduction was performed in a resistance-heated graphite element furnace (HP50-7010G, Thermal Technology, Santa Rosa, CA) under mild vacuum (~3 Pa), at 1660 °C for 2.5 h. The reacted samples were crushed by mortar and pestle and sieved through 100-mesh metallic sieve. Reacted powders were loaded in a graphite die that was lined with graphite foil. As prepared samples were densified using two-step spark plasma sintering, as described in our previous studies. [Feng *et al.*, *J. Eur. Ceram. Soc.* 40, 3815 (2020); Feng *et al.*, *Scr. Mater.* 199, 113855 (2021)] The first step was heating to 1650 °C under 15 MPa applied pressure with heating rate 100 °C/min, and dwell time at the highest temperature for 5 min. This step was used to promote removal of residual oxides and surface oxide impurities.

Q1.5. Also, in some cases, extra carbon was added to reduce oxides. Does this then have an effect on the phase stability?

A1.5. In borides, oxygen is eliminated through carbo-reduction until correct stoichiometry is reached, as described in answer **A1.2**. For carbonitrides, the addition of extra carbon beyond what is required by carbo-reduction does not influence the phase stability in these systems due to the large range of solubility of carbon in the rock-salt carbide [Dippo *et al.*, *Sci. Rep.* 10, 78175 (2020)]. In fact, we have confirmed that adding additional carbon (5 at%) to (MoNbTaTiZr)CN brought no changes to the overall stability. Further added carbon would precipitate as graphite [Réjasse *et al.*, *RSC Adv.* 6, 100122 (2016); Réjasse *et al.*, *J. Am. Ceram. Soc.* 100, 3757 (2017)]. More discussion about carbon-graphite precipitation can be found in answer **A1.2**.

The article has been modified as follow (changes are highlighted in blue):

Page 5 – in the “DEED for carbonitrides” section

The precursors were dry mixed for 24h at 200RPM with 9:1 BPR, using a 2mm YSZ media and 125mL HDPE bottles. An additional 5 at% C was included in each powder blend to promote oxide reduction at elevated temperatures, without affecting the phase stability of the high-entropy ceramics, as previously reported in literature [Dippo *et al.*, *Sci. Rep.* 10, 78175 (2020); Réjasse *et al.*, *RSC Adv.* 6, 100122 (2016); Réjasse *et al.*, *J. Am. Ceram. Soc.* 100, 3757 (2017)] (see Figure SIV in the Supplementary Information). Then the precursors were sieved through a 60-mesh and sintered at various soak conditions (e.g. 2200 °C/30 min, 2300 °C/60 min) using a 25 ton FAST system. A detailed description of the process can be found in the Method section.

Supplementary, Page 15 – Figure SIV added

FIG. SIV. X-ray diffraction comparison of adding 5 at% trace carbon on a typical synthesized high-entropy carbonitride, (MoNbTaTiZr)CN, sintered at 2300 °C/60 min.

Supplementary, Page 16 – in “Supplementary Methods 1. Experimental information and for high-entropy carbonitrides”

The entropy stabilization is not only invariant to minor changes in relative cation content, but also in anion content as well. As shown in Figure SIV, two identical high-entropy carbonitrides were synthesized under the same processing conditions, but one with 5 at% C added and one without. It is seen that no significant change in phase can be discerned from this minor anion content variation. At most, a slight change in relative peak intensity (<5%) may be attributed to the slight differences in carbon stoichiometry. Therefore, the DEED descriptor-based predictions remain accurate despite minor changes in anion content with respect to ideal equimolarity.

Q1.6. *Can this be correlated to the computational work in some way?*

[Text and figure redacted]

[Text redacted]

The article has been modified as follow (changes are highlighted in blue):

Page 5 – in the “DEED for carbides” section

The addition of even more carbon and increasing its chemical potential (with extra pressure, for example) can also be described by DEED. It requires the recalculation of the enthalpy- stability landscape, the formation enthalpies distances of the POCC-tiles, and the statistical momenta as function of the reference carbon chemical potential. It can then be a priori predicted if the carbide synthesizability is preserved or if other phases, with smaller metal solubility, are promoted.

Q1.7. *If better controlled specimens can be prepared ([Q1.7a] without oxide contamination), this manuscript would be outstanding. At the moment, however, there is a lack of rigor in the experiments, variability is not clear from sample to sample ([Q1.7b] for example, did the authors prepare more than just one sample of each composition?), and [Q1.7c] there seems to be changes in methodologies for sintering that are not fully justified.*

A1.7. We thank the referee for the very positive words. We have taken all the concerns of the reviewer into great consideration.

Therefore, we have considerably extended the experimental analysis: we have more than doubled the set of experiments, validated synthesizability versus large/low DEED, discussed the formation of microstructures in small DEED compounds, explained the reason for the two different experiment locations, addressed the oxygen impurity stability, the effect of carbon addition in carbo-reduction, and proposed methods to study impurities with large or small concentrations.

We have divided the sentence of the reviewer into three questions **Q1.7a**, **Q1.7b**, and **Q1.7c**.

A1.7a. Borides have no oxide contamination due to the line-compound nature of transition-metal borides. The traces of oxides in carbonitrides are due to the transition-metal oxycarbide and oxycarbonitride invariant points and do not affect the phase stability. Some metal-oxides precipitate in grain boundaries. The issue has been discussed in more detail in answer **A1.2**.

A1.7b. Yes, samples at the same compositions were prepared to double check the statements. Due to the cost of the precursors, as well as the finite amount of FAST/SPS and X-ray time available, we chose to prioritize the production of many different compositions. This allowed us to achieve a clearer picture regarding the existence/absence of the solid-solution, and their morphology for more systems. Repeated samples were only used to establish consistent experimental protocols within different materials classes. Relevant examples are in Supplementary Figure SIII, showing the influence of soaking time on the X-ray diffraction patterns.

We have also added Supplementary Figures SVII and SXII to show the actual samples and their usage.

A1.7c. Different families - borides versus carbides/carbonitrides - require different protocols as described in the Methods. Within each family, all the samples are now prepared consistently. The details are also given in the Method section.

To conclude, all the Reviewer 1 questions were answered and followed by modifications to the article, which we now believe it has greatly improved. We hope the reviewer can now consider the manuscript to be “outstanding” in its full extent.

The article has been modified as follow (changes are highlighted in blue):

Supplementary, Page 15 – Figure SIII added

FIG.SIII. X-ray diffraction comparison of high-entropy carbonitride phase evolution from mixed powders across soak conditions of **a** (HfNbTiVZr)CN, **b** (HfTaTiVZr)CN, **c** (HfTaTiWZr)CN, and **d** (HfTiVWZr)CN.

Supplementary, Page 17 – Figure SVII added

FIG. SVII. Sample disks of replicated compositions. Panel **a** presents the samples used to study different FAST soak conditions (2200 °C/30 min, 2300 °C/60 min) and samples in panel **b** were used to understand the effect of using additional Carbon during the manufacturing of the samples. Either through carbothermal reduction treatment (CTR - 25 at% C, 1450 °C/3 h) or by direct addition of 5 at% trace carbon. All samples in **b** were sintered at 2300 °C/60 min.

Supplementary, Page 22 – Figure SXII added

FIG. SXII. Sample disks of high-entropy borides.

Reviewer #2 (Comments to the Authors)

Q2.1. *In this manuscript, the authors present a disordered enthalpy-entropy descriptor, called DEED, that provides a classification of the so-called “synthesizability” of multicomponent ceramics. They show the application of DEED to high-entropy carbides, carbonitrides and borides based on the published experimental data together with several new compounds synthesized by the authors. The results, as summarized in Table 1, show improved classification compared to other descriptors such as valence electron concentration (VEC) and entropy-forming-ability (EFA).*

A2.1. We thank Reviewer 2 for his/her synopsis of the work presented in our paper. When summarizing DEED, our disordered enthalpy-entropy descriptor, we appreciate them pointing out the need to define “synthesizability”, since the understanding and application of this concept goes hand in hand with DEED, and therefore with the relevance and impact of our work. It made the authors dig deeper into literature, allowing us to look critically and realize that behind the lack of consensus about its definition, synthesizability goes beyond an abstract concept. Instead, here we speak of a synthesizability that depends on the material and the process (i.e., we look at fixed processes and then look for the material), and therefore a functional synthesizability.

At the same time, we need to emphasize that a theoretical attempt to define “synthesizability” does not have a consensus. For instance, there have been attempts to define synthesizability based on activation energies of transformations [Oganov, *Modern Methods of Crystal Structure Prediction*. 1st ed. Wiley, (2010); Dellago *et al.*, *J. Chem. Phys.* 108, 1964 (1998)], on difference in energies between glassy and ordered configurations/polymorphs [Aykol *et al.*, *Sci. Adv.* 4, eaaq0148 (2018)], remnant metastability [Sun *et al.*, *Sci. Adv.* 2, e1600225 (2016)], enthalpies of the decomposition reactions [Bartel *et al.*, *npj Comput. Mater.* 5, 4 (2019)], and distance from the convex-hull (self-consistent with experiments or observed) [Wang *et al.*, *Joule* 2, 914 (2018); O’Donnell *et al.*, *Chem. Mater.* 32, 3054 (2020); Singstock *et al.*, *J. Am. Chem. Soc.* 143, 9113 (2021)].

The similarity in “synthesizability” language is only superficial and it does not translate in an operative procedure. In fact, these listed approaches are unsuitable for our classes of systems as they **i.** do not consider heat-exchange (entropy) during the nucleation of the phase to inject (and retain) the disorder in the phase for a functional approach to synthesis (functional synthesizability) and **ii.** deal with the formation of ordered-compound and not multi-component disordered solid-solution (compound versus disorder).

Functional Synthesizability. For instance, in a known experiment about the samarium-boron alloy system, in which both SmB₄ and SmB₆ are line-compounds, it was possible to nucleate SmB₄ at {Sm+6B} composition (with some precipitation of samarium) and SmB₆ at {Sm+4B} composition (with some precipitation of boron) [Yong *et al.*, *Appl. Phys. Lett.* 105, 222403 (2014)]. The synthesizability conundrum was explained in terms of heat-exchange during nucleation, and the way the latter could be “tailored” by operating on substrate temperatures. Same results could be extended to the yttrium boride system. A non-functional approach to synthesizability would not address the apparent paradox.

Since the ultimate test of synthesizability is the tangible production of the material, the first step to understand global synthesizability is the operation of a chosen process to determine its synthesis rules. Here, our current work focuses on hot-pressed sintering transition metal carbides/carbonitrides and borides ceramics, and the descriptor needs to deal with synthesizability in its functional emphasis. As such we understand “synthesizability” as the capability to thermally stabilize classes of materials with a functional procedure and to retain that stabilization upon removal of the synthesis conditions. To avoid misunderstanding, we call this concept “functional synthesizability”.

Compounds versus disorder. Disorder requires entropy, and entropy requires heat. While increasing temperature to a mixture is relatively easy, quantifying its entropy-content and establishing if it is preserved upon removal of heat, is not. Such a study, extended to a large set of disordered unknown candidates so that materials discovery can be performed, becomes quite challenging.

To the best of our knowledge, almost all the works in synthesizability relate to compound formation. With the exception of the “entropy forming ability” proposed by us in 2018 for carbides, the other discussion in solid-solution synthesizability is based on energetic picture: the distance from the hull of a few combinations of sampled configurations (no “entropy-content” of the mixture) [O’Donnell *et al.*, *Chem. Mater.* 32, 3054 (2020)]. The challenge in trying to capture entropy-content following the Boltzmann idea of state-degeneracy remains untackled.

DEED was created to exactly address this issue. With the calculation of an exhaustive set of chemical decorations (up to a cut-off), the thermodynamic density of states is evaluated. The ensemble first-momentum gives the enthalpy-cost of a solution with respect to phase separation. The ensemble second momentum gives the entropy-gain. The picture resembles the Boltzmann idea, except that for phase space sampling: discrete versus continuous. The ratio between enthalpy-cost and entropy-gain gives the temperature of the order-disorder transition (without considering precursors).

DEED is a descriptor defined as the square root of enthalpy-cost/entropy-gain, and it indicates the strength of the miscibility gap transition temperature (DEED has dimensions inverse temperature). DEED relates to a quantity which can be measured (the transition temperature), and it helps experimentalists in ceramics’ synthesis: miscibility gaps are quite difficult to obtain due to the slow sintering-kinetic of the soaking period.

To enhance clarity of the article, we have extended the introduction with additional text.

The article has been modified as follow (changes are highlighted in blue):

Pages 1, 2

To overcome the impasse in discovering new high-entropy ceramics, two factors need to be considered: synthesizability and descriptors quantifying it.

Materials and functional synthesizability. The lack of consensus about “synthesizability” in literature [Oganov, *Modern Methods of Crystal Structure Prediction*. 1st ed. Wiley, (2010); Dellago *et al.*, *J. Chem. Phys.* 108, 1964 (1998); Sun *et al.*, *Sci. Adv.* 2, e1600225 (2016); Aykol *et al.*, *Sci. Adv.* 4, eaaq0148 (2018); Wang *et al.*, *Nature* 561, 914 (2018); Bartel *et al.*, *npj Comput. Mater.* 5, 4 (2019); O’Donnell *et al.*, *Chem. Mater.* 32, 3054 (2020); Singstock *et al.*, *J. Am. Chem. Soc.* 143, 9113 (2021)] does not translate into an operative procedure for the

synthesis, critical for autonomous materials discovery [Abolhasani *et al.*, Nat. Synth. 2, 483 (2023)]. The conundrum lies in the observer's reference. Synthesizability is a functional of both the material and the synthesis conditions, and the user needs to explore the appropriate path across these two degrees of freedom, i.e., the direction maximizing the outcome. While the priority is usually put on “synthesizability as an intrinsic material property” with consequent synthesis conditions to be determined later (materials' point of view), here we focus on the complementary picture (process' point of view). Starting from a chosen process, we build a descriptor to establish which materials are synthesizable with it. We call it “functional synthesizability”, which is intrinsically connected to a given manufacturing process. In the present case – transition-metals disordered ceramics – the process is hot-pressed sintering with all its various implementations [Oses *et al.*, Nat. Rev. Mater. 5, 295 (2020)], and the classes of materials are high-entropy carbides, carbonitrides and borides.

Building an entropy/enthalpy descriptor related to an observable. For our purposes, the descriptor must relate to an observable quantity and also be able to answer tangible synthesis questions [Abolhasani *et al.*, Nat. Synth. 2, 483 (2023); Hart *et al.*, Nat. Rev. Mater. 6, 730 (2021)]. Since we are interested in the formation of high-entropy materials, an effective descriptor should consider the “entropic-gain in generating disorder”, which, through Boltzmann's view, is represented by the degeneration of the states (in a discretized population), or more generally, by the thermodynamic density of states spectrum $\Omega(E)\delta E$ (in a continuum population). This is the essence of EFA [Sarker *et al.*, Nat. Commun. 9, 4980 (2018)], which ranked the candidates' solid-solution synthesizability by associating the entropic gain to the variance (second momentum) of Ω [Sarker *et al.*, Nat. Commun. 9, 4980 (2018)]. EFA worked flawlessly for high-entropy carbides, but had difficulties in systems characterized by non-homogeneous enthalpy landscapes [Hossain *et al.*, Adv. Mater. 33, 2102904 (2021)]. In addition, when EFA was introduced in 2018, there was very little available ab initio enthalpy convex-hull data for high-entropy ceramics in the various databases, so the evaluation of the “enthalpy-costs to generate disorder” could not be performed in its full extent. In fact, the entropy-gain can be estimated through the statistical ensembles of configurational formation enthalpies weighted with their degeneracies (the reference ground state is not necessary due to the differential nature of Ω), the same cannot be said of the enthalpy-cost: the latter requires the ensemble of the distances of such configurational formation enthalpies to the convex-hull – the Gibbs free-energy line of the stable configurations (the underlying ground state). As such, an effective descriptor balancing “entropic-gain” versus “enthalpy-cost” for solid-solution formation requires two ingredients: the first is given by an ensemble quantifying the spread of $\Omega(E)$ to capture how many configurations accumulate around E , and the second is an ensemble quantifying the expectation of the distance ΔH of such phases from the ordered ground-state hull. Both quantities can now be quantified for several classes of high-entropy ceramics, as we have populated the aflow.org repository with ample and appropriate thermodynamic data [Esters *et al.*, Comput. Mater. Sci. 216, 111808 (2023)] and developed the appropriate computational modules for recursive phase diagram searches [Oses *et al.*, J. Chem. Inf. Model. 58, 2477 (2018); Oses *et al.*, Comput. Mater. Sci. 217, 111889 (2023)]. Now, we can introduce a new forming ability descriptor which includes the statistical information of the thermodynamic density of states. The disordered enthalpy-entropy descriptor (DEED) allows us to classify the functional synthesizability of multi-component ceramics, including the ones with non-flat enthalpy landscapes. DEED uses the first and second momenta of Ω , associating the enthalpy-loss to the former, and the entropy-gain to the inverse of the latter, respectively.

DEED is intrinsically connected to an observable. In fact, by DEED being defined as the ratio between entropy-gain and enthalpy-loss, it retains the physical meaning of the inverse of an order/disorder temperature, the cross-over between two ideal environments – a perfectly ordered and a perfectly disordered one. As such, the inverse of DEED is correlated to the miscibility gap critical temperature and can be used as a descriptor for functional synthesizability for hot-pressed sintering: large values of DEED will indicate small miscibility gap temperature, easy to overcome by the sintering temperature for the formation of the solid-solution. Small values of DEED will indicate large critical temperatures, possibly above the synthesis temperature, no formation of solid solution, and instead nucleation of microstructures. The threshold for functional synthesizability predictions of DEED can be found self-consistently from available experiments, and then extrapolated to predict the formation, or not, of new solid-solutions. In essence, different processes, structures, and chemistries can be captured by the DEED framework, as long as functional synthesizability can be mapped into a logical relationship between synthesis and miscibility gap temperatures. Here, we show that DEED accelerates the computational discovery of novel disordered ceramics by allowing rapid pre-screening, avoiding expensive experimental preparation, and enabling structure and property predictions [Kaufmann *et al.*, *npj Comput. Mater.* 6, 42 (2020); Hart *et al.*, *Nat. Rev. Mater.* 6, 730 (2021)].

...

Ultimately, the comparison between the DEED ranking and the experimentally validated synthesizability corroborates the accuracy of the descriptor and even suggested systems with formation of microstructures (pearlite).

Page 3 - in the “DEED descriptor” section

Since high DEED indicates ease of synthesis, Θ is a fingerprint of the order-disorder transition, the critical temperature of the miscibility gap $T_c^{(mg)}$. Θ is expected to be monotonically correlated with $T_c^{(mg)}$ as it has the functional shape of a ratio between order-disorder enthalpy versus entropy estimated changes, and therefore it represents the temperature locus of the order-disorder Gibbs free energies crossover. Θ is only an approximation of $T_c^{(mg)}$ due to the many approximations that are made to create DEED, e.g. finite configurational sampling of Ω , lack of precursors/fluctuations and the finite size effects of the calculations. However, Θ can still be used to compare and rank synthesizability of similar systems.

Q2.2. *Overall, the descriptor DEED serves to provide a classification of multicomponent ceramics only in terms of the formation of single vs. multi-phases. It has a limited value and lacks a general applicability for guiding the development of new multicomponent ceramics. Hence, the work lacks general relevance and potential impact to warrant its publication in Nature.*

A2.2. We respectfully disagree. In our work, we have built, calculated, and compared DEED and its thresholds with the available experimental results. While we have chosen high-entropy carbides, carbonitrides, and borides due to their enormous technological importance, the approach is applicable to many other problems. **DEED is a thermodynamic quantity** correlating with the inverse of the configurational order/disorder transition temperature, constructed to retain

calculation accessibility for rapid deployment of materials discovery. There is nothing preventing DEED to be used on metallic solid-solutions, high-entropy alloys, or other type classes of materials in which entropy promotion, through latent-heat, controls synthesis. All the systems in which the thermodynamic density of excited states can be parameterized are accessible to DEED. The relationship between DEED and observable quantities are discussed in answer **A2.1**.

Our choice of materials is due to their technological relevance: high-hardness and high-temperature, chemical-resilience, thermal barrier protection, wear and corrosion-resistant coatings, thermoelectrics, batteries, and catalysts [Oses *et al.*, Nat. Rev. Mater. 5, 295 (2020); Jiang *et al.*, Science 371, 830 (2021); Li *et al.*, Ceram. Int. 47, 14341 (2021); Liu *et al.*, J. Mater. Sci. Technol. 88, 143 (2021); Wei *et al.*, Nat. Mater. 19, 1175 (2020); Zou *et al.*, Nat. Commun. 6, 7748 (2015); Lee *et al.*, Nat. Commun. 12, 5474 (2021)], and the recent discovery ultra-high-temperature plasmonics [Calzolari *et al.*, Nat. Commun. 13, 5993 (2022)] with possible revolutionary impacts in energy-production/heat-management and aerospace.

Q2.3. *A serious issue of the work is that the candidate materials considered are limited to the stoichiometric compounds consisting of early transition metal elements with a priori knowledge of a specific lattice structure.*

A2.3. We respectfully disagree again: compounds of different compositions were calculated and used to get DEED.

Stoichiometry. Contrary to the statement of the referee about “stoichiometric compounds”, anionic variable-stoichiometry is used by the convolutional approach to produce descriptors in region of the concentration-space, which is difficult for computational characterization (see Figure 1b). In the article we have considered transition metal carbonitrides with the carbon and nitrogen concentrations spanning $\{C_1, N_2\}$, $\{C_1, N_1\}$, and $\{C_2, N_1\}$, and we have used “spectral convolution” to get the DEED spectra across the solid-solution. All the five-transition metal-carbonitrides were calculated at these stoichiometries and produced DEED appropriately.

The theory and the appropriate convergence tests have always been included in the Method subsection “*Theory: convoluting POCC-tiles*” which also includes a benchmark between non-stoichiometry vs stoichiometry approaches. We are confident the referee oversaw this section, and therefore to avoid future misunderstandings we have changed this section title to “*Theory: convoluting variable concentration POCC-tiles*”. In addition, DEED has no issue in describing other types of off-stoichiometry anionic concentrations (e.g., carbon vacancies) by adding extra species or with perturbation theory as we have discussed in the answer **A1.3** to Reviewer 1.

Equiatomic metal-composition of the transition-metals were chosen so the compound prediction would be as restrictive as possible in the characterization of functional synthesizability. As mentioned before, DEED is a measure of the locus of order-disorder transition temperature (miscibility gap). As such, the highest critical temperature is expected to be near the center of the phase diagram, at equal composition. Calculations and constant-temperature sintering experiments performed at that composition will be the most possible restrictive with respect to the DEED prediction, and the experiment vs theory comparison will give the most accurate DEED threshold. As such, we disagree with the opinion about the issue. For us it is not an issue, but the best composition choice to get accurate DEED thresholds.

To avoid misunderstandings about stoichiometry and crystal structure, we have extended the article.

The article has been modified as follow (changes are highlighted in blue):

Pages 3, 5 - in the “DEED descriptor” section

Calculating DEED becomes impossible when there is a growing number of species, due to the immense number of representative states that need to be computed. To overcome this challenge, we have developed an approximated DEED that can be calculated out of a convolution of possible-to-obtain sets of POCC- tiles with the same stoichiometry as the parent structure. This convolutional approach is illustrated in Figure 1b and described in the Methods section. Global stoichiometry was chosen to be equiatomic for the transition-metals, as the miscibility gap transition temperature tends to be higher around the central part of the phase diagram. This guarantees that the characterization of functional synthesizability by DEED is as restrictive as possible. Crystal structures are the ones forming in these classes of high-entropy ceramics at that particular composition [Mehl *et al.*, *Comput. Mater. Sci.* 136, S1 (2017); Hicks *et al.*, *Comput. Mater. Sci.* 161, S1 (2019); Hicks *et al.*, *Comput. Mater. Sci.* 199, 110450 (2021)]: rock salt for carbides/carbonitrides and AlB₂ for borides [Osés *et al.*, *Nat. Rev. Mater.* 5, 295 (2020); Sarker *et al.*, *Nat. Commun.* 9, 4980 (2018); Kaufmann *et al.*, *npj Comput. Mater.* 6, 42 (2020); Zhang *et al.*, *J. Mater. Chem. A* 7, 22148 (2019); Feng *et al.*, *Annu. Rev. Mater. Res.* 51, 165 (2021)]. DEED is a thermodynamic descriptor and is not limited to stoichiometric compounds. The user can choose compositions, crystal structures, and chemistries of interest.

Q2.4 *It is not clear if the DEED can be used to address some of the more basic questions on the synthesis of multicomponent ceramics, such as whether a crystalline or an amorphous structure will form;*

A2.4. DEED is a purely thermodynamic quantity and therefore can be obtained for amorphous systems. However, we agree that it was unclear. For amorphous systems, two possible pathways are available to obtain a sampling of these states. First, one can create samples using a melt-quench method [Drabold, *Eur. Phys. J. B* 68, 1 (2009)]. In this case, the starting point is an equilibrated atomic configuration of the material in the liquid phase. From there, the liquid is quenched numerically (using molecular dynamics) to repeatedly simulate the glass phase transition. The resulting set of atomic configurations without long-range order can then be used to calculate the thermodynamic density of states.

The second technique is a spatially informed POCC approach based on local atomic environment expansion [Perim *et al.*, *Nat. Commun.* 7, 12315 (2016)]. In this case, the weights of POCC tiles are reshuffled so that the distribution of the unique atomic environment in all POCC tiles is aligned with the distribution characterizing the amorphous system of interest. This ensures that the proper close-range order is preserved. Based on the thermodynamic density of states generated from either pathway, DEED can be calculated to investigate amorphous systems.

The article has been modified as follow (changes are highlighted in blue):

Page 2 – in the “DEED descriptor” section

DEED is not limited to crystalline phases. Being a thermodynamic descriptor predicting functional synthesizability by ranking order/disorder transition temperatures in chemically disordered systems, it can be easily extended to other classes of materials as long as its ingredients (roughness of the enthalpy landscape and thermodynamic density of states) can be accessed. For instance, in non-crystalline/amorphous systems, these quantities can be computed using the melt-quench method [Drabold, *Eur. Phys. J. B* 68, 1 (2009)] to extract a thermodynamic density of states, or through considerations about local atomic environment decomposition [Perim *et al.*, *Nat. Commun.* 7, 12315 (2016)].

Q2.5. *what type of crystal structure(s) will form for single or multi-phase systems;*

A2.5. The crystal structure for these high-entropy ceramics classes is already well established in literature. The equi-composition field is rock-salt for HE-carbides and HE-carbonitrides and AlB_2 for HE-diborides [Sarker *et al.*, *Nat. Commun.* 9, 4980 (2018); Kaufmann *et al.*, *npj Comput. Mater.* 6, 42 (2020)]. Given our experimental experience, extending crystal structures at this composition is a second priority than identifying the systems with the highest probability of forming.

We have updated the manuscript for further clarification. The appropriate text is reported in answer **A2.3** and **A1.3**.

Q2.6. *what is the applicability of DEED for non-stoichiometric ceramics.*

A2.6. DEED does not differentiate stoichiometric versus non-stoichiometric ceramics. DEED is simply the measure of the interplay between enthalpy-cost and entropy-gain, regardless of composition. To address stoichiometries different than the calculated ones ($\{\text{C}_1, \text{N}_2\}$, $\{\text{C}_1, \text{N}_1\}$, $\{\text{C}_2, \text{N}_1\}$), it is necessary to calculate enthalpies for a complete set of configurations, such as POCC-tiles, obtain their thermodynamic density of states (tDOS), get the convex-hull and calculate the momenta of the tDOS. As previously discussed, DEED is general and can apply to any disordered system in which the thermodynamic density of excited states can be parameterized. Within a solid-solution field, stoichiometries should be chosen wisely to make use of the convolutional-POCC (Figure 1b) to accelerate the calculation. For instance, below we show possible calculation effort for different MoNbTaTiZr carbonitrides stoichiometries. The POCC systems are based on the $\text{AB}_{\text{cF8}}\text{225}_a\text{b}$ prototype. The average calculation time of 1050 CPU hours per calculation is based on the $(\text{MoNbTaTiZr})_2\text{C}_4\text{N}_6$ system presented in the work. Cluster time is calculated based on the median CPU cores (226592) of the TOP100 supercomputers in June 2023 (see <https://www.top500.org/lists/top500/list/2023/06/>). The POCC algorithm can also be used to find the right compositions to perform the off-stoichiometric convolutional-POCC so that the parameterization is the fastest.

system	structures	uniq. structures	CPU years	cluster time
(MoNbTaTiZr) ₂ C ₅ N ₅	514382400	17492020	2.10 million	9.25 years
(MoNbTaTiZr) ₂ C ₄ N ₆	6000	490	58.7	2.3 h
Mo ₃ Nb ₁ Ta ₂ Ti ₂ Zr ₂ C ₄ N ₆	285768000	8943545	1.07 million	4.73 years
Mo ₄ Nb ₁ Ta ₁ Ti ₂ Zr ₂ C ₄ N ₆	142884000	5005200	600 thousand	2.65 years
Mo ₅ Nb ₁ Ta ₁ Ti ₁ Zr ₂ C ₄ N ₆	57153600	2236835	268 thousand	1.18 years
Mo ₆ Nb ₁ Ta ₁ Ti ₁ Zr ₁ C ₄ N ₆	19051200	801614	96.1 thousand	0.42 years

In the manuscript, we mention the LTVC method [Lederer *et al.*, Acta Materialia 159, 364 (2018)] as a possible approach to calculate the miscibility gap for all-possible concentrations by using the POCC and Cluster Expansion data obtained from the stoichiometric concentration.

We have updated the manuscript for further clarification. The appropriate text is reported in answer **A2.3** and **A1.3**.

Q2.7. *Anther serious issue is that the term of “synthesizability” is used vaguely throughout the manuscript.*

A2.7. This topic has already been discussed in our answer **A2.1**, and to enhance clarity of the article, we have changed the wording synthesizability to “functional synthesizability” in the manuscript as appropriate. Nevertheless, we thank Referee 2 for his/her comments about synthesizability because it made the authors dig deeper into literature beyond the current paper. It allowed us to look critically and realize that behind the lack of consensus about its definition, synthesizability goes beyond an abstract concept. Instead, we speak of a synthesizability that depends on the material and the process (i.e., we look at fixed processes and then look for the material); therefore a functional synthesizability. We have clarified “synthesizability”, making it more precise and consistent throughout the manuscript. We have addressed the issue in full extent in answer **A2.1**. The term is quite vague in literature, due to the lack of consensus in its definition. Since our presented work is about the capability of sintering transition metal ceramics, the descriptor needs to deal with synthesizability in its functional power. As such we understand “synthesizability” as the capability of being able to thermally stabilize classes of materials with a functional procedure and retain that stabilization thereafter. To avoid misunderstanding we call this concept “functional synthesizability”.

Nevertheless, to enhance clarity of the article, we have changed the wording synthesizability to “functional synthesizability” in the manuscript as appropriate.

Q2.8a. *The calculated DEED value is not directly connected to any experimentally measurable material property.* **Q2.8b.** *For the ceramic systems studied in the manuscript, the threshold value of DEED for classification is dependent on each specific set of constituent elements chosen, and likely dependent sensitively on (stoichiometric/non-stoichiometric) composition and* **Q2.8c** *crystal structure, etc.* **Q2.8d** *It is not clear how the threshold value is determined and what is the amount of experimental input is needed.* **Q2.8e.** *Moreover, as the authors state, “synthesizability depends on the experimental process”.*

Q2.8. We respectfully disagree with these arguments.

Q2.8a. DEED is directly connected to a materials property, the miscibility gap temperature.

As discussed in answers **A2.1**, and **A2.2**, DEED is defined as a thermodynamic quantity, the ratio between enthalpy-cost and entropy-gain for the formation of a solid-solution. As such, DEED has the same thermodynamic connection to the temperature locus for zero Gibbs free energy difference. The order/disorder locus, i.e. the miscibility-gap boundary line. Since the sampling of the configurational phase space is discrete, instead of continuous, and we do not consider vibrational effects which are not always relevant in carbides [Esters *et al.*, *Nat. Commun.* 12, 5747 (2021)], then the DEED temperature, $\Theta \equiv [k_B(\text{DEED})]^{-1}$ Equation (4), only approximates the true miscibility gap line. By increasing the sampling (larger POCCs, Cluster Expansion, LTVC, all giving a denser thermodynamic density, other entropic phenomena), it is possible to obtain a DEED value closer to the true inverse of the miscibility gap temperature. It is simply a matter of better parameterizing the computational ingredients of DEED. Hence, the definition remains the same.

Furthermore, by performing experimental sintering at a given temperature, a solid solution will form only if its critical miscibility temperature is lower than the sintering temperature. This implies that it must have high DEED. The opposite is also true, low DEED implies multi-phase. As such, the DEED threshold is not due to DEED but to the synthesis conditions, concentrations and chemical families that are operated in hard ceramics sintering. DEED is a descriptor for “functional synthesizability”. The difficulty in measuring miscibility gaps in ceramic systems makes DEED even valuable for these classes of materials.

Q2.8b. The threshold obviously follows the choice of constituent and stoichiometries.

As mentioned before, it is a matter of reference systems: materials seeking for equipment versus equipment seeking for materials. In the former, “experimental procedures” are searched for the purpose of synthesizing a chosen mixture. In the latter, “experimental procedures” are chosen, and synthesizable materials are then searched. Miscibility gaps in alloys and ceramics have concave dome-like shapes (possibly truncated if peritectoid/eutectoid isotherms are present) and have divergent solvus-lines with decreasing temperature. Given a fixed synthesis temperature, the synthesizability threshold will change as the difference between the synthesis temperature and the miscibility-gap line. As such, the threshold does follow composition and chemistry, as per thermodynamic considerations.

Q2.8c. The threshold does not depend on the crystal structure.

The threshold relates to the miscibility gap and sintering temperature and therefore does not depend on crystal structure. As such the DEED threshold remains the same. However, if the user were calculating the wrong crystal structures, his/her DEED values would be low and not suggest the formation of solid solution. As such, a wise researcher would quickly calculate DEED for several parent lattices and find the one with the highest DEED, hence the highest probability of mixing. The materials under considerations do not require this extra step as their structure is already well known.

Q2.8d. The threshold is determined self-consistently.

Given a monotonic set of DEED values, a type of synthesis technique, and a set of experimental findings, the threshold of DEED is the boundary number between solid-solution and multi-phase. Near the threshold experiments should be well performed to allow enough soaking time to overcome kinetics.

Q2.8e. Functional synthesizability clearly depends on the experimental process. If we take a set of metals carbides, very high-pressure/high-temperature sintering does generate diamond with metal impurities, while medium pressure/high-temperature will generate graphitic carbon plus carbides, showing diamond functionally synthesizable. As discussed in the answers to Referee 1, all these intrinsic conditions can be included in DEED, so that the parameterization is done for the functional process under consideration.

The article has been modified as follow (changes are highlighted in blue):

Page 2

Small values of DEED will indicate large critical temperatures, possibly above the synthesis temperature, no formation of solid solution, and instead nucleation of microstructures. The threshold for functional synthesizability predictions of DEED can be found self-consistently from available experiments, and then extrapolated to predict the formation, or not, of new solid-solutions. In essence, different processes, structures, and chemistries can be captured by the DEED framework, as long as functional synthesizability can be mapped into a logical relationship between synthesis and miscibility gap temperatures. Here, we show that DEED accelerates the computational discovery of novel disordered ceramics by allowing rapid prescreening, avoiding expensive experimental preparation, and enabling structure and property predictions [Kaufmann *et al.*, npj Comput. Mater. 6, 42 (2020); Hart *et al.*, Nat. Rev. Mater. 6, 730 (2021)].

Q2.9. *Considering all of those limitations, the descriptor of DEED does not have a general and strong value for guiding the synthesis of multicomponent ceramics.*

I regret to summarize that I feel this work does not represent a clear and significant advancement for the development of multicomponent ceramics. It shows one improved descriptor for classification of formation of single vs. multi-phases within a limited range of material design space. The work does not represent a strong breakthrough which should be the main criterion for a journal as Nature.

A2.9. We thank the referee for the comments which improved the manuscript. We hope that our response addresses all questions. We are certain that the new information adds clarity, precision, and relevance of DEED as a functional synthesizability descriptor for guiding the synthesis of multicomponent ceramics.

Reviewer Reports on the First Revision:

Referees' comments:

Referee #1 (Remarks to the Author):

In response to my original review, the authors have provided complete and highly suitable responses to my original concerns. Congratulations to the authors on their excellent work.
--Olivia A. Graeve

Referee #2 (Remarks to the Author):

I have carefully reviewed both the revised manuscript and the authors' response to the previous review comments. I find that the main criticisms from my initial review still apply. While the authors present lengthy arguments, they have not effectively addressed the fundamental weaknesses in their work. Therefore, I regret to conclude that this paper does not represent a clear and significant advancement for the development of high-entropy ceramics. It introduces an improved descriptor, but it falls short of meeting the criteria for publication in a journal like Nature, which typically seeks groundbreaking contributions.

As I emphasized in my previous review, the authors framed their work around the abstract and loosely defined concept of "synthesizability". In the revised manuscript, they acknowledge that "The lack of consensus about "synthesizability" in literature does not translate into an operative procedure for the synthesis...". To address this issue, they revise "synthesizability" to "functional synthesizability", which they use to convey that "synthesizability is a functional of both the material and the synthesis conditions". "Functional" is a mathematical term, and its exact definition varies depending on the subfield. According to Wikipedia, a functional often refers to the mapping of a function to the value of the function at a point. The newly introduced concept of "functional synthesizability" may potentially confuse most readers even more than the vague one of "synthesizability" did. Clearly, the authors' difficulty in defining "synthesizability" demonstrates a fundamental weakness in their work.

Furthermore, the authors add an assertion that "functional synthesizability is intrinsically connected to a given manufacturing process", i.e., hot-pressed sintering for transition-metal disordered ceramics in this work. However, they later state that "different processes, structures, and chemistries can be captured by the DEED framework". To validate the broad applicability of the DEED framework, it should be demonstrated for various processing routes and compared with hot-pressed sintering processes. I will not further iterate other concerns I raised in my previous review.

To summarize, the DEED descriptor appears to provide improved classification of early transition metal element ceramics, primarily in terms of single versus multi-phase formation. Its utility is limited, and it lacks the generality needed to guide the development of new high-entropy ceramics. Consequently, the work does not have sufficient general relevance and potential impact to warrant publication in Nature.

Author Rebuttals to First Revision:

Referee #1 (Remarks to the Author)

Q1.1. *In response to my original review, the authors have provided complete and highly suitable responses to my original concerns. Congratulations to the authors on their excellent work.*
--Olivia A. Graeve

A1.1. We are grateful for Dr. Graeve's time and positive feedback. We are delighted that both the manuscript and the authors received such a high praise. Thank you!

Referee #2 (Remarks to the Author)

Q2.1. *I have carefully reviewed both the revised manuscript and the authors' response to the previous review comments. I find that the main criticisms from my initial review still apply. While the authors present lengthy arguments, they have not effectively addressed the fundamental weaknesses in their work. Therefore, I regret to conclude that this paper does not represent a clear and significant advancement for the development of high-entropy ceramics. It introduces an improved descriptor, but it falls short of meeting the criteria for publication in a journal like Nature, which typically seeks groundbreaking contributions.*

A2.1. We thank the Referee for the time invested in evaluating the revised manuscript and responses. We respectfully disagree with the opinion about the manuscript.

Q2.2. *As I emphasized in my previous review, the authors framed their work around the abstract and loosely defined concept of "synthesizability". In the revised manuscript, they acknowledge that "The lack of consensus about "synthesizability" in literature does not translate into an operative procedure for the synthesis...". To address this issue, they revise "synthesizability" to "functional synthesizability", which they use to convey that "synthesizability is a functional of both the material and the synthesis conditions". "Functional" is a mathematical term, and its exact definition varies depending on the subfield. According to Wikipedia, a functional often refers to the mapping of a function to the value of the function at a point. The newly introduced concept of "functional synthesizability" may potentially confuse most readers even more than the vague one of "synthesizability" did. Clearly, the authors' difficulty in defining "synthesizability" demonstrates a fundamental weakness in their work.*

A2.2. We respectfully disagree. In our previous detailed report, we provided evidence to address all of the referee's concerns. Still, to avoid any potential confusion, we revised the paragraph discussing "functional synthesizability" (first page second column, blue text is the modified part). It now reads:

Defining functional synthesizability. Synthesizability depends on both the material and the synthesis conditions, and the user needs to explore the appropriate path across these two degrees of freedom. The conundrum lies in the observer's reference. Usually, the priority is put on "synthesizability as an intrinsic material property" with synthesis conditions to be determined later; i.e., from the materials' point of view. Unfortunately, this approach, along with the lack of consensus about "synthesizability" [28–35], does not translate into

an operative procedure for the synthesis, critical for autonomous materials discovery [36]. Instead, to maximize the outcome, we focus on the complementary direction, or the process' point of view; i.e., starting from a chosen process, we build a descriptor to establish which materials are synthesizable with it. We define this as "functional synthesizability", to specify that it is intrinsically a function of the chosen manufacturing process. In the present case — transition-metals disordered ceramics — the process is hot-pressed sintering with all its various implementations [2], and the classes of materials are high-entropy carbides, carbonitrides and borides.

We have also added a line in the "conclusion". It now reads:

We have introduced DEED, a descriptor that captures the balance between entropy gains and enthalpy costs upon the formation of homogeneous solid-solutions. We have also introduced the concept of functional synthesizability associated to the process' point of view. Our experimental results, combined with published data, confirm DEED as a reliable tool enabling computational discovery of novel high-entropy ceramics.....

For our definition of "functional synthesizability", we dismiss Wikipedia for its lack of scientific authority.

Q2.3. *Furthermore, the authors add an assertion that "functional synthesizability is intrinsically connected to a given manufacturing process", i.e., hot-pressed sintering for transition-metal disordered ceramics in this work. However, they later state that "different processes, structures, and chemistries can be captured by the DEED framework". To validate the broad applicability of the DEED framework, it should be demonstrated for various processing routes and compared with hot-pressed sintering processes. I will not further iterate other concerns I raised in my previous review.*

A2.3. We respectfully disagree since do not find any contradictions between the two assertions. In addition, regarding experiments, Referee 1, a world expert in synthesis of ceramics, has commended the authors for their "excellent" work, stating that all of her concerns have been satisfactorily answered. We did not address any Referee 2's experimental concerns because there were none in the previous review.

Q2.4. *To summarize, the DEED descriptor appears to provide improved classification of early transition metal element ceramics, primarily in terms of single versus multi-phase formation. Its utility is limited, and it lacks the generality needed to guide the development of new high-entropy ceramics. Consequently, the work does not have sufficient general relevance and potential impact to warrant publication in Nature.*

A2.4. We thank the Referee for the time invested in evaluating the manuscript. We respectfully disagree with the final opinion.

Regards,

Stefano Curtarolo